# S-nitrosylation of EZH2 alters PRC2 assembly, methyltransferase activity, and EZH2 stability to maintain endothelial homeostasis

Ashima Sakhuja [1,3], Ritobrata Bhattacharyya[1,3], Yash Tushar Katakia [1], Shyam Kumar Ramakrishnan[1], Srinjoy Chakraborty [1], Hariharan Jayakumar[1], Shailesh Mani Tripathi[2], Niyati Pandya Thakkar[1], Sumukh Thakar[1], Sandeep Sundriyal [2], Shibasish Chowdhury [1] & Syamantak Majumder [1] ✉

Nitric oxide (NO), a versatile bio-active molecule modulates cellular functions through diverse mechanisms including S-nitrosylation of proteins. Herein, we report S-nitrosylation of selected cysteine residues of EZH2 in endothelial cells, which interplays with its stability and functions. We detect a significant reduction in H3K27me3 upon S-nitrosylation of EZH2 as contributed by the early dissociation of SUZ12 from the PRC2. Moreover, S-nitrosylation of EZH2 causes its cytosolic translocation, ubiquitination, and degradation. Further analysis reveal S-nitrosylation of cysteine 329 induces EZH2 instability, whereas S-nitrosylation of cysteine 700 abrogates its catalytic activity. We further show that S-nitrosylation-dependent regulation of EZH2 maintains endothelial homeostasis in both physiological and pathological settings. Molecular dynamics simulation reveals the inability of SUZ12 to efficiently bind to the SAL domain of EZH2 upon S-nitrosylation. Taken together, our study reports S-nitrosylation-dependent regulation of EZH2 and its associated PRC2 complex, thereby influencing the epigenetics of endothelial homeostasis.

Polycomb repressive complex 2 (PRC2) is a chromatin-modifying group of proteins that works as repressor of gene expression. PRC2 is formed by three major classes of proteins; the functional enzymatic component of the PRC2, Enhancer of Zeste Homolog-2 (EZH2); suppressor of zeste 12 (SUZ12), a PRC2 subunit and a scaffolding protein, and embryonic ectoderm development (EED) which assembles and stabilizes the PRC2 complex[1,2]. The importance of EED and SUZ12 in maintaining the histone methyltransferase (HMT) activity of EZH2 and in the stability of the PRC2 complex is well studied[3,4]. EZH2 is responsible for methylating H3K27 resulting in the repression of gene expression[2,5]. Many studies have highlighted the importance of EZH2 as a crucial player in mediating cellular differentiation during development and its role in various disease conditions[6]. Moreover, EZH2 mediates gene silencing in embryonic stem cells, thereby promoting pluripotency, and in contrast also in the activation of differentiation[6,7]. Several reports have highlighted the role of EZH2 in cancer, wherein it regulates gene expression by depositing methylation marks to suppress gene expression which in turn contributes to cancer progression and metastasis[8,9]. Indeed, EZH2 dependent suppression of the tumor suppressor genes during cancer progression promotes tumorigenesis[1,2].

The regulation of the methyltransferase activity of EZH2 by the post-translational modifications (PTMs) such as ubiquitination[10],

[1]Department of Biological Sciences, Birla Institute of Technology and Science (BITS) Pilani, Pilani Campus, Rajasthan Pilani, India. [2]Department of Pharmacy, Birla Institute of Technology and Science (BITS) Pilani, Pilani Campus, Rajasthan Pilani, India. [3]These authors contributed equally: Ashima Sakhuja, Ritobrata Bhattacharyya. ✉e-mail: syamantak.majumder@pilani.bits-pilani.ac.in

phosphorylation[11], O-GlcNAcylation[12], sumoylation[13], and methylation[14] have been studied extensively. Ubiquitination of EZH2 has been reported to affect its catalytic activity, resulting in its degradation[10]. Phosphorylation of EZH2 by AKT[3] and AMPK[4] leads to the suppression of its methyltransferase activity[15]. Serine O-GlcNAcylation in the SET domain of EZH2 is also associated with regulation of its methyltransferase activity[16]. Similarly, several other PTMs at specific residues of the EZH2 may regulate its stability and catalytic activity. However, S-nitrosylation of EZH2 and its role in regulating the catalytic activity, localization, and degradation of this critical epigenetic modifier has not been reported till date.

S-Nitrosylation is one of the scarcely studied PTMs known to regulate protein function, stability, and cellular localization. It occurs through NO, a versatile free radical that non-canonically forms S-NO by selective modification of the protein at cysteine/methionine residues and mediates numerous biological functions[17]. Moreover, many cells harbor endogenous NO producing machinery including NO synthase (NOS) class of enzymes. The NOS family of enzymes utilizes L-arginine to endogenously produce NO which plays diverse roles in different cell types[18]. Endothelial cells (EC) contain a very cell-type-specific NO producing enzyme known as endothelial NOS (eNOS). eNOS-dependent release of NO mediates various signaling cascade that are essential for endothelial migration, survival, and growth. Indeed, eNOS-driven release of NO uses S-nitrosylation-dependent regulation of proteins to govern endothelial functions[19–22]. However, the mechanism highlighting the role of eNOS-dependent release of NO in enabling alteration in gene expression in EC through chromatin regulation remains elusive.

In this study, we addressed whether NO-mediated post-translational modification of HMT protein EZH2 could influence its catalytic activity, localization, and stability. Through the present study, we establish that S-nitrosylation of cysteine residue(s) in EZH2 protein amends its function as an epigenetic modulator, thereby proving the role of NO as a direct modulator of epigenetic processes and further regulating gene expression changes. Our study connects the link between eNOS-dependent NO release and its function as an epigenetic modulator through regulation of EZH2 to alter chromatin structure and thereby dictate NO dependent gene expression changes.

## Results

### SNP/GSNO interplayed with EZH2 including PRC2 assembly, its methyltransferase activity, subcellular localization, stability and further altering its binding partners

NO is well-reported to regulate gene expression changes in EC[22]. However, whether such an effect is dependent on epigenetic processes specifically via regulation of EZH2 and PRC2 complex has not been reported. We therefore used Sodium Nitroprusside (SNP) as an external nitric oxide donor. We quantified cellular nitrite level using Griess assay which revealed a time dependent increase in cellular nitrite level upon SNP challenge to transformed endothelial cell line EA.hy926 cells (Supplementary Fig. 1A). However, SNP did not alter the level of DNA damage response gene γH2AX in EA.hy926 cells (Supplementary Fig. 1B). Cell viability and cellular apoptosis also remained unchanged upon SNP exposure to EA.hy926 cells (Supplementary Figs. 1C, D). Parallel to observations in EA.hy926 cells, we also confirmed time dependent elevation in cellular nitrite level in HUVEC upon SNP exposure (Supplementary Fig. 1E). Interestingly, unlike EA.hy926 cells, HUVEC treated with SNP for different time points manifested a temporal increase in DNA damage response gene γH2AX (Supplementary Fig. 1F) suggesting a cell type dependent effect of cellular nitrite. Next, we assessed the effect of SNP on EZH2 protein and its catalytic product H3K27me3 level in EC. Notably, SNP exposure resulted in a time-dependent reduction in the level of EZH2 protein specifically, revealing a significant reduction at 2 h (Fig. 1A). We also observed a significant reduction in the catalytic product H3K27me3.

However, the reduction in H3K27me3 was detected within 1 h of SNP treatment, much earlier than the degradation of EZH2 (Fig. 1B). To confirm if such an effect of NO on EZH2 is exerted through transcriptional regulation, we performed qPCR analysis of EZH2 transcripts which revealed no alteration in EZH2 mRNA level upon SNP exposure (Supplementary Fig. 2A).

We next confirmed the changes in EZH2 and H3K27me3 levels ex vivo using rat aortic rings exposed to SNP. Comparable to the observations of the in vitro studies, diminished EZH2 and H3K27me3 levels were detected in rat aorta exposed to SNP (Fig. 1C). Because EC possess endogenous NO-producing machinery namely eNOS, we next questioned whether activation of endogenous NO-production machinery would also affect the levels of EZH2 and H3K27me3. We thus used two independent yet potent inducers of eNOS namely bradykinin and Vascular Endothelial Growth Factor (VEGF). Induction of EC by bradykinin resulted in a dose- and time-dependent reduction in EZH2 and H3K27me3 levels in HUVEC (Supplementary Fig. 2B–E) and time-dependent depletion in EZH2 and H3K27me3 levels in EA.hy926 cells (Supplementary Fig. 2F). Moreover, rat aorta without the endothelial layer when induced with bradykinin did not exhibit reduction in EZH2 or H3K27me3 level (Supplementary Fig. 2G, H) suggesting bradykinin driven induction of endothelial specific eNOS machinery is key to alteration in EZH2 and H3K27me3 level. Furthermore, VEGF treatment reduced EC EZH2 and H3K27me3 levels in a time dependent manner in both EA.hy926 cells (Supplementary Fig. 3A–C) as well as HUVEC (Supplementary Fig. 3D). To understand the cell-specific effect of SNP on EZH2 and H3K27me3, we next exposed HEK-293, a non-endothelial human cell type, to NO donor and measured the levels of EZH2 and H3K27me3 and noted similar result to that of EC (Supplementary Fig. 3E, F). Hence, we established that SNP exposure caused a reduction in EZH2 and associated H3K27me3 levels independent of the cell type. We further evaluated the effect of SNP on EZH2 overexpressed through plasmid construct and confirmed that SNP exposure also affects the level of overexpressed EZH2 as detected through HA-tag or by detecting the total EZH2 (Supplementary Fig. 3G, H).

As observed through our data, a reduction in H3K27me3 level upon SNP exposure much before the EZH2 degradation was rather intriguing. The association of SUZ12 with EZH2 is essential for maintaining the catalytic activity of the EZH2-PRC2 complex and SUZ12 deficient cells exhibited diminished activity of the PRC2/3 complexes[23]. Because we observed a reduction in H3K27me3 level much earlier than the EZH2 degradation, we next questioned whether SNP exposure interplayed with the association of EZH2 and SUZ12 at an early time point which further led to diminished catalytic activity of the EZH2-PRC2 complex much before the EZH2 degradation. We therefore performed a co-immunoprecipitation experiment using EZH2 antibody and observed that SUZ12 dissociate from EZH2 in EC exposed to SNP for 30 min (Fig. 1D). All other components of the PRC2 complex remain associated with EZH2 at least until 30 min of SNP exposure to EC (Fig. 1D). Interestingly, the protein level of other components of PRC2 including SUZ12, AEBP2, EED, and JARID2 in total cell lysates remained unaltered in SNP challenged EC (Supplementary Fig. 4A). Therefore, this set of data suggests the following time-dependent and critical observations in EC exposed to SNP/VEGF/bradykinin exposed EC which are as follows; (i) Dissociation of SUZ12 from the EZH2-bound PRC2 complex by 30 min, (ii) significant reduction in H3K27me3 level by 1 h, likely due to the presence of the demethylases UTX and JMJD3, and (iii) EZH2 degradation by 2 h.

Being a histone-modifying enzyme and part of the PRC2 complex, EZH2 is primarily localized in the nucleus. However, previous reports have highlighted cytosolic translocation of EZH2 and also noted its cytosolic substrates such as Talin[24] and small GTPases[25] to cause actin polymerization. We thus wanted to explore the effect of SNP exposure on EZH2 localization. Subcellular fractionation (Fig. 1E) and

 

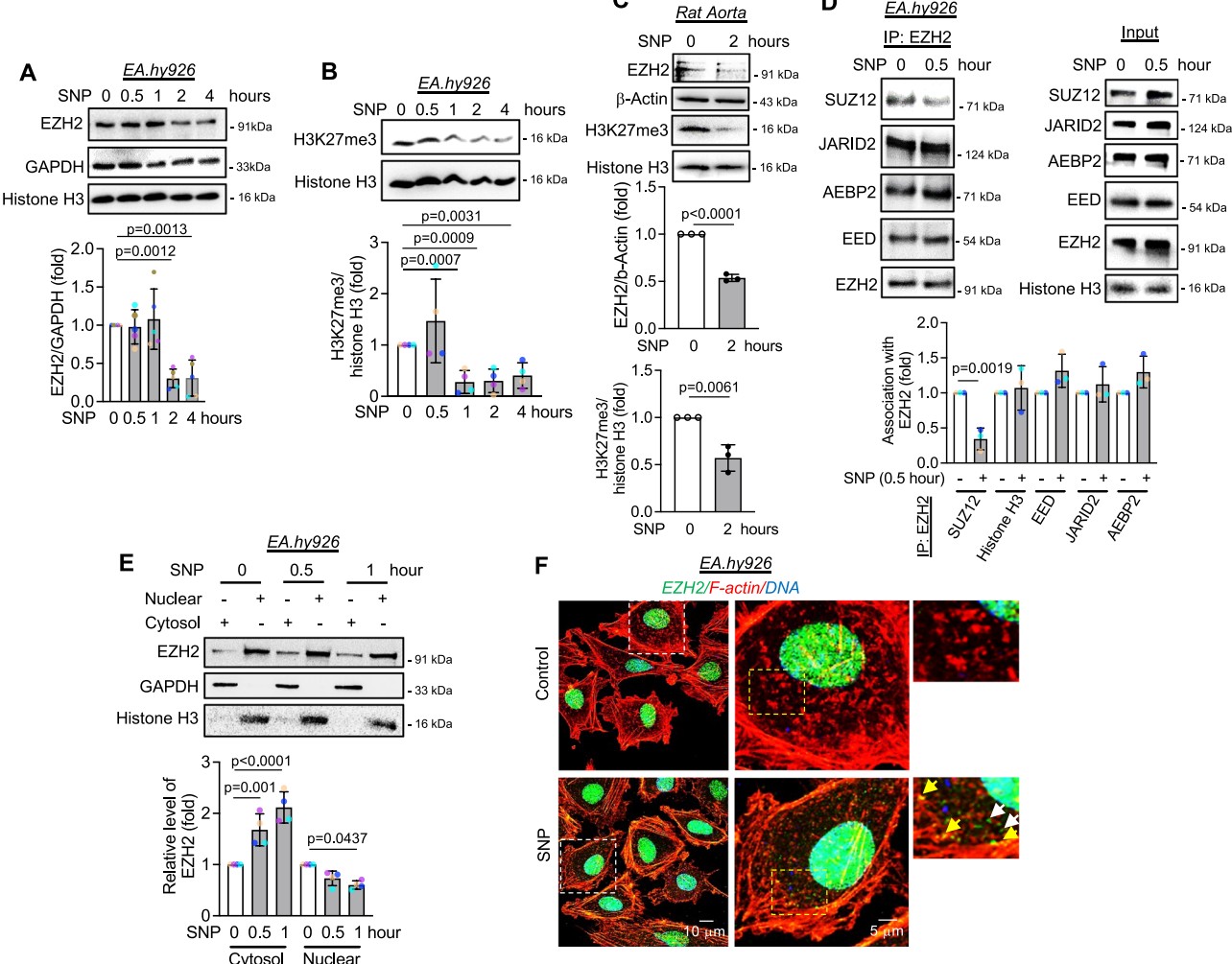

**Fig. 1 | Nitric oxide caused cytosolic localization and degradation of EZH2 protein accompanied with an early reduction in H3K27me3 levels due to dissociation of SUZ12. A, B** Immunoblots analysis of EZH2 ($n = 4$, biological replicate) and H3K27me3 ($n = 5$, biological replicate) levels in EA.hy926 cells exposed to SNP (500 μM) at variable time points (0, 0.5, 1, 2, and 4 h). **C** Immunoblot analysis using lysate of rat aortic explants exposed to SNP (500 μM) for 2 h. ($n = 3$, biological replicate). **D** EA.hy926 cells treated with SNP (500 μM) for 0.5 h were subjected to co-immunoprecipitation using EZH2 antibody followed by immunoblotting to show the association of EZH2 with other subunits of the PRC2 complex. ($n = 3$, biological replicate). **E** Immunoblotting analysis of EZH2 in nuclear and cytosolic fractions obtained from EA.hy926 cells treated with SNP (500 μM for 0, 0.5, and

1 h). GAPDH and histone H3 were used to confirm cytosolic and nuclear fractions respectively. ($n = 3$, biological replicate) **(F)** Immunofluorescence followed by confocal imaging of EA.hy926 cells exposed to 1 h of SNP to show cytosolic translocation of EZH2 (green). F-actin (red) is stained with phalloidin-Alexa Fluor 555. DAPI staining is shown in blue (Scale bar: 10 μm). White arrow head indicates the presence of cytosolic EZH2 not bound to F-actin while yellow arrow head shows EZH2 co-localized with F-actin. ($n = 3$, images are representative of biological replicate). All data are presented as mean values ± SD. All statistical analyses are either performed by One-way ANOVA with a post-hoc Tukey test for multiple groups or by two-tailed unpaired $t$-test for two groups.

immunofluorescence (Fig. 1F) followed by confocal imaging strongly indicated cytosolic translocation of EZH2 in EC upon SNP exposure. Moreover, confocal imaging also indicated the recruitment of EZH2 protein to the filamentous actin (Fig. 1F), which confirms what others have reported. Interestingly, the turnover of H3K27me3 is not only dependent on the methyltransferase EZH2 but also H3K27me3-specific demethylases such as UTX and JMJD3. We thus detected the levels of UTX and JMJD3 protein in EC exposed to SNP or GSNO. UTX and JMJD3 protein levels remained unaltered in EC exposed to SNP or GSNO for 2 h (Supplementary Fig. 4B–D).

To exclude the possibility of the effect of SNP on EZH2 as non-specific, we next used S-nitrosoglutathione (GSNO), which is more well-accepted as an inducer to cause S-nitrosylation of cellular proteins. As a result, we found a similar reduction in the level of EZH2 after 2 h of GSNO exposure to EA.hy926 cells (Fig. 2A) or HUVEC

(Supplementary Fig. 4E). Like SNP-treated EC, further analysis of H3K27me3 in the GSNO treated cells revealed a time-dependent depletion of H3K27me3 by 1 h which remained significantly low at least up to 2 h post-GSNO treatment to EA.hy926 cells (Fig. 2A) or HUVEC (Supplementary Fig. 4F). GSNO significantly reduced EZH2 and H3K27me3 levels in rat aorta when treated ex vivo (Fig. 2B, C). In parallel to SNP, GSNO also promoted cytosolic localization of endogenous EZH2 in EC (Fig. 2D, E) or overexpressed EZH2 WT in HEK293 cells (Supplementary Fig. 5A). We next performed cycloheximide chase experiment in the presence of GSNO and detected the level of EZH2 when cellular translation was compromised. Absence of translation-dependent replenishment of EZH2 protein in GSNO treated cells revealed a significant reduction in EZH2 protein level within 0.5 and 1 h which we failed to detect in translationally active normal cells (Supplementary Fig. 5B). Hence, this data confirmed the quick

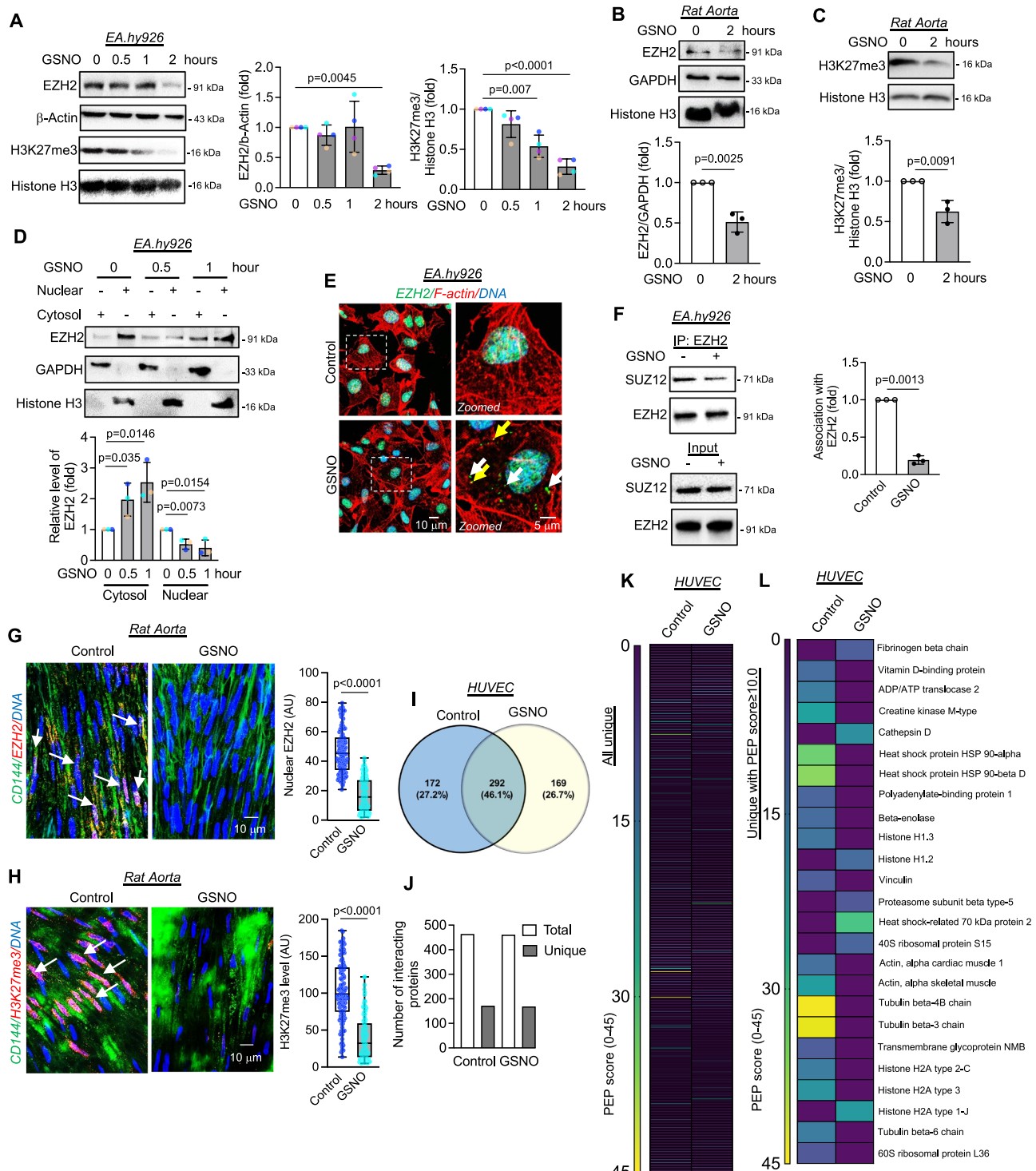

turnover of EZH2 protein through degradation pathways upon S-nitrosylation. GSNO could not alter the cellular localization of other proteins of the PRC2 complex, including SUZ12, AEBP2, EED, and JARID2 (Supplementary Fig. 6A–C). Moreover, we also confirmed the association of EZH2 and SUZ12 when EC were exposed to GSNO for 0.5 h. In parallel to SNP and bradykinin, the GSNO challenge restrained the association of EZH2 with SUZ12 (Fig. 2F).

To confirm these findings in tissue, we further performed *en face* staining of the rat aorta to evaluate the level and localization pattern of EZH2 in EC within the tissue. Through imaging, we identified that EZH2 localization in EC of the aorta was altered upon GSNO exposure. Notably, the level of nuclear EZH2 significantly diminished in tissues

treated with GSNO for 2 h (Fig. 2G). In addition, the H3K27me3 level also declined in the EC of the aorta exposed to GSNO (Fig. 2H). We next wanted to explore the comprehensive interacting partners of EZH2 upon S-nitrosylation to understand the effect of such PTMs on its methyltransferase activity, translocation, and degradation. Mass spectrometric analysis revealed the comprehensive association map of EZH2 in control and GSNO treated cells (Fig. 2I). A total of 464 proteins were identified to be associated with EZH2 while 461 different proteins associated with EZH2 upon GSNO exposure (Fig. 2I, J). Out of 464 proteins in control condition, 172 unique proteins were associated with EZH2 which completely dissociated upon GSNO exposure. In contrast, 169 (out of 461 in total) unique proteins were bound to EZH2 upon

**Fig. 2 | GSNO exposure caused reduction in EZH2 and H3K27me3 level along with altering the interacting partners of EZH2. A** Immunoblotting for EZH2 and H3K27me3 in cultured EA.hy926 cells exposed to GSNO (100 µM). ($n$ = 4, biological replicate) **B, C** Immunoblot analysis of EZH2 (**B**) and H3K27me3 (**C**) in rat aortic explants exposed to GSNO (100 µM). ($n$ = 3, biological replicate) **D** Immunoblotting analysis of EZH2 in nuclear and cytosolic fractions obtained from EA.hy926 cells treated with GSNO (100 µM). ($n$ = 3, biological replicate) **E** Immunofluorescence followed by confocal imaging of EA.hy926 cells exposed to 1 h of GSNO (100 µM) and labeled for EZH2 (green) and F-actin (red). DAPI staining is shown in blue. White and yellow arrow head indicates EZH2 not bound and bound to F-actin respectively. (Scale bar: 10 µm), $n$ = 3, biological replicate. **F** EA.hy926 cells treated with GSNO (100 µM) for 0.5 h were subjected to co-immunoprecipitation using EZH2 antibody followed by immunoblotting for SUZ12. ($n$ = 3, biological replicate) **G, H** *En face* preparations were double stained for VE-cadherin (green) and EZH2 (red, **G**) or

H3K27me3 (red, **H**). Images were captured from the luminal surface of the aorta (Scale bar: 10 µm). The levels of nuclear EZH2 and H3K27me3 were analyzed using Image J. Both EZH2 (**G**) and H3K27me3 (**H**) analysis, control ($n$ = 112) and GSNO ($n$ = 113) from three independent biological replicate. Middle: Box whisker plots show nuclear EZH2, with minimum, first quartile (lower bound), median, third quartile (upper bound), maximum values, whiskers down to the minimum and up to the maximum value. **I–L** Venn-diagram (**I**) representing the percentage of common/overlapping proteins in control and 0.5 h GSNO (100 µM) exposed HUVEC. Data (**J**) showing the number of total and unique interacting proteins. Heat map-based visualization of all the uniquely interacting proteins (**K**) or proteins with a PEP score ≥ 5. ($n$ = 2, biological replicate). All data are presented as mean values ± SD. All statistical analyses are either performed by One-way ANOVA with a post-hoc Tukey test for multiple groups or by two-tailed unpaired $t$-test for two groups.

GSNO exposure which were not detected to be associated with EZH2 in untreated cells (Fig. 2I, J).

We next performed a close analysis of the interactome data which indicated a few interesting association patterns of EZH2 in untreated and GSNO treated conditions. Firstly, as observed through co-immunoprecipitation experiment, EZH2 was found to be associated with histone H3 in untreated cells, however, such association was not detected upon GSNO exposure. Surprisingly, nuclear-to-cytosolic shuttling protein 14-3-3 epsilon was found to be associated with EZH2 in untreated conditions while a complete loss of association was observed upon GSNO challenge (Fig. 2K and Supplementary Datas 1 and 2). Although, we expected such association with 14-3-3 epsilon to be prominent upon GSNO exposure due to the shuttling of EZH2 after S-nitrosylation, but we observed an opposite correlation. Such data indicated that EZH2 may be using 14-3-3 epsilon during conventional cytosolic localization while its cytosolic localization upon S-nitrosylation is likely to be driven by other unknown factors. More interestingly, upon GSNO exposure, we detected EZH2 association with many proteins of the endosome/lysosome/proteasome pathway including several Rab family of proteins, HSP70&90 chaperon proteins, cathepsin D lysosomal protease, lysosome-associated membrane glycoprotein 1, proteasome subunit beta type-5/alpha type-4, lysosomal acid ceramidase (Fig. 2K, L and Supplementary Datas 1 and 2). EZH2 was not found to be associated with these proteins in control conditions. Because we observed alteration in EZH2 association with EZH2 upon 30 min of SNP/GSNO challenge, we therefore decided to analyze another parameter obtained through the MS evaluation wherein the relative abundance of each of the EZH2-interacting proteins was provided. We normalized the abundance level of each of the PRC2 complex proteins including SUZ12, EED, AEBP2, and JARID2 to that of the EZH2 protein to obtain an EZH2 normalized ratio. In so doing, we observed an interesting pattern. SUZ12 association was reduced by 30% upon GSNO exposure as also observed through our co-immunoprecipitation studies (Supplementary Fig. 7A). Furthermore, in the MS data, EED and AEBP2 association with EZH2 remained unchanged in accordance with our co-immunoprecipitation studies in SNP-treated EC (Supplementary Fig. 7A). Surprisingly, in the MS data, we also detected a diminished association of JARIDI2 with EZH2 upon GSNO challenge (33% reduction, Supplementary Fig. 7A) which was not observed in our co-immunoprecipitation studies at least in SNP-treated EC. Further, to confirm a part of these findings, we performed co-immunoprecipitation experiment using EZH2 followed by analyzing its association with 14-3-3 epsilon in GSNO exposed EC. In doing so, we confirmed a significant loss of the association of EZH2 with 14-3-3 epsilon upon GSNO exposure (Supplementary Fig. 7B).

### SNP/GSNO exposure caused S-Nitrosylation of EZH2 leading to early SUZ12 dissociation and diminished catalytic activity
In the non-canonical NO signaling pathway, many proteins are post-translationally modified through nitrosylation of cysteine/methionine/

tyrosine residue thereby regulating the function of such proteins. We, therefore, questioned whether such could be the case with EZH2 as well. Thus, after establishing the effect of NO exposure on the methyltransferase activity, localization, and degradation of EZH2, we then evaluated whether NO post-translationally modified EZH2 through S-nitrosylation thereby regulating its function. Previous studies reported regulation of EZH2 localization and function via phosphorylation of distinct residues. S-nitrosylation of EZH2 protein has not been reported earlier and therefore we first assessed whether EZH2 is S-nitrosylated upon NO exposure. Through the iodoTMT assay using truncated recombinant EZH2 protein (carrying residues from 429 to 728 aa) in a cell-free system, we first confirmed that NO exposure caused S-nitrosylation of the truncated form of the EZH2 protein (Supplementary Fig. 8). Further to confirm this in a cellular system, EC were exposed to SNP followed by biotin switch assay and immunoprecipitation using EZH2 antibody. Such an experiment also demonstrated S-nitrosylation of EZH2 protein in EC subjected to NO treatment (Fig. 3A, B). Further to ascertain the S-nitrosylation, we next used a pan S-nitrosylated antibody and confirmed through an immunoprecipitation followed by an immunoblot experiment and showed that EZH2 in EC was indeed S-nitrosylated upon NO exposure (Fig. 3C).

Next, we performed a co-immunoprecipitation experiment using EC which were exposed to bradykinin, a natural inducer of endogenous NO production. In doing so, we detected a robust S-nitrosylation of EZH2 protein concurrent with the loss of EZH2 binding with SUZ12 of the PRC2 complex (Fig. 3D). Furthermore, to support this data, we also analyzed the S-nitrosylation of HA-tagged EZH2 protein in a HEK-293 overexpression system. Through a co-immunoprecipitation experiment using HA antibody, we again confirmed S-nitrosylation of overexpressed EZH2 protein upon exposure to NO (Fig. 3E).

Next, we also assessed the S-nitrosylation of EZH2 in EC exposed to GSNO. The iodo-TMT assay confirmed increased S-nitrosylation of EZH2 when the EC nuclear fractions were treated with GSNO (Fig. 3F, G). Because showing a reduction in the H3K27me3 level may not alone prove the compromised methyltransferase activity of the PRC2 complex upon SNP or GSNO treatment, we performed an H3K27me3 HMT activity assay with the nuclear fractions of EC exposed to SNP or GSNO. This assay revealed a significant reduction in methyltransferase activity of the PRC2 complex in the nuclear fraction when incubated with SNP or GSNO (Fig. 3H, I). Taken together, these data suggest that S-nitrosylation of EZH2 caused early dissociation of SUZ12 from the PRC2 complex thereby diminishing its methyltransferase activity.

### S-nitrosylation of EZH2 promoted its degradation primarily through autophagosome-lysosome pathway while inhibition of endogenous nitric oxide machinery reversed nitric oxide-dependent degradation, activity, and localization of EZH2
As we observed an association of many endosomal/lysosomal and proteasomal proteins, we therefore questioned the role of lysosomal

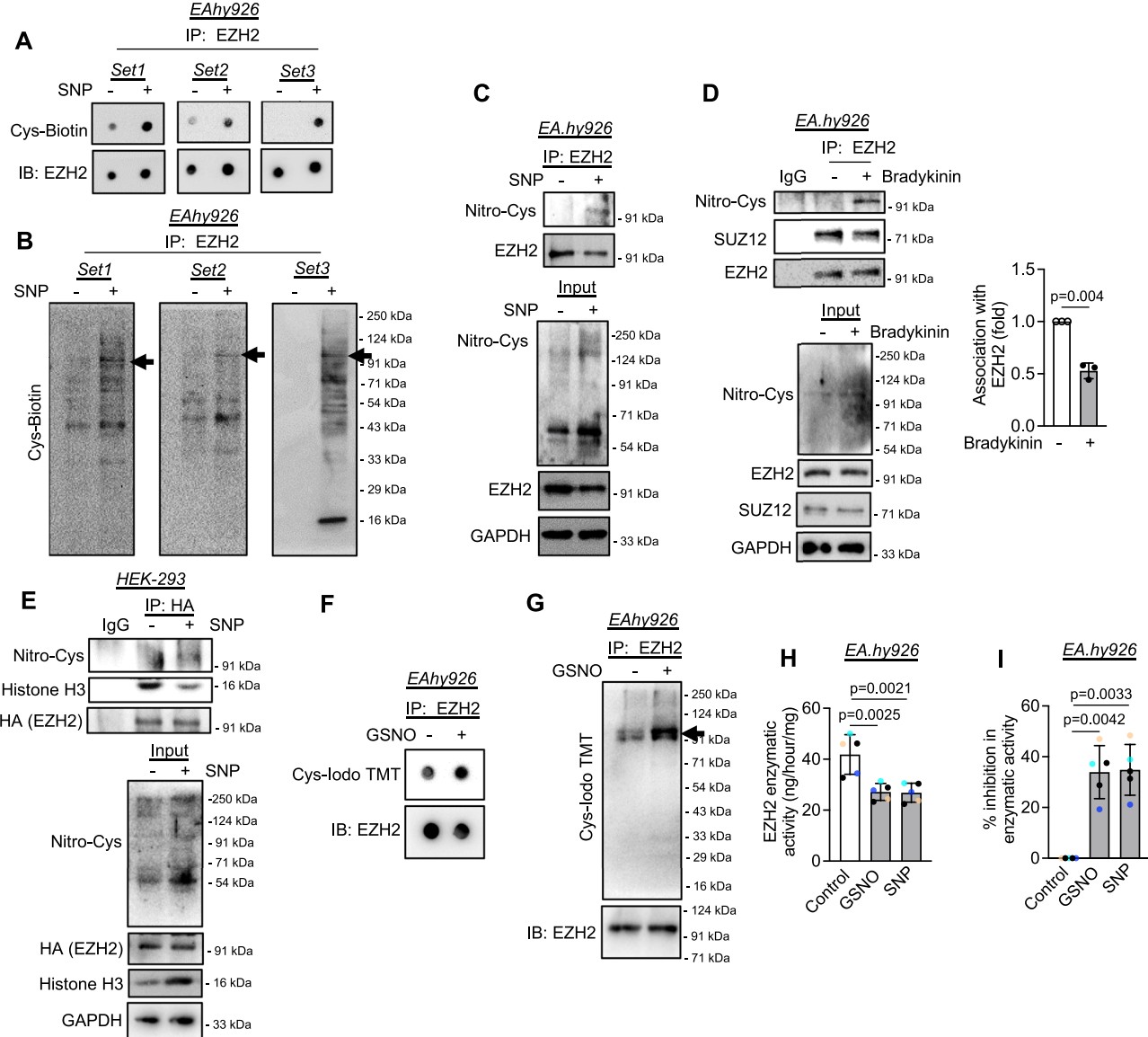

**Fig. 3 | External nitric oxide supplementation, endogenous induction of nitric oxide producing machinery, or treatment with GSNO caused S-nitrosylation of EZH2 and dissociation of SUZ12 and histone H3. A** Dot blot of protein lysates collected from EA.hy926 cells exposed to SNP (500 μM) for 0.5 h and processed through biotin-switch assay followed by immunoprecipitation with EZH2 antibody. Blots were incubated with Streptavidin-HRP followed by developing with chemiluminescence substrate for visualization. (*n* = 3, biological replicate) **B** Same samples were run through SDS-PAGE followed by transfer to nitrocellulose membrane followed by incubation with Streptavidin-HRP and developing the blot with chemiluminescence substrate for visualization. (*n* = 3) **C, D** EA.hy926 cells were exposed to either SNP (500 μM, **C**) or bradykinin (10 μM, **D**) for 0.5 h followed by immunoprecipitation with EZH2 antibody and further immunoblotting to show the presence of S-nitrosylation of EZH2 using Nitro-Cysteine antibody. (*n* = 3, biological replicate) **E** Plasmid containing HA-tagged EZH2 were transfected in HEK-293 cells followed by exposing to SNP (500 μM) for 0.5 h. Cell lysates were then immunoprecipitated using HA antibody followed by immunoblotting with respective antibodies. (*n* = 3, biological replicate) **F, G** Dot blot (**F**) and SDS-PAGE followed by immunoblot (**G**) of protein lysates collected from EA.hy926 cells exposed to GSNO (100 μM) for 0.5 h followed by immunoprecipitation with EZH2 antibody and processing through iodoTMT protocol. Blots were incubated with anti-IodoTMT antibody. Blots were developed with chemiluminescence substrate for visualization. (*n* = 3, biological replicate) **H, I** Histone methyltransferase activity (**H**) and inhibition (**I**) analysis of EZH2 in EA.hy926 cells exposed to GSNO (100 μM) or SNP (500 μM) for 2 h. (*n* = 5, biological replicate). All data are presented as mean values ± SD. All statistical analyses are either performed by One-way ANOVA with a post-hoc Tukey test for multiple groups or by two-tailed unpaired *t*-test for two groups.

and proteasomal pathways in the degradation of EZH2 upon S-nitrosylation. The role of post-translational modification-dependent regulation of EZH2 protein degradation is least studied. Because we observed EZH2 protein degradation upon NO exposure, we explored the key degradation machinery responsible for degrading the S-nitrosylated EZH2. Since ubiquitination of proteins plays a crucial role in their degradation and previous reports have indicated ubiquitination mediated degradation of EZH2[26], we first assessed the level of EZH2 ubiquitination upon SNP exposure and found that SNP challenge

caused a significant increase in ubiquitination of EZH2 protein (Fig. 4A). Most notably, cells use lysosomal and proteasomal degradation pathways for protein degradation; we thus used pharmacological inhibitors of proteasomal and autophagosome-lysosome pathway to evaluate their relative contribution towards degradation of the S-nitrosylated EZH2. Inhibition of proteasomal pathway using MG132 was unable to reverse the level of EZH2 upon SNP exposure (Fig. 4B). In contrast, inhibition of autophagosome-lysosome pathway using bafilomycin A showed a partial yet significant reversal of EZH2

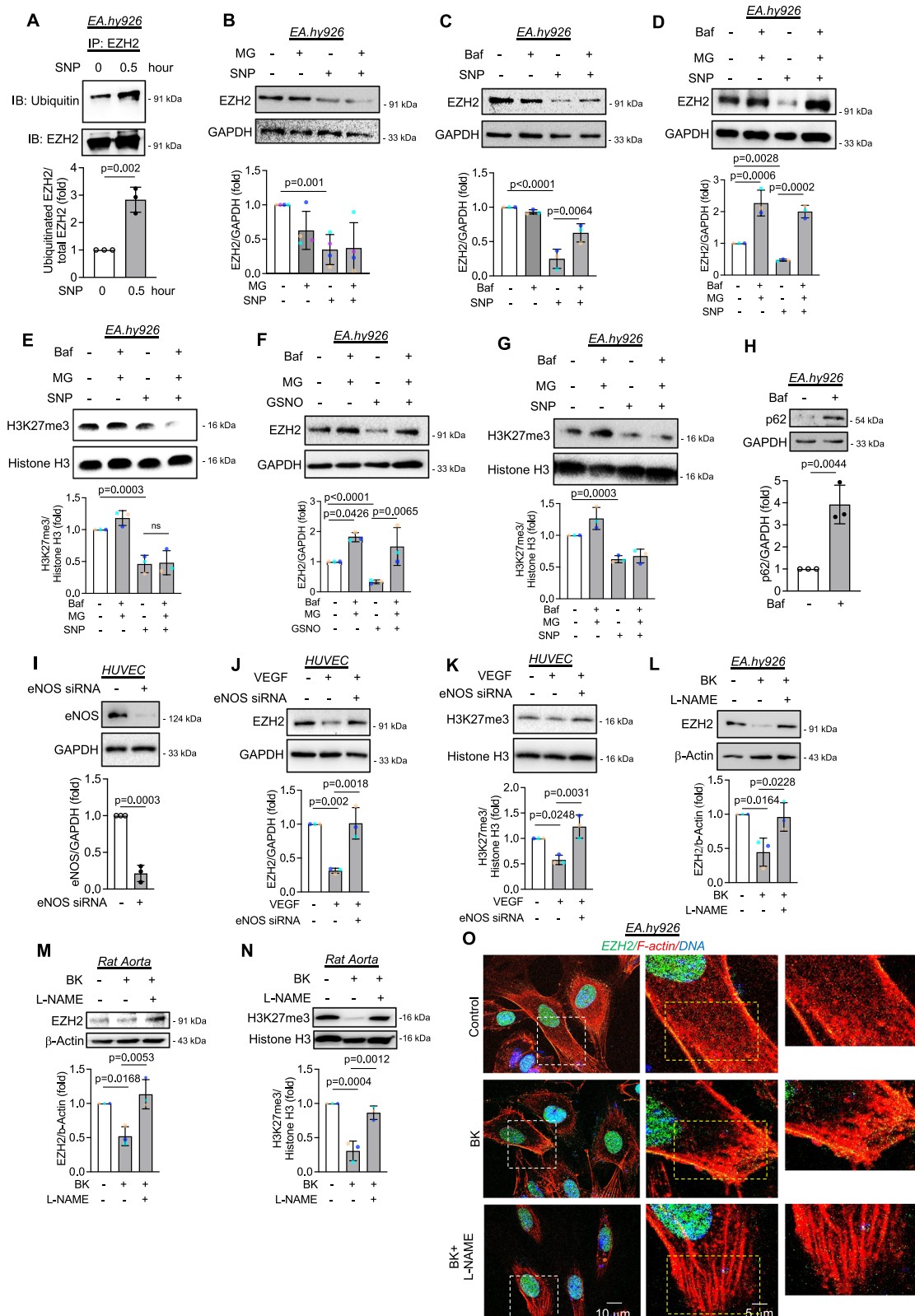

degradation upon SNP challenge (Fig. 4C). We then inhibited both proteasomal and autophagosome-lysosome pathway to evaluate the effect of such combined inhibition on EZH2 protein level. We then detected complete reversal of SNP dependent degradation of EZH2 protein when both proteasomal and autophagosome-lysosomal pathways were blocked (Fig. 4D). Interestingly, abrogation of proteasomal and autophagosome-lysosomal pathways was unable to reverse the

level of H3K27me3 in SNP treated EC (Fig. 4E). We observed a parallel effect wherein a combination of MG132 and bafilomycin A reversed GSNO-induced reduction in EZH2 level but failed to rescue the level of H3K27me3 (Fig. 4F, G). All these data indicated that degradation of EZH2 upon SNP or GSNO exposure primarily occurred through autophagosome-lysosome pathway. However, an inhibition of the autophagosome-lysosome pathway could likely switch S-nitrosylated

**Fig. 4 | SNP or GSNO challenge to endothelial cells caused degradation of EZH2 primarily through the autophagosome-lysosome pathway while blocking endogenous NO production machinery in bradykinin induced endothelial cells reversed EZH2 and H3K27me3 level. A** EA.hy926 cells exposed to SNP (500 µM) followed by co-immunoprecipitation with EZH2 and immunoblotted with ubiquitin antibody. ($n = 3$, biological replicate) **B** Immunoblotting analysis of EZH2 in EA.hy926 cells treated with MG132 (MG) (1 µM) and SNP (500 µM). ($n = 4$, biological replicate) **C** Immunoblotting for EZH2 in EA.hy926 cells treated with Bafilomycin A1 (Baf) (100 nM) and SNP (500 µM). ($n = 4$, biological replicate) **D, E** Immunoblotting EZH2 (**D**) and H3K27me3 (**E**) in cultured EA.hy926 cells treated with both MG132 (1 µM) and bafilomycinA1 (100 nM) followed by exposing to SNP (500 µM). ($n = 4$ for EZH2, and $n = 3$ for H3K27me3, biological replicate) **F, G** Immunoblotting EZH2 (**F**) and H3K27me3 (**G**) in cultured EA.hy926 cells pretreated with both MG132 (1 µM) and bafilomycinA1 (100 nM) followed by exposing to GSNO (100 mM). ($n = 3$, biological replicate) (**H**) Immunoblotting for p62 in EA.hy926 cells treated with bafilomycin A1 (100 nM). **I** Immunoblotting for eNOS in HUVEC transfected with eNOS siRNA. ($n = 3$, biological replicate) **J, K** Immunoblotting for EZH2 (**J**) and H3K27me3 (**K**) in HUVEC transfected with eNOS siRNA and induced with VEGF (10 ng/mL). ($n = 3$, biological replicate) (**L**) Immunoblotting for EZH2 in EA.hy926 cells treated with L-NAME (1 mM) and bradykinin (10 mM). ($n = 3$, biological replicate) **M, N** Rat aortic rings were treated with L-NAME (1 µM) and bradykinin (10 µM) followed by immunoblotting to analyze the EZH2 (**M**) and H2K27me3 (**N**). ($n = 3$, biological replicate) (**O**) EZH2 (green) along with F-actin (Red) staining of EA.hy926 cells after treating with L-NAME (1 µM) and bradykinin (10 µM). DAPI shown in blue. (Scale bar: 10 µm) ($n = 3$, biological replicate). All data are presented as mean values ± SD. All statistical analyses are either performed by One-way ANOVA with a post-hoc Tukey test for multiple groups or by two-tailed unpaired $t$-test for two groups.

EZH2 degradation through proteasomal pathway. More importantly, we also established that MG132- and bafilomycin A-driven rescue of EZH2 in SNP or GSNO treatment conditions failed to recover the dampened level of H3K27me3 due to limited catalytic activity of the rescued EZH2 protein which are still likely S-nitrosylated. Therefore, these S-nitrosylated EZH2 possess restricted catalytic activity to recover the H3K27me3 level in SNP or GSNO treated EC. Next, we questioned whether blocking the EZH2 degradation pathway allows enhanced cytosolic localization of EZH2 protein due to retention of S-nitrosylated EZH2 within cell cytosol. To analyze this, EC cells were pretreated with MG132 and bafilomycin A combination followed by challenging with SNP or GSNO for 2 h. In so doing, we observed that blocking the degradation pathways caused significant accumulation of EZH2 in the cytosol of EC in both SNP (Supplementary Fig. 9A) and GSNO treated conditions (Supplementary Figs. 9B, C). Successful inhibition of autophagosome-lysosomal pathway upon bafilomycin A treatment was confirmed through increase in p62 level (Fig. 4H).

We next questioned whether inhibition of endogenous NO producing machinery could alter the downstream effect of natural inducers of endogenous NO production machinery in EC such as VEGF or bradykinin. We thus used eNOS siRNA to knockdown eNOS in HUVEC cells followed by stimulation with VEGF for 2 h. We confirmed successful knockdown of eNOS (Fig. 4I) in HUVEC. As observed earlier, stimulation of EC with VEGF caused significant reduction in EZH2 and H3K27m3 level while eNOS siRNA knock down completely abrogated VEGF dependent reduction in in EZH2 and H3K27m3 (Fig. 4J, K). EC were exposed to L-NAME, a nonselective inhibitor of all types of NOS prior to inducing with bradykinin. As observed earlier in our study, bradykinin induction caused a reduction in EZH2 level which is completely reversed upon inhibition of NOS with L-NAME (Fig. 4L). We also performed the assay in ex vivo rat aorta model which further revealed reversal of EZH2 protein degradation and protection of H3K27me3 in aortic tissues exposed to L-NAME before bradykinin treatment (Fig. 4M, N). We then performed localization analysis of EZH2 in cells exposed to bradykinin alone or combined with L-NAME. As expected, bradykinin induction caused cytosolic translocation of EZH2, which was also found to be localized with actin cytoskeleton. However, inhibition of NOS family of protein in bradykinin treated EC using L-NAME restricted the localization of EZH2 to the nucleus (Fig. 4O).

## EZH2-H3K27me3 axis is enhanced in diabetic kidney disease in vivo while targeting this axis maintains endothelial homeostasis in physiological and pathological conditions in vitro and ex vivo

Regulation of cellular function by EZH2 primarily dependent on its regulation of gene expression changes through repressive H3K27me3 mark in the chromatin. Moreover, NO signaling pathway converges to changes in expression of genes associated with endothelial survival, proliferation and migration. We therefore wanted to investigate whether EZH2-dependent catalysis of H3K27me3 plays a role in dictating NO driven regulation of endothelial function and gene expression changes. To do so, we took a retrograde approach in which we pre-incubated the cells with GSK-J4, a selective inhibitor of H3K27me3-specific demethylases JMJD3 and UTX. We first confirmed that inhibition of JMJD3 and UTX reversed SNP dependent reduction in H3K27me3 level (Fig. 5A). We next performed endothelial migration using wound healing assay and observed that SNP-induced endothelial migration was abrogated upon preserving the level of H3K27me3 through inhibition of demethylases JMJD3 and UTX (Fig. 5B). To compare the effect of other modulators of the EZH2-H3K27me3 pathway including bradykinin, GSNO, EZH2 siRNA, and GSK126 (a selective small molecule inhibitor of EZH2) on endothelial migration, we performed scratch wound healing assay using monolayer of EC. A significant increase in endothelial migration was detected when cells were induced with bradykinin, GSNO, EZH2 siRNA, or GSK126 (Supplementary Fig. 10A). Therefore, such effect of SNP, bradykinin, or GSNO on endothelial migration were comparable with that of EZH2 siRNA and GSK126, which is likely due to the regulation of EZH2-H3K27me3 axis by these factors.

We next detected the transcript level expression of NO-responsive genes in EC exposed to SNP and/or GSK-J4 using qPCR assay. Based on previously reported data sets, we choose to detect the transcript level of the following NO responsive genes: *VEGFa*, *TBX20*, *MMP2*, *FGF2*, *KDR*, *TIE2*, *TEK*, and *Angiopoietin-2*[27]. Through this analysis, we detected a significant increase in the transcript levels of *VEGFa*, *TBX20*, *MMP2*, *FGF2*, and *Angiopoietin-2* upon SNP exposure (Fig. 5C). Pretreatment of EC with GSK-J4 prior to SNP exposure completely abrogated SNP-dependent increment in *VEGFa*, *TBX20*, *MMP2*, *FGF2*, and *Angiopoietin-2* transcript level (Fig. 5C). Although previous reports have indicated changes in the gene expression levels of *KDR*, *TIE2*, and *TEK* genes upon NO induction, in the current setting, however, we did not observe any changes (Fig. 5C). Furthermore, we confirmed the changes in the expression of these genes in EC exposed to GSK126 or EZH2 siRNA, wherein we detected a similar pattern of increase in the expression of the genes mentioned above, similar to the SNP treatment group. However, unlike SNP, where no changes were recorded in *KDR*, *TIE2*, and *TEK* gene transcript, the transcript level of *TEK* in GSK126 treated EC and the transcript level of *TEK*, *KDR* genes in EZH2 siRNA transfected EC were increased (Supplementary Fig. 10B, C). We also confirmed EZH2 siRNA-dependent reduction in EZH2 gene in EC (Supplementary Fig. 10D) and associated reduction in H3K27me3 in both EZH2 siRNA (Supplementary Fig. 10D) and GSK126-treated EC (Supplementary Fig. 10E, F). Next, using the Chromatin-immunoprecipitation (ChIP) experiment, we assessed whether changes in this gene expression upon SNP challenge are driven by loss of H3K27me3 enrichment in the promoter region of the selected genes. A significant reduction in the enrichment of H3K27me3 was detected in gene promoters of *VEGFa*, *TBX20*, *MMP2*, *FGF2*, and *Angiopoeitin-2* in EC exposed to SNP (Fig. 5D).

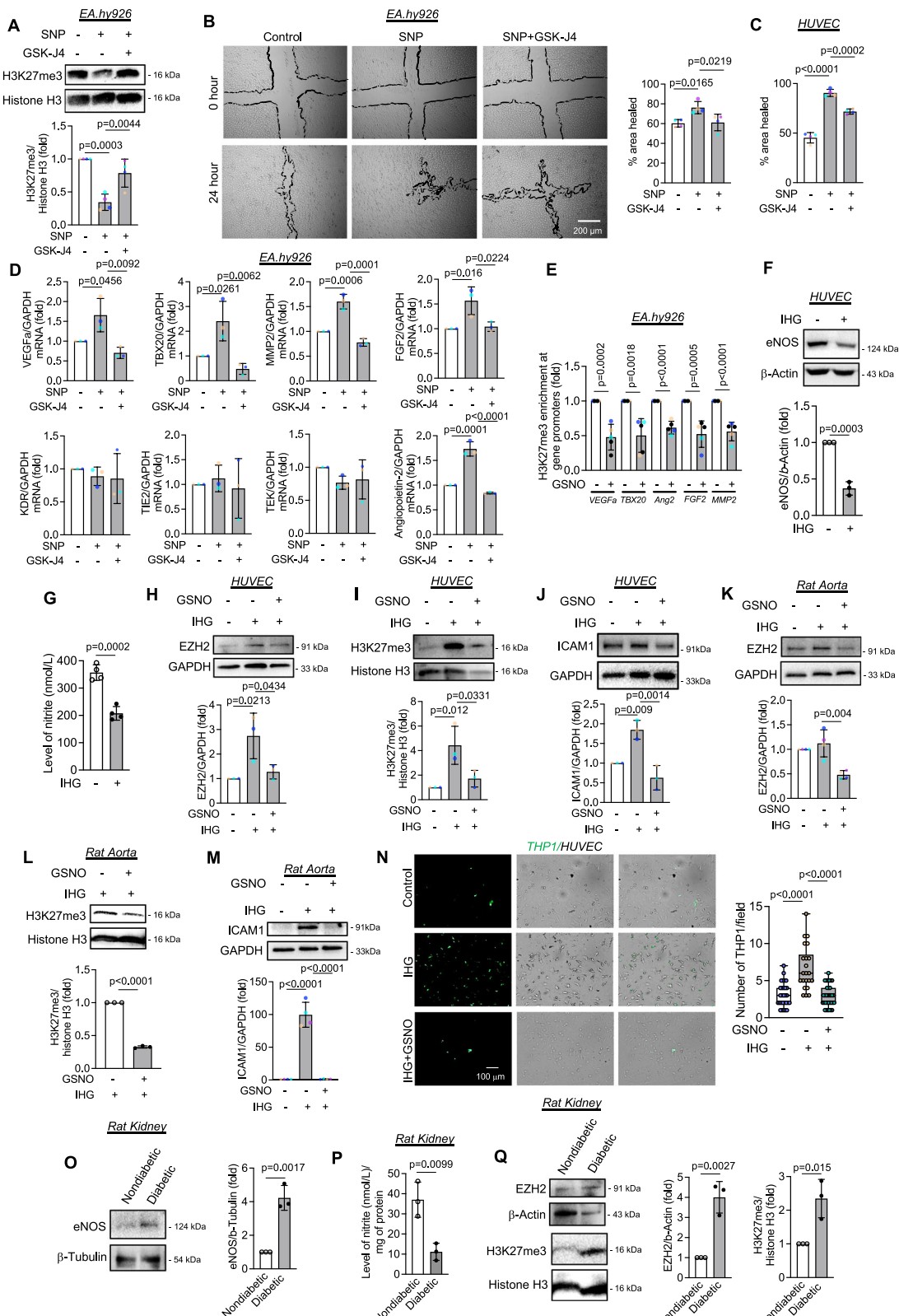

We next explored the efficacy of regulating EZH2 through S-nitrosylation on reversing pathological phenotype in a disease setting in vitro and ex vivo. Our previous findings highlighted the pathological role played by the activation of EZH2·H3K27me3 pathway in driving endothelial inflammation during hyperglycemia stress[28]. In the same study, we reported that suppression of eNOS upon hyperglycemia was one of the key drivers of endothelial inflammation,

however, we did not evaluate whether such downregulation of eNOS could in turn regulate EZH2 stability and function. Indeed, previous findings reported compromised eNOS activation and limited nitric oxide production caused diabetic vascular dysfunction contributing to increased stroke size[29]. We first confirm the reduction in eNOS level (Fig. 5F) and its catalytic product nitric oxide (Fig. 5G) in HUVEC exposed to intermittent high glucose conditions. We, therefore, next

**Fig. 5 | Interplaying with the level of EZH2 and H3K27me3 axis by SNP or GSNO maintain endothelial homeostasis in both physiological and pathological settings. A** Immunoblotting for H3K27me3 in EA.hy926 cells treated with GSK-J4 (5 µM) and SNP (500 mM). (n = 4, biological replicate) **B, C** Scratch wound healing assay using EA.hy926 (**B**) and HUVEC (**C**) treated with GSK-J4 (5 µM) and SNP (500 µM). Bright-field images acquired using microscope adapted with a camera. Healing rate was analyzed using Image J. (n = 4, biological replicate) (**D**) ChIP-qPCR for gene promoters of *VEGFa*, *TBX20*, *Angiopoietin-2*, *FGF2*, and *MMP2* following enrichment with H3K27me3 antibody in EA.hy926 cells treated with SNP (500 µM). (n = 5, biological replicate) (**E**) RT-qPCR analysis to measure the transcript level of *VEGFa*, *TBX20*, *MMP2*, *FGF2*, *KDR*, *TIE2*, *TEK*, and *Angiopoietin-2* in EA.hy926 cells treated with GSK-J4 (5 µM) and SNP (500 µM). (n = 4, biological replicate) (**F, G**) Immunoblot and Griess assay analysis in HUVEC exposed to intermittent high glucose to quantify eNOS (**F**, n = 3, biological replicate) and nitric oxide (**G**, n = 4, biological replicate) respectively. **H–J** Immunoblot analysis of intermittent high glucose (IHG) and/or GSNO (100 µM) exposed HUVEC to detect EZH2 (**H**, n = 3, biological replicate), H3K27me3 (**I**, n = 3, biological replicate), and ICAM1 (**J**, n = 3,

biological replicate). **K–M** Immunoblot analysis of intermittent high glucose (IHG) and/or GSNO (100 µM) exposed rat aorta tissue to detect EZH2 (**K**, n = 4, biological replicate), H3K27me3 (**L**, n = 3, biological replicate), and ICAM1 (**M**, n = 4, biological replicate). **N** Pre-stained THP-1 were incubated on EA.hy926 monolayer which underwent intermittent high glucose (IHG) and/or GSNO (100 µM) treatment. Adherent THP-1 (green dots) were counted per field and plotted. Middle: Box whisker plots show number of monocyte attached with endothelial monolayer, with minimum, first quartile (lower bound), median, third quartile (upper bound), maximum values, whiskers down to the minimum and up to the maximum value. (n = 28, six different fields from three biological replicate) **O–Q** Immunoblot and Griess assay analysis of cortical kidney tissue lysates collected from rats with diabetic kidney disease to measure eNOS (**O**, n = 3, biological replicate), nitric oxide (**P**, n = 3, biological replicate), EZH2 (**Q**, n = 3, biological replicate), and H3K27me3 (**Q**). (n = 3, biological replicate). All data are presented as mean values ± SD. All statistical analyses are either performed by One-way ANOVA with a post-hoc Tukey test for multiple groups or by two-tailed unpaired $t$-test for two groups.

assessed the effect of GSNO in reversing the hyperglycemia-dependent inflammatory switch of EC when cultured EC or rat aortic rings were exposed to intermittent high glucose conditions. Hyperglycemia challenge increased the level of EZH2 and H3K27me3 in both cultured EC (Fig. 5H, I) and rat aorta which was reversed upon GSNO exposure (Fig. 5K, L). Moreover, GSNO treatment also normalized the level of hyperglycemia-induced expression of inflammatory adhesion molecule, Intercellular Adhesion Molecule-1 (ICAM1) in both cultured EC (Fig. 5J) and rat aorta (Fig. 5M). Because immune cells including monocytes adhere to EC expressing adhesion molecules such as ICAM1, we therefore assessed monocyte adhesion in EC exposed to hyperglycemia in the absence and presence of GSNO using the protocol tested previously by our group[30]. Hyperglycemia, which leads to the expression of ICAM1 by EC, promoted monocyte adhesion to the endothelial bed while GSNO treatment abrogated hyperglycemia-induced attachment of monocytes to EC (Fig. 5N). Furthermore, we also assessed the level of EZH2 and its catalytic product in rat kidney tissues with established diabetic kidney disease (Supplementary Fig. 11). We initially measured the level of eNOS in the cortical kidney tissue lysate and surprisingly found an enhanced level of eNOS in the tissues collected from rats with diabetic kidney disease (Fig. 5O). However, diminished level of nitrite was detected in the same kidney tissues from rats with diabetic kidney disease (Fig. 5P). Although surprising, however, many studies reported that eNOS expression was shown to be upregulated in early (1–6 weeks) diabetic kidneys[31–33], however, in contrast studies measuring NO production demonstrated decreased NO level in diabetic kidney, even when eNOS expression is upregulated[34–36]. Because we observed limited nitric oxide/nitrite level in kidney tissues from 6-week diabetic rats, we therefore decided to assess the level of EZH2 and H3K27me3. In so doing, we detected enhanced levels of EZH2 and H3K27me3 in cortical kidney tissues from diabetic rats (Fig. 5Q). Therefore, these data employing in vitro, ex vivo, and in vivo models of hyperglycemia advocate the possibility of S-nitrosylation-mediated regulation of EZH2 as a potential therapeutic strategy to counter diabetic vascular complications.

## S-nitrosylation of EZH2 at cysteine 329 and cysteine 700 is important for its stability and methyltransferase activity respectively

Upon confirming the effect of NO on S-nitrosylation of EZH2 and its associated changes in its stability, catalytic activity, PRC2 assembly and translocation, we next focused on detecting the possible residues of EZH2 that could be S-nitrosylated. To confirm this, we used GPS-SNO (http://sno.biocuckoo.org/) prediction tools as stipulated in the Methodology section. Such analysis predicted three possible sites at cysteine 260 (C260), 329 (C329), and 700 (C700) of EZH2 protein with a score of 3.158, 2.576, and 3.109, respectively, which is beyond the set

cut off value of 2.443 (Supplementary Fig. 12A). Predicted cysteine residues 260 and 329 lies within the domain II of EZH2 which essentially allows SUZ12 association with EZH2, in contrast, predicted cysteine residue at 700 lies within the catalytic SET domain of EZH2 which is essential for its enzymatic activity.

To further decipher the role of each of these cysteine residues, we generated point mutated constructs of these cysteines by using a site-directed mutagenesis kit to convert the codon to code for serine in places of the actual cysteine (TGT/TGC to AGT/AGC) residues (Supplementary Fig. 12B). Using overexpression of these constructs in HEK-293 cells, we performed biotin switch assay to assess the S-nitrosylation of these mutated EZH2. Such analysis revealed partial loss of S-nitrosylation of EZH2 for each of the mutants EZH2 C260S, EZH2 C329S, and EZH2 C700S. Interestingly, a complete loss of S-nitrosylation signal was not detected with single mutation indicating multiple cysteine residues were likely to be S-nitrosylated in EZH2 protein upon SNP exposure (Fig. 6A). We then analyzed the effect of SNP treatment on EZH2 protein and its catalytic product H3K27me3 level in cells overexpressed with WT and mutated form of the EZH2 gene. As observed earlier, SNP caused significant loss of HA-tagged EZH2 or total EZH2 protein and H3K27me3 level in cells overexpressed with HA-tagged EZH2 WT gene (Fig. 6B). A comparable loss of HA-tagged EZH2 or total EZH2 protein and H3K27me3 level was also detected in cells overexpressed with HA-tagged EZH2 C260S mutant gene (Fig. 6B). Interestingly, SNP challenge did not alter the level of HA-tagged EZH2 or total EZH2 protein in cells overexpressed with HA-tagged EZH2 C329S mutant gene, but the H3K27me3 levels in these cells were significantly reduced upon SNP exposure (Fig. 6B). In contrast, SNP significantly diminished the level of HA-tagged EZH2 or total EZH2 protein in cells overexpressing HA-tagged EZH2 C700S mutant. However, it was interesting to note that the H3K27me3 level upon SNP challenge remained unaltered in these cells (Fig. 6B).

We next decided to generate a double mutant form of EZH2 gene where both cysteine residues at location 329 and 700 are mutated to serine, thereby generating a double mutant named EZH2 C329S C700S mutant. Through iodoTMT assay, we confirmed that EZH2 with C329S and C700S mutation exhibited significantly lower S-nitrosylation signal than wild type EZH2 variant in GSNO stimulated EC (Supplementary Fig. 12C). We then overexpressed this double mutant in HEK-293 cells followed by treating the cells with SNP. The double mutated (at cysteine 329 and 700 residues) form of the HA-tagged EZH2 was completely insensitive to SNP exposure both in context to the level of HA-tagged EZH2 or total EZH2 protein as well as its product H3K27me3 (Fig. 6C). In similar settings, cells overexpressing WT EZH2 gene showed significant reduction in both HA-tagged EZH2 or total EZH2 protein and H3K27me3 level (Fig. 6C). We next performed immuno-fluorescence experiment and confocal imaging to ascertain the effect

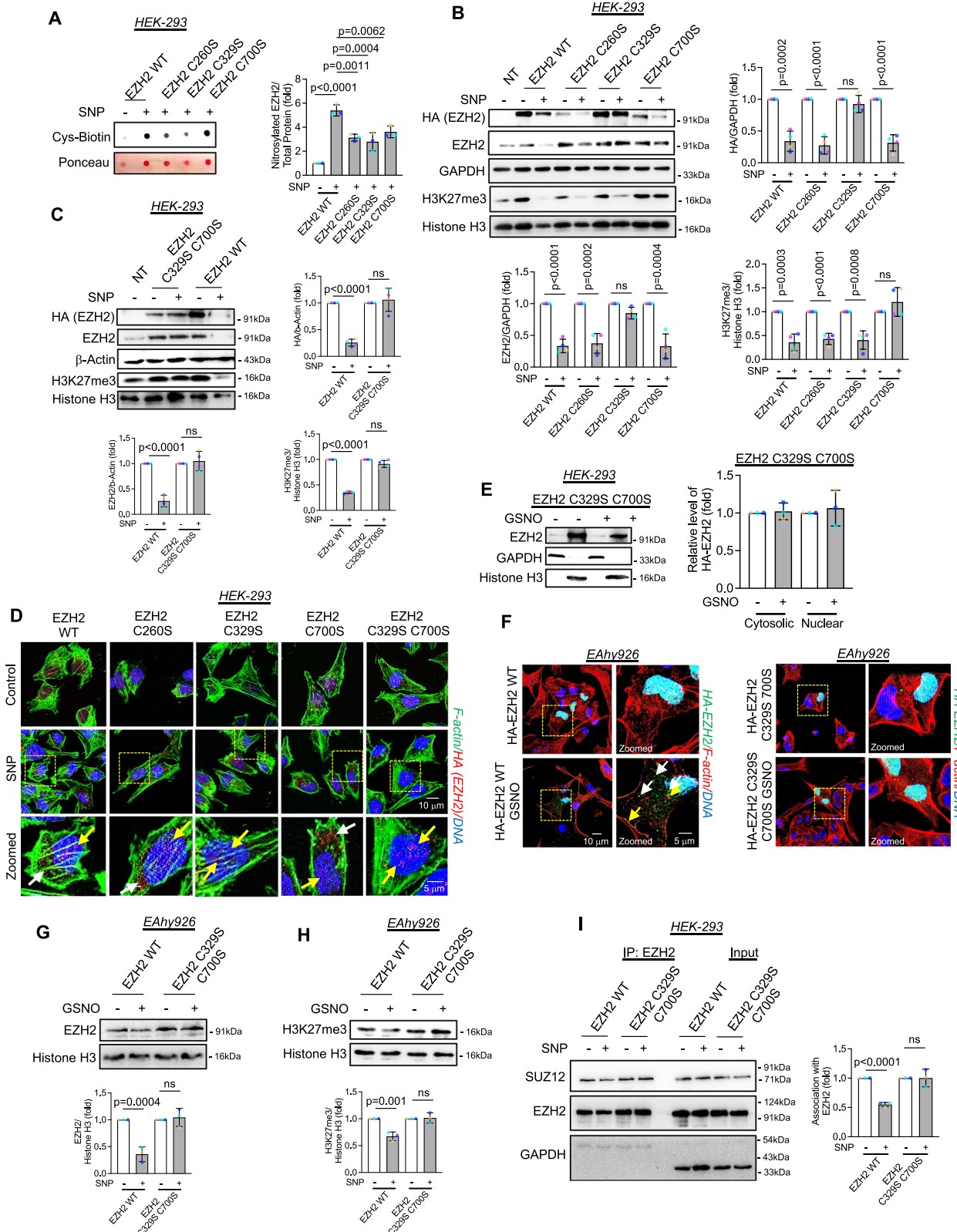

of these individual and double mutations of EZH2 on SNP driven HA-tagged EZH2 localization. In so doing, we observed that SNP exposure caused significant cytosolic delocalization of HA-tagged EZH2 WT, C260S, and C700S mutated forms of HA-tagged EZH2 (Fig. 6D). Interestingly, SNP was unable to induce significant cytosolic localization of HA-tagged EZH2 C329S and double mutated HA-tagged EZH2 C329S C700S protein as confirmed through imaging (Fig. 6D).

Moreover, we also confirmed the inability of GSNO to promote cytosolic localization of HA-tagged EZH2 C329S C700S protein in HEK-293 cells using subcellular fractionation coupled with immunoblot assays (Fig. 6E).

Because we primarily used HEK-293 cells for all the over-expression study, we therefore repeated some of the crucial experiment using overexpression of HA-tagged EZH2 WT and EZH2 C329S

**Fig. 6 | Point mutated EZH2 protein responded differently to nitric oxide in the context of its effects on the stability and catalytic activity of EZH2. A** Dot-blot analysis of HA immunoprecipitated cell lysates of HEK-293 cells overexpressing WT and mutant EZH2 and challenged with SNP (500 μM). (*n* = 3) (**B**) Immunoblotting for HA, EZH2, and H3K27me3 in HEK-293 cells containing wild type or mutated versions of EZH2 in SNP (500 μM) treated cells. (*n* = 4, biological replicate) (**C**) Immunoblotting experiments to show the levels of HA, EZH2, and H3K27me3 in HEK-293 cells transfected with wild-type and the double mutant version of EZH2 and exposed to SNP (500 μM). (*n* = 4, biological replicate). **D** Immunostaining for HA (EZH2) (Red) along with F-actin (green) staining in HEK-293 cells in wild-type and mutated EZH2 expressing cells treated with SNP (500 μM). DAPI shown in blue. White and yellow arrow head indicates the presence of cytosolic and nuclear EZH2 respectively. (Scale bar: 10 μm). (*n* = 3, biological replicate) (**E**) Immunoblotting analysis of EZH2 in nuclear and cytosolic fractions obtained from HEK-293 cells

overexpressing EZH2 C329S C700S and treated with GSNO (100 μM). GAPDH and histone H3 were used to confirm cytosolic and nuclear fractions respectively. (*n* = 3) (**F**) Immunostaining for HA (EZH2) (green) along with F-actin (red) staining in EA.hy926 cells overexpressing EZH2 WT and EZH2 C329S C700S versions, following their exposure to GSNO (100 μM). DAPI shown in blue. White and yellow arrow head indicates EZH2 not bound and bound to F-actin respectively. (Scale bar: 10 μm). (*n* = 3) (**G, H**) Immunoblotting for EZH2 and H3K27me3 in EA.hy926 cells containing EZH2 WT or EZH2 C329S C700S and were exposed to GSNO (100 μM). (*n* = 3) (**I**) HEK-293 cells overexpressing EZH2 WT or EZH2 C329S C700S mutant were treated with SNP (500 μM) followed by co-immunoprecipitation using EZH2 antibody and immunoblotting with SUZ12. (*n* = 3). All data are presented as mean values ± SD. All statistical analyses are either performed by One-way ANOVA with a post-hoc Tukey test for multiple groups or by two-tailed unpaired *t*-test for two groups.

---

C700S mutant in cultured EA.hy926 cells. Through such analysis, we ascertained that GSNO exposure induced the cytosolic localization of HA-tagged EZH2 WT protein; however, failed to cause cytosolic localization of HA-tagged EZH2 C329S C700S mutant (Fig. 6F). We then assessed the effect of GSNO on the relative level of EZH2 and H3K27me3 in EC transfected with HA-tagged EZH2 WT and EZH2 C329S C700S mutant. In parallel to the observation in HEK-293 cells, GSNO reduced the level of EZH2 (Fig. 6G) as well as H3K27me3 (Fig. 6H) in EC transfected with HA-tagged EZH2 WT, whereas such treatment did not alter the level of EZH2 C329S C700S mutant (Fig. 6G) and H3K27me3 (Fig. 6H). We next evaluated the consequence of SUZ12 association with the EZH2 C329S C700S mutant when exposed to SNP. Interestingly, SNP exposure failed to cause SUZ12 dissociation from EZH2 having C329S and C700S mutations (Fig. 6I). This data altogether suggests a critical role played by the cysteine residues in EZH2 protein at C329 and C700 which controls its stability, localization, and catalytic activity through S-nitrosylation.

### S-nitrosylation of EZH2 protein at C329 and C700 causes conformational changes in EZH2-SUZ12 complex leading to a weak association of SUZ12 with the SAL domain of EZH2

Upon experimental validation of EZH2's S-nitrosylation and its effect on EZH2-SUZ12 interaction, catalytic activity, localization and EZH2 protein stability, we next employed molecular dynamics analysis to understand the effect of S-nitrosylation on EZH2 interaction with SUZ12 protein. Through AlphaFold modeling of EZH2-SUZ12, we showed the initial conformations of the EZH2-SUZ12 complex in both the EZH2 WT and EZH2 S-nitrosylated at C324 (originally C329) and C695 (originally C700) residues (Fig. 7A) while also clearly visualizing the S-nitrosylation at C324 (Fig. 7B) and C695 (Fig. 7C). The RMSD calculations elucidate the extent to which the conformations of the EZH2 WT and EZH2 S-nitrosylated complexes deviated from their initial states during the course of the molecular dynamics (MD) simulations. In comparison to the WT complex, a sharp rise of RMSD values is observed for S-nitrosylated complex after 100 ns indicating possible changes in the overall conformations during the simulations. A system showed larger conformational changes for EZH2 S-nitrosylated than for EZH2 WT complex (Fig. 7D).

Gamblin et al. (2016) reported that residues 112 to 121 within the SAL pack engage with the SET-1 region, serving to stabilize its conformation within the active complex[37]. Importantly for regulation in the human PRC2 complex, the conserved acidic residues 584–588 of SUZ12 reciprocally pack against residues 112–121 of SAL. Consequently, we conducted measurements of the backbone RMSD for both residue ranges, 112–121 of SAL and 584–588 of SUZ12, revealing a significant fluctuation in the RMSD within the SAL region of EZH2 S-nitrosylated with little to negligible RMSD fluctuation for SUZ12 residues ranging from 584 to 588 (Fig. 7E, F).

Another crucial parameter of MD is root mean square fluctuation (RMSF) to determine the flexibility of protein residues. The RMSF value

of both EZH2 WT and EZH2 S-nitrosylated remained relatively stable with a nearly similar trend (Supplementary Fig. 13A). Primarily, the residues near the SAL and long loop region (residue 112–121 and 345–421) display significant fluctuations. Apart from these loops and SAL region of EZH2, RMSF analysis shows stable interactions in both the simulated complexes. Subsequently, we visualized the interface hydrogen bond interactions between SAL residues 112–121 and SUZ12 residues 584–588 throughout 1 μs simulation. Interestingly, in the EZH2 WT, we observed the oxygen atom of LEU166 in EZH2 interacting through the hydrogen bond with LYS587 of SUZ12 and the HE22 atom of GLN117 in EZH2 interacting with the OD2 atom of ASP585 in SUZ12 (Fig. 7H, I) These interactions persisted throughout the simulation. In the case of S-nitrosylated EZH2, these two hydrogen bond interactions were initially established and maintained up to 300 ns. However, beyond this time point, the interactions were no longer observed.

Hydrogen bonds (H-bonds) are important non-covalent interactions that help stabilize protein-ligand and protein-protein complexes. To better understand the dynamics of these interactions at the interface, we analyzed the H-bonds formed between specific residues 112–121 of SAL and 584–588 of SUZ12. This examination allowed us to gain insights into the overall stability and binding strength of the complexes (Supplementary Fig. 13B). During the simulations, we closely observed the formation of H-bonds within a distance of 3.5 Å. Upon conducting the analysis, it was discovered that the specific residues 112–121 of SAL were found to interact with 584–588 of SUZ12 in the EZH2 WT complex. This interaction resulted in an average of 2 hydrogen bonds. Conversely, the S-nitrosylated counterpart exhibited a significantly lower average of 0.35 hydrogen bonds suggesting EZH2 WT complex have stronger interactions than EZH2 S-nitrosylated variant. This observation also aligns with the energy profile (Supplementary Fig. 13C). We further measured the distance between the oxygen atom of LEU166 in EZH2 and the hydrogen atom of LYS587 in SUZ12, as well as the distance between GLN117 in EZH2 and ASP585 in SUZ12. A sharp increase in the distance between GLN117 and the interacting residue ASP585 after 150 ns in the EZH2 S-nitrosylation complex simulation (Fig. 7G) was observed, that ultimately led to the disruption of the interaction. Such distance between the oxygen atom of LEU166 in EZH2 and the hydrogen atom of LYS587 in SUZ12, as well as the distance between the HE22 atom of GLN117 in EZH2 and its interacting atom OD2 of ASP585 in SUZ12 at 100 ns (after stabilization of the complex upon initial simulation) and at 800 ns, was also presented through the structure analysis (Fig. 7H–K).

To determine the more accurate free energy binding of all complexes, we utilized the MM/PBSA-based method. The binding free energy in this context refers to the total of all non-bonded interaction energies between the EZH2 and SUZ12 during the MD simulation, including Van der Walls, electrostatic, polar solvation, and SASA energies. A more negative binding free energy indicates a stronger affinity between the EZH2 and the SUZ12. We calculated the binding free energies using 1 μs MD trajectory, revealing that EZH2 WT and EZH2

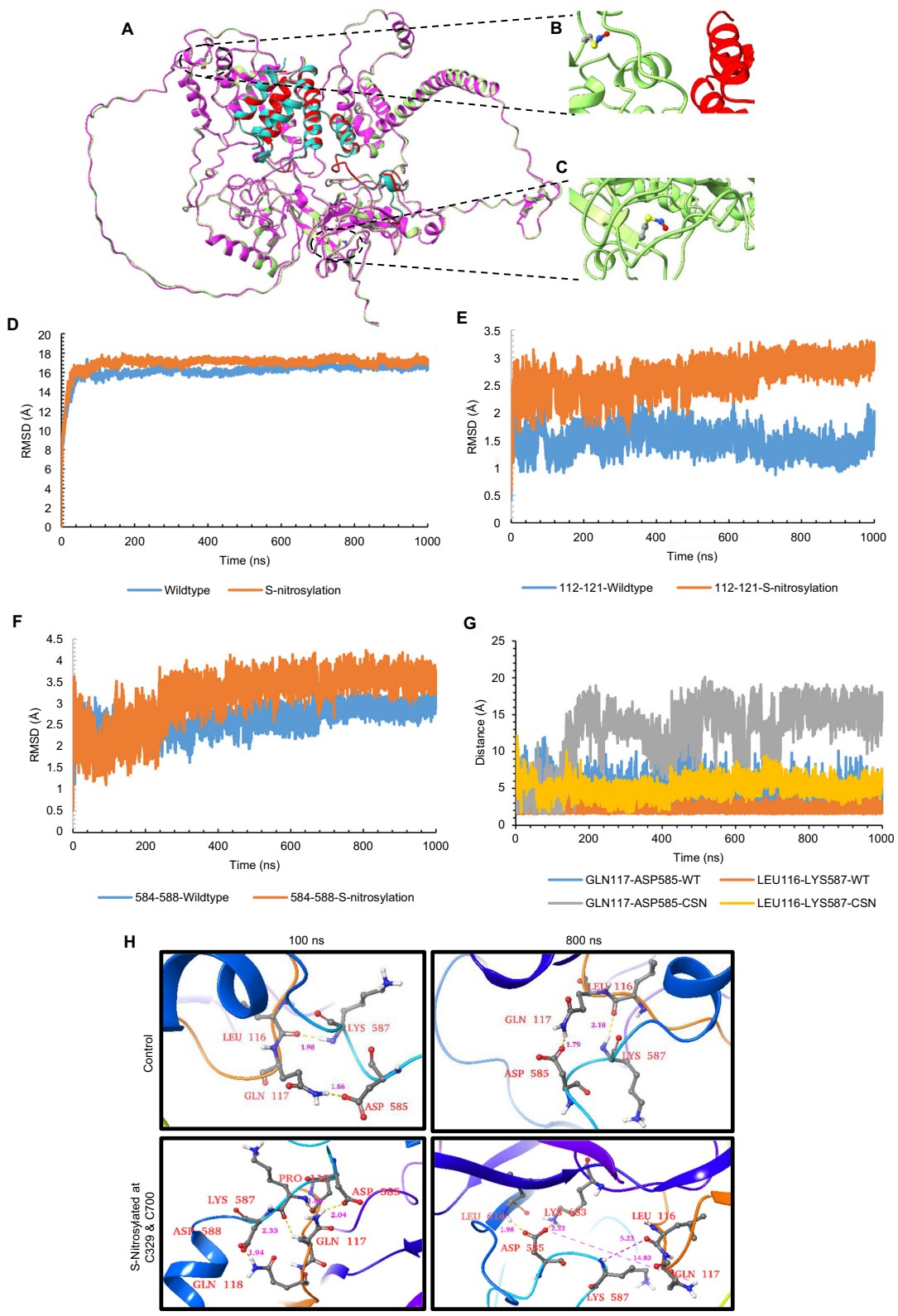

S-nitrosylation have a binding free energy of −4649.807 ± 454.449 kJ/mol and −4582.981 ± 453.056 kJ/mol, Subsequently, estimation of their contributions to overall ΔΔG_bind = 66 kJ/mol indicates that the contributions vary from one system to the next. On average, the WT state tends to have a marginally higher contribution (Supplementary Fig. 13C and Supplementary Table 1). Furthermore, the comparison of the binding energy decomposition between selected residue of SAL and SUZ12 in EZH2 WT and EZH2 S-nitrosylated complexes located at the binding interface reveals that the EZH2 WT predominantly contributes to binding in a locally stabilizing manner. Conversely, EZH2 S-nitrosylation displays less contribution to binding energy and these effects might be the consequence of introduced nitrosylation at cysteine 324 and 695 (Supplementary Fig. 13D, E). Overall, these modeling results suggest that S-nitrosylation of the cysteine residues might induce a conformational change in the SAL region of EZH2 which is crucial for the regulation of the PRC2 complex. This conformational

**Fig. 7 | Performing Molecular Dynamic (MD) simulation studies through GROMACS 5.1.5 version using WT and S-nitrosylated EZH2 at C324 and C695.** A–C Structural illustration of the initial structural alignment of the wildtype and S-nitrosylated EZH2 protein of the EZH2-SUZ12 complex (A). The magenta and green color represents the EZH2 chain of both WT and S-nitrosylated version, while the cyan and red color indicates the SUZ12 chain. It also provides a close-up view of S-nitrosylation at position C324 of EZH2 in the S-nitrosylated protein (B) and zooms in on the S-nitrosylation at position C695 of EZH2 in the S-nitrosylated protein (C). D, E Backbone RMSD of whole EZH2 WT and EZH2 S-nitrosylated (D). Backbone

RMSD of EZH2 WT and EZH2 S-nitrosylated specifically within the residue ranges between 112 and 121 of SAL (E). Backbone RMSD of residues in SUZ12 ranging between 584 and 588 (F). G, H Structural illustration of EZH2-SUZ12 complex near the SAL (112–121) domain of EZH2 in both EZH2 WT and EZH2 S-nitrosylated forms at 100 ns and 800 ns of simulation. Structural data reflects the distance between unique residues of EZH2 (LEU116 and GLN117) and SUZ12 (ASP585 and LYS587) to reflect potential for hydrogen bonds which were lost at 800 ns only in EZH2-SUZ12 complex having S-nitrosylated form of EZH2 at C324 and C695.

change weakens the SUZ12-SAL interaction, ultimately resulting in the destabilization of the SUZ12-EZH2 complex.

## Discussion

A key regulatory pathway of NO-dependent cellular function is through S-nitrosylation of protein which is a reversible post-translational modification. Extensive studies have shown that S-nitrosylation regulates diverse physiological and pathological processes such as angiogenesis, adaptive immunity, diabetes, heart failure, stroke etc. by modulating stability, activity, subcellular localization, conformation change, or protein-protein interaction of target proteins[38]. Although NO signaling alters gene expression changes in vascular endothelium[39], the role of NO-dependent S-nitrosylation of proteins in defining the epigenetic landscape of EC is not clearly understood. Herein, we report that S-nitrosylation of EZH2 protein caused early dissociation of SUZ12 from the EZH2 bound PRC2 complex leading to inhibition of its catalytic activity followed by cytosolic translocation of S-nitrosylated EZH2 to undergo ubiquitination and degradation primarily through autophagosome-lysosome pathway (Fig. 8). Moreover, mass spectrometric analysis revealed S-nitrosylation of EZH2 significantly alters its interactome preferentially allowing association with proteins of the endosome/lysosome/proteasome pathways. We also demonstrated that induction of endogenous NO producing machinery in EC also diminishes EZH2 and H3K27me3 level. Through site-directed mutagenesis studies, we specifically identified that S-nitrosylation of C329 and C700 residues of the EZH2 protein were responsible for its cytosolic translocation/degradation and catalytic deactivation, respectively. Molecular dynamic analysis suggested structural changes in EZH2 protein upon S-nitrosylation at C329 and C700 causing instability of EZH2-SUZ12 complex. Further, we demonstrated that S-nitrosylation of EZH2 was responsible for NO-dependent epigenetic regulation of endothelial gene expression and endothelial migration. Moreover, using an established hyperglycemia model system wherein the EZH2-H3K27me3 pathway is activated, we also showed that targeting this pathway using GSNO reversed endothelial inflammation and monocyte adhesion in diabetic conditions in vitro and ex vivo, which is likely by the S-nitrosylation of EZH2 protein (Fig. 8).

Post-translational modification of EZH2 protein through phosphorylation at several sites were reported earlier[39–42]. Such modifications of EZH2 differentially regulated its localization, its catalytic functions and its association with other proteins of the PRC2 complex. For instance, AMPK phosphorylated EZH2 at T311 to disrupt the interaction between EZH2 and SUZ12 thereby attenuating PRC2-dependent methylation of histone H3 at Lysine 27 in ovarian and breast cancer. Moreover, such phosphorylation of EZH2 by AMPK caused upregulation of PRC2 target genes, many of which are known tumor suppressors thereby suppressing the growth of tumor cells[42]. Similarly, we observed that S-nitrosylation of EZH2 protein at C329 and C700 caused its dissociation from SUZ12 thereby abrogating the catalytic potential of PRC2 complex. Moreover, such catalytic deactivation of PRC2 via EZH2 S-nitrosylation caused loss in deposition of H3K27me3 on gene promoters thereby inducing the expression of several genes associated with endothelial function. Furthermore, such catalytic deactivation through S-nitrosylation of EZH2 also reversed

hyperglycemia-induced endothelial inflammation and monocyte adhesion. During breast cancer metastases, EZH2 was shown to be phosphorylated at T367 via p38 MAP kinase which induces cytosolic localization of EZH2 and further promotes EZH2 binding to many cytoskeletal regulators including vinculin which play critical role in cell migration and invasion. Interestingly, a phospho-deficient T367A-EZH2 mutant when forced expressed inhibited the cytoplasmic localization of EZH2 further interplaying with its binding to cytoskeletal regulators, thereby compromising EZH2-mediated adhesion, migration, invasion, and development of spontaneous metastasis[39]. In the present study, we observed that S-nitrosylation of EZH2 specifically at C329 caused its cytosolic localization, a transient binding to actin cytoskeleton, ubiquitination of EZH2 protein followed by degradation primarily through autophagosome-lysosome pathway. In addition, we also reported enhanced endothelial migration upon exposure to S-nitrosylating agent which may partly be driven by such cytosolic translocation of S-nitrosylated EZH2 and its transient binding to the F-actin cytoskeleton. Because NO is well known to promote endothelial migration[43,44], our current observations highlight the importance of EZH2-regulated cytosolic and nuclear pathways dictating NO-dependent endothelial migration. A very recent study demonstrated DCAF1-mediated phosphorylation of EZH2 at T367 to augment its nuclear stabilization and enzymatic activity in colon cancer cells. Such DCAF1-mediated EZH2 phosphorylation followed by nuclear localization led to elevated levels of H3K27me3 and altered expression of growth regulatory genes in cancer cells[40]. However, in our study we report that S-nitrosylation of C329 residue of EZH2 caused its cytosolic localization followed by its degradation through autophagosome-lysosomal pathways. Moreover, S-nitrosylation at C700 led to inactivation of the catalytic activity of EZH2 likely due to early dissociation of SUZ12 protein from PRC2 complex. Although, phosphorylation-dependent regulation of EZH2 protein and the PRC2 complex are well documented, other PTMs especially S-nitrosylation dependent regulation of EZH2 and PRC2 complex has not been reported before. Herein, we showed that activation of endogenous NO producing machinery of EC or external supplementation of NO using donors or S-nitrosylating agent such as GSNO regulated the binding of EZH2 with SUZ12, its localization, and protein stability. Such S-nitrosylation dependent regulation of EZH2 contributed towards NO-driven gene expression changes and endothelial migration as well as reversed endothelial dysfunction in diabetic settings. These effects were primarily achieved by S-nitrosylation of EZH2 protein at C329 and C700 residues.

In EC endogenous eNOS-dependent release of NO drives cell survival, migration and growth typically acting through canonical and non-canonical pathways[45,46]. Canonically, NO induces soluble guanyl cyclase (sGC) and in downstream Protein Kinase G to mediate actin remodeling required for cell migration[47,48]. Moreover, NO signaling also promotes endothelial gene expression changes that are associated with endothelial survival and growth[49,50]. Such gene expression changes by NO in EC were thought to be primarily controlled by gene transcription and mRNA translation via iron-responsive elements[51]. Previous findings by Mitić et al., 2015 reported EZH2-dependent regulation of eNOS by enhancing the level of H3K27me3 abundance onto regulatory regions of eNOS gene promoters. Moreover, gene silencing

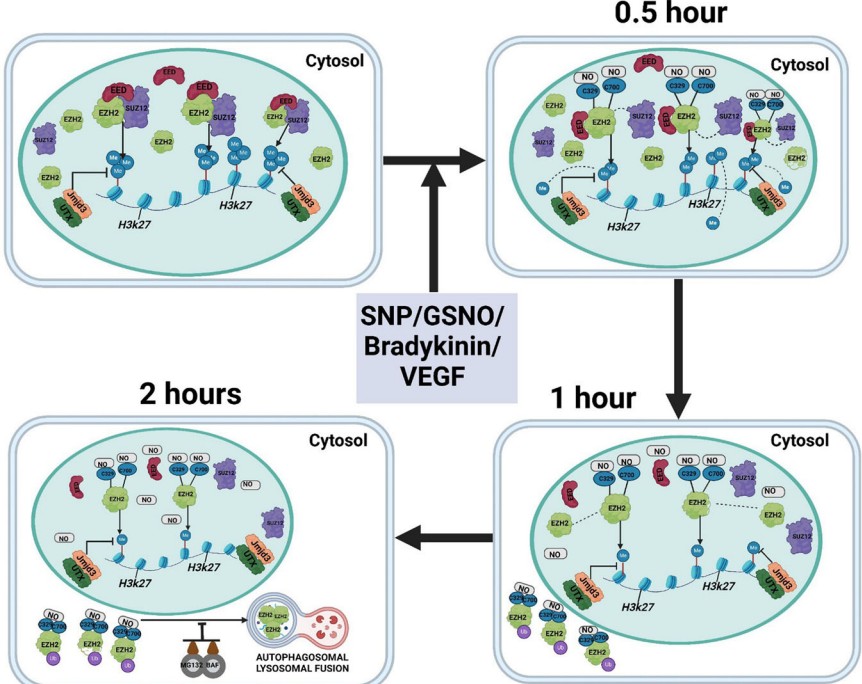

**Fig. 8 | Schematic depicting the effect of EZH2 S-nitrosylation on its stability, translocation and catalytic activity of Polycomb Repressive Complex-2.** PRC2 consists mainly of EZH2, SUZ12, and EED, residing primarily in the nucleus. The major subunit of PRC2 i.e., histone methyltransferase EZH2 localizes primarily in the nucleus and is responsible for deposition and maintenance of the levels of H3K27me3. These cells also have a limited cell migration capacity. On exposure of these cells to SNP/GSNO/bradykinin/VEGF, S-Nitrosylation of specific cysteine residues on EZH2 occur leading to a decline in their stability which is marked by reduction in EZH2 protein levels (by 2 h) along with early dissociation of SUZ12 (by 0.5 h) from EZH2-bound PRC2 complex. Upon disassembly, S-Nitrosylated EZH2 translocate from the nucleus to the cytosol (by 0.5 to 1 h) where it undergoes ubiquitination followed by its degradation primarily through the autophagosome-

lysosome pathway. Additionally, there is a significant effect on the catalytic activity of EZH2, which is shown by a reduction in the deposition of H3K27me3 marks on histone H3 (by 1 h). Such reduction in H3K27me3 level upon catalytic inactivation of EZH2 through S-Nitrosylation was contributed by demethylase UTX and JMJD3, both of which remained unchanged upon SNP/GSNO challenge to EC. Such reduction in H3K27me3 through EZH2 S-nitrosylation was responsible for NO dependent induction of endothelial gene expression and migration. Moreover, abrogating EZH2-H3K27me3 axis using GSNO in hyperglycemia exposed EC, reversed endothelial inflammation and monocyte adhesion. The schematic diagram was prepared using BioRender (Agreement number: HD28OFT6PP). Created in BioRender. Chowdhury, R. (2025) https://BioRender.com/t71o935.

and pharmacological inhibition of EZH2 increased eNOS mRNA and protein levels and elevated the functional capacities of EC (migration, angiogenesis) under both normoxia and hypoxia conditions. Although, EZH2-dependent regulation of eNOS gene was reported earlier, whether, eNOS and its catalytic product NO control EZH2 and PRC2 complex has not been explored[52]. Herein, we reported that eNOS-dependent NO release instigates non-canonical pathway through S-nitrosylation of EZH2 protein and its product H3K27me3 thereby controlling the epigenetic regulation of gene expression. This was supported by our data wherein preserving the level of the catalytic product of EZH2 i.e., H3K27me3 through inhibition of the demethylases JMJD3-UTX reversed NO dependent endothelial gene expression changes and cell migration. Moreover, we also observed that the regulation of EZH2 through S-nitrosylation is independent of the cell type, as reported in our study wherein we observed similar trends in HEK-293 cells exposed to SNP to that of EC. In parallel to well reported function of NO[53,54] as well as EZH2[24,55] in actin cytoskeleton remodeling, we observed that NO exposure caused transient localization of EZH2 on the actin cytoskeleton most likely prior to its degradation through autophagosome-lysosomal pathway. In addition, preserving the level of H3K27me3 via inhibition of demethylases reversed the effect of SNP on cell migration. However, it is still unclear whether the effect of NO on endothelial migration is mainly driven by its regulation of EZH2-actin cytoskeleton remodeling axis or via regulation of EZH2-H3K27me3-gene expression changes. Nonetheless, NO dependent remodeling of actin cytoskeleton may not alone be driven by canonical

sGC-cGMP-PKG-VASP pathway, in addition, regulation of EZH2 via NO may contribute towards its actin remodeling function.

S-nitrosylation via endogenous NO machinery including endogenous GSNO has a protective role in the cardiovascular system. For example, S-nitrosylation is an NO-based signaling mechanism regulating endothelial protein trafficking and suppression of nuclear factor κB (NF-κB)-dependent expression of proinflammatory cytokines and adhesion molecules[56]. Exocytosis of granules carrying Weibel–Palade bodies were altered through S-nitrosylation of N-ethylmaleimide sensitive factor which in turn causes exocytosis of P-selectin, an adhesion molecule thereby inhibiting rolling of leukocyte and inflammation of vascular wall[57]. Platelets also exhibited similar mechanism which further lead to limited activation, adhesion, aggregation, and thrombosis[58]. Moreover, inhibitory S-nitrosylation of both NF-κB and its activating enzyme complex, inhibitory-κB kinase were also reported thereby regulating its downstream signaling cascade[59]. Enzymes such as GSNO reductase (GSNOR) selectively metabolises endogenous GSNO thereby depleting the pool GSNO causing a shift in the equilibrium, thereby limiting the levels of S-nitrosylated proteins. Mouse lacking GSNOR manifested increased levels of S-nitrosylated proteins[60]. Moreover, cardiac infract size and remodeling were significantly diminished in mice lacking GSNOR while significantly improving survival and cardiac function after myocardial infarction[61]. Thus, NO dependent anti-inflammatory actions through S-nitrosylation of proteins are important in several kind of cardiovascular diseases such as autoimmune disorders, and atherosclerosis. In

the current study, we confirmed diminished NO level in kidney tissues of rats with diabetic kidney disease with a concurrent increase in EZH2 and H3K27me3 level. Moreover, we report that S-nitrosylation of EZH2 has a vascular protective role at least in a diabetes setting. This is supported by our observation in a well-established model of hyperglycemia treated EC where we had previously reported activation of EZH2-H3K27me3 axis causing endothelial inflammation[28]. Furthermore, administration of GSNO in the same model attenuated hyperglycemia-induced EZH2-H3K27me3 activation, endothelial inflammation and further diminished monocyte adhesion both in vitro and ex vivo. However, because S-nitrosylation of EZH2 has not been reported before, whether such cardiovascular protective effects of NO or GSNO are partly driven by the regulation of EZH2 through S-nitrosylation need to be comprehensively established through future pre-clinical and clinical studies.

In summary, our study reports that the NO signaling pathway converges to epigenetically regulate gene expression by controlling the stability and catalytic activity of PRC2 complex, more specifically the EZH2 component of the complex. This is primarily driven by NO through S-nitrosylation of C329 and C700 residues of EZH2 which facilitates early dissociation of SUZ12 from EZH2-PRC2 complex followed by cytosolic localization of EZH2 and its degradation through autophagosome-lysosome pathways. We also showed that such an effect of NO on EZH2 is cell-type independent. Moreover, we also reported that NO-driven gene expression changes in EC primarily depend on NO-dependent regulation of EZH2 and its catalytic product H3K27me3. Furthermore, in a disease setting, we also reported the reversal of hyperglycemia-driven endothelial inflammation upon S-nitrosylation of EZH2. Our molecular dynamics simulation provided evidence that S-nitrosylation of EZH2 at C329 and C700, causes changes in the structure of EZH2 SAL domain leading to its dissociation with SUZ12. Furthermore, site-directed mutagenesis of EZH2 at C329 and C700 reveals that S-nitrosylation of C329 is responsible for EZH2 cytosolic translocation and degradation while S-nitrosylation of C700 is primarily responsible for catalytic inactivation of EZH2. Although we reported loss of eNOS gene expression or catalytic inhibition through siRNA or L-NAME treatment respectively reversed bradykinin or VEGF dependent downregulation of EZH2 and H3K27me3 in vitro and ex vivo, we could not prove this in an in vivo settings especially with eNOS knockout mice. Future studies exploring more NO-dependent regulation of epigenetic pathways through S-nitrosylation of nuclear proteins in such in vivo model could open avenues for understanding their role in patho-physiology. Altogether, our findings shed light into a mechanism of NO-mediated regulation of EZH2 and PRC2 complex to regulate gene expression which we found to be associated with endothelial function.

## Methods

### Study approvals
All animal studies were carried out according to the Committee for Control and Supervision of Experiments on Animals guidelines for the use of laboratory animals. All the experimental procedure involving rodent studies were reviewed and approved by the Institutional Animal Ethics Committee (IAEC) of BITS Pilani, Pilani Campus and was in accordance with the protocol approved by IAEC, BITS-Pilani (Protocol Approval No: IAEC/RES/31/13/REV-1/33/19, and Protocol Approval No: IAEC/RES/23/05/Rev-1/28/27).

### Cell culture
Experiments were done primarily in EA. hy926 (immortalized human umbilical vein EC), purchased from ATCC, Manassas, USA (#CRL-2922). The cells were cultured in Dulbecco's modified Eagle's medium (DMEM) (#AL006A; HiMedia Laboratories) supplemented with 10% fetal bovine serum (#RM1112, FBS; HiMedia Laboratories) and 1% penicillin/streptomycin (#10378, PS; Gibco). The cells were passaged

every 2 and 3 days. Human umbilical vein EC (HUVEC) (#CL002-2XT25, HiMedia Laboratories) were cultured and passaged using HiEndoXL™ EC Expansion medium (#AL517, HiMedia Laboratories) along with 4% growth factor, 5% fetal bovine serum and 1% penicillin/streptomycin antibiotic. Human Embryonic Kidney HEK-293 cell line was a kind gift from Prof. Uma Dubey (Department of Biological Sciences, Birla Institute of Technology and Science Pilani, Pilani Campus, India) and were maintained in Minimum Essential Media Eagle (MEM) (#AT154; HiMedia Laboratories). Transformed human monocytes cell line THP-1 were cultured in RPMI-1640 media (#AL028A, Hi Media) which was supplemented with FBS (10%) and mixture of penicillin and streptomycin (1%). All the cells were maintained in a humidified $CO_2$ incubator at 37 °C.

### Plasmid, siRNA and Transfection
Transfection experiments were performed with human WT pCMVHA hEZH2 plasmid (#24230, Addgene), a kind gift from Dr. Kristian Helin (Biotech Research and Innovation Centre, University of Copenhagen, Copenhagen, Denmark). HEK-293 cells with 80% of confluency were transfected using 250 ng/ml of pCMVHA hEZH2 and Lipofectamine 2000 (#11668030, Invitrogen, Thermo Fisher Scientific) for 4 h. After 48 h of incubation, the cells were then used for immunoblotting and immunofluorescence experiments. EZH2 small-interfering RNA (SignalSilence® Ezh2 siRNA I #6509) and control siRNA (SignalSilence® Control siRNA#6568) were obtained from Cell Signaling Technology. eNOS siRNA was designed (Sense: 5′ GAA GAG GAA GGA GUC CAG UAA CAC A 3′ and Antisense: 5′ UGU GUU ACU GGA CUC CUU CCU CUU C 3′) obtained from Eurogentec. EZH2 siRNA or eNOS siRNA was used at a concentration of 40 nM. OptiMEM containing lipofectamine 2000 (#11668030, Thermo Fisher Scientific, Waltham, MA, USA) and siRNA were subjected to EA.hy926/HUVEC for 4 h followed by incubating the transfected cells in complete DMEM/HiEndoXL™ EC Expansion medium for another 24 h. Cells were either analyzed for wound healing assay as detailed later or harvested upon completion of incubation time either with Trizol reagent (#15596, TRIzol™ Reagent; Life Technologies, Thermo Fisher Scientific) for RNA isolation or with RIPA buffer (#9806; Cell Signaling Technology) for western blot analysis.

### Inducers and inhibitors
Sodium Nitroprusside dihydrate (SNP, 500 µM)[62] (#71778, Sigma Aldrich), and S-nitrosoglutathione (GSNO, 100 µM) (#HY-D0845, Med Chem Express) was used as an exogenous donor of NO in our experiments. Natural induction of NO was activated using 5 or 10 µM of bradykinin (#B3259; Sigma Aldrich) or using human VEGF165 (#100−20, Peprotech) at a concentration of 10 ng/mL[63]. Inhibition of NO in cell system was achieved by treating the cells with $N_\omega$-Nitro-L-arginine methyl ester hydrochloride (L-NAME, (1 mM)[64] (#N5751; Sigma Aldrich) an analog of arginine. Bafilomycin A1 (100 nM)[65] (#11038; Cayman Chemicals) was used as an autophagy Inhibitor. MG132 (1 µM)[66] (#M7449), a proteasomal inhibitor was from Sigma Aldrich. As reported by our group earlier, we have used 5 µM GSK-J4[67] (#SML-0701, Sigma Aldrich), a H3K27me3 demethylase inhibitor and a competitive SAM inhibitor of EZH2, GSK126 (#HY-13470, Med Chem Express) was used at a final concentration of 10 µM[28]. DMSO was used as vehicle control. EA.hy926 cells were treated with GSK126 in the specified concentration for 24 h to undertake the respective analysis.

### High glucose treatment condition
Based on our previous findings, we implemented the high glucose treatment condition to impart robust endothelial dysfunction including endothelial inflammation[28,68]. In brief, HUVEC were trypsinized and seeded in six-well plates at a confluence of $1 \times 10^5$. After achieving a 70% confluence, they were then subjected to hyperglycemia condition with high glucose level of 25 mM. An intermittent high glucose (referred to as IHG) treatment condition was established through 12 h of high

glucose (25 mM) and 12 h of normal glucose (5.5 mM) cycle for 3 cycles, totaling 72 h of treatment time. In control treatment condition, cells were constantly exposed to normal glucose (5.5 mM).

## Griess assay

Cells grown to 85% confluence in the 96 well plate, were provided treatment conditions, following which the cells were washed with PBS. This was followed by addition of 50 µl of sulfanilamide solution (prepared by diluting 1% sulfanilamide in 5% orthophosphoric acid) to all the wells or collected conditioned media and incubated for 10 min in dark at room temperature. Next, we added 50 µL of NED solution (prepared by diluting 0.1% N-1-napthylethylenediamine dihydrochloride in water) to each well followed by incubating again in dark conditions for 10–15 mins. For analyzing the level of nitrite in tissues, we have taken tissue lysates prepared using RIPA lysis buffer followed by sonication. Fixed volume of tissue lysates of 50 µL were added to each well of 96 well plate followed by addition of the respective reagents and incubation as mentioned above. The absorbance was measured using a spectrophotometer at 540 nm. For nitrite analysis in tissues, we have normalized the nitrite concentration to that of the protein concentration of the tissue lysate to obtain nitrite level per µg of protein.

## Monocyte adhesion assay

To perform the monocyte adhesion experiment, THP-1 suspension culture cells were stained using 3,3'-Dioctadecyloxacarbocyanine perchlorate green fluorescent lipophilic dye (#D275, Thermo Fisher Scientific) for 10 min. EC were exposed to intermittent high glucose with or without GSNO followed by incubation of fluorescence labeled THP-1 cells ($2 \times 10^5$ cells/well) for 30 min. THP-1 cells which did not adhere to the EC surface were washed away with 1× PBS followed by fixing the adhered THP-1 cells using 4% PFA. Images, both fluorescence and DIC were acquired by Zeiss ApoTome 2.0 microscope (Carl Zeiss). We next counted the number of adhered THP-1 monocytes (appeared as green dots) on endothelial monolayer and analyzed using ImageJ software.

## Subcellular fractionation

Following the treatment with SNP for the given time point conditions, EA. hy926 cells were washed with phosphate-buffered saline (PBS) followed by scraping. The cell pellet was resuspended in ice-cold PBS containing 0.1% Nonidet P-40 (NP-40) and 0.1% protease inhibitor (#P8340; Sigma Aldrich) and then centrifuged at 10,000 rpm at 4 °C for 10 min. Upon centrifugation, the supernatant was collected as the cellular and cytoplasmic fraction, and the nuclear fraction was further pelleted down. The cellular and nuclear lysate obtained was sonicated for 10 s (×2) followed by Immunoblotting experiments.

## Immunoblotting

Cells grown to 80% confluency were provided treatment conditions followed by washing with sterile 1× PBS. They were then incubated in RIPA buffer (#9806; Cell Signaling Technology) containing protease inhibitor for lysis after washing with sterile 1× PBS. After scraping, the cells were subjected to repeated cycles of sonication for 10 s (×3). This was followed by a centrifugation step at $10,000 \times g$ for 10 min to pellet down the cell debris and collect the supernatant. Bradford assay was performed to quantify total protein concentration. After this, the protein samples were mixed with 5× laemmli buffer and heated at 100 °C for 10 min. They were then processed for SDS polyacrylamide gel electrophoresis with 10–250 kDa prestained protein ladder (#PG-PMT2922; Genetix) used for molecular weight reference. Proteins were transferred onto a nitrocellulose membrane (Bio-Rad) at 15 V, 2.5 A for 35 min. The membrane was blocked with 5% skimmed milk for 1 h. The membrane was incubated overnight at 4 °C with the primary antibodies: EZH2 mAb (1:1000; #5246), JARID2 Rabbit mAb (1:1000;

#13594), SUZ12 Rabbit mAb (1:1000; #3737), EZH1 Rabbit mAb (1:1000; #42088), EED Rabbit mAb (1:1000; #51673), AEBP2 Rabbit mAb (1:1000; #14129). H3K27me3 (1:1000), UTX (1:1000), JMJD3 (1:1000), HA-Tag Rabbit mAb (1:500; #3724), GAPDH Rabbit mAb (1:2000; #5174), β-actin (1:2000; #3700), H3 Rabbit mAb (1:2000, #4499) (Cell Signaling Technology), S-Nitrocysteine mAb (1:500; #94930) Ubiquitin mAb (1:1000;#3933). Subsequently, the blots were washed with TBS-T(x3) and incubated with a secondary peroxidase-conjugated anti-rabbit or mouse IgG antibody (1:2000) (#7074 or #7076, Cell Signaling Technology) overnight. These conjugation reactions were detected using the Clarity™ (#1705061) or Clarity™ Max Western ECL Substrates (#1705062) (Bio-Rad). Details of all the antibodies used for immuno-blot analysis are provided in Supplementary Data 3.

## Immunofluorescence and Confocal Microscopy

Following SNP treatment in EA.hy926 and transfected HEK-293, the cells on coverslips were fixed using 2% paraformaldehyde (10 min). The coverslips were then treated with 0.1% Triton X-100 (5 min) for permeabilizing the cells. Blocking was done with 2% bovine serum albumin for 60 min at room temperature. Next, cells were incubated with primary antibodies EZH2 Rabbit mAb (1:1000) or HA-tag (Alexa Fluor™ mAb 647 Conjugate) (1:2000; #37297) overnight at 4 °C. The cells were co-stained with Alexa Fluor™ Plus 480 conjugated antirabbit IgG secondary antibody (1:4000; #A32732) (Thermo Fisher Scientific) for 1 h followed with phalloidin for staining F-actin (1:5000; #A22287 or #A12379, Thermo Fisher Scientific) for 30 min. At last, the cells were stained with 1 µM of DAPI (#D9542; Sigma-Aldrich) for 10 min for nuclear staining. Imaging was done using Zeiss ApoTome 2.0 microscope (Carl Zeiss) and fully Spectral Confocal Laser Scanning Microscope (#LSM 880Carl Zeiss).

## Histone Methyltransferase Activity/Inhibition assay

EZH2 methyltransferase activity assay was performed using Epi-Quik™ HMT Activity/Inhibition Assay Kit (H3-K27) (# P-3005, Epigentek) and the standard protocol as provided by the supplier was followed. In brief, nuclear extracts were prepared using the subcellular fractionation protocol as mentioned above. Specified volumes of histone assay buffer, adomet, and biotinylated substrate were added to each strip well followed by adding nuclear extracts (10 µg) along with SNP (500 µM) or GSNO (100 µM) and mixing gently. The assay strips were then incubated for 60 min at 37 °C. For the blank, nuclear extracts were replaced with equivalent volume of histone assay buffer. For the standard curve, HMT standard solution at different concentrations (ex: 0.2, 0.5, 1.0, 2.0, 5.0, and 10.0 ng/µl) were added in place of nuclear extract. Each well was then washed with wash buffer for three times followed by adding capture antibody (dilution of 1:100) and incubated at room temperature for 1 h on an orbital shaker. Wells were then washed four times with wash buffer followed by the addition of detection antibody and incubated for 30 min at room temperature. The wells were again washed four times prior to adding developing solution, incubating at room temperature for 10 min and measuring the absorbance at 450 nm. HMT inhibition was then calculated using a standard formula as referred by the manufacturer.

## Coimmunoprecipitation experiments

Protein extraction was carried out using 1× RIPA buffer. After washing the Dynabeads (SureBeads™ Protein A Magnetic Beads, #1614013, BioRad) in PBS-T(×3), they were preincubated with the EZH2 mAb, HA-tag mAb, ubiquitin mAb or S-Nitrocysteine mAb (1:50) for 2 h. The beads were then washed and incubated overnight with a protein cell lysate containing 1000 µg of total protein for pulldown of the desired protein from the cell lysate. The cell lysate was then heated at 70 °C for 10 min and magnetized to be separated from the beads. This was then processed for immunoblotting experiments.

## Ex vivo experiments using rat aortic tissues

All the experimental procedure involving rodent studies were reviewed and approved by the IAEC of BITS Pilani, Pilani Campus (Protocol Approval No: IAEC/RES/23/05/Rev-1/28/27). Ex vivo experiments were performed using Male Wistar rats aged 12–16 weeks. After anesthetization, the rats were dissected from the ventral end. PBS was used to perfuse the heart and aorta, eliminating blood cells from the vessels. The primary aortas were then collected, and the fatty tissue layers were removed carefully. For acquiring the aortic explants, the aorta was cut in a size of 4 and 5 mm cylindrical pieces. Following a PBS wash, the explants were cultured in HiEndoXL™ EC expansion medium and incubated in the complete growth medium for 12 h before initiating the treatment. Afterward, the tissue fractions were homogenized and subjected to sonication after suspending them in RIPA lysis buffer. The protein was estimated using Bradford assay followed by immunoblotting studies.

## En face immunohistochemistry

All the experimental procedure involving rodent studies were reviewed and approved by the IAEC of BITS Pilani, Pilani Campus (Protocol Approval No: IAEC/RES/23/05/Rev-1/28/27). Male Wistar rats aged 12–16 weeks were anesthetized and dissected from the ventral end. PBS was used to perfuse the heart and aorta to eliminate blood cells from the vessels. The primary aortas were then collected, and the fatty tissue layers were removed carefully. For acquiring the aortic explants, the aorta was cut in a size of 8–10 mm cylindrical pieces. Following a PBS wash, the explants were cultured in HiEndoXL™ EC expansion medium and incubated in the complete growth medium for 12 h before initiating the treatment with GSNO (100 μM) for 2 h. Next, aortic rings were cut open longitudinally and permeabilized with PBS containing 0.1% Triton X-100 for 10 min and blocked by TBS containing 10% goat serum and 2.5% Tween 20 for 30 min. Aortas were incubated with 10 μg/ml rabbit anti-EZH2 (#5246, Cell Signaling Technology; rabbit IgG was used as a control) and 12.5 μg/ml mouse anti–VE-cadherin (an EC marker; Santa Cruz) in the blocking solution overnight. After a PBS rinse, anti–rabbit IgG and anti–mouse IgG (1:1000 dilution; Alexa Fluor 546 and 488, respectively; Invitrogen) were applied for 1 h at room temperature. Nuclei were stained using DAPI (Invitrogen). Images were acquired using a fully Spectral Confocal Laser Scanning Microscope (#LSM 880, Carl Zeiss).

## Diabetic kidney disease model

Male Sprague Dawley (SD) rats aged eight weeks (200–220 g) were procured from the Central Animal Facility (CAF) of Birla Institute of Technology and Science Pilani (BITS-Pilani) in accordance with the protocol approved by the IAEC, BITS-Pilani (Protocol Approval No: IAEC/RES/31/13/REV-1/33/19). Animals were maintained under standard environmental conditions with feed and water ad lib. Animal studies are reported following the ARRIVE guidelines[69]. For Type 1 diabetes induction, male SD rats were injected with single dose of streptozotocin (STZ, 55 mg/kg; Sigma-Aldrich, St. Louis, MO) in 0.1 M citrate buffer (pH 4.5) or citrate buffer alone by tail vein injection after an overnight fast[70]. After 48 h of STZ injection, rats with plasma glucose levels >280 mg/dL were included in the study as diabetic rats and were housed for 6 weeks to develop diabetic kidney disease. Urine albumin and creatinine excretion was measured by the Albumin Assay kit (#ab108789 Rat Albumin ELISA kit, Abcam) and Creatinine Assay (#ab65340, Creatinine colourimetric Assay, Abcam) respectively after housing rats individually in metabolic cages for 24 h to analyze albumin-to-creatinine ratio. Upon perfusion with PBS, tissues were harvested and snap frozen in liquid nitrogen for further analysis.

## RNA Isolation, cDNA Synthesis and qPCR

Reverse transcriptase-quantitative polymerase chain reaction (RT-qPCR) was performed to measure different gene expressions at the transcription level. Upon reaching 80% confluency EA.hy926 cells underwent GSK-J4 treatment for 4 h followed by SNP exposure for 2 h. After that, RNA was isolated from the cells using Trizol Reagent (#15596, TRIzol™ Reagent; Life Technologies, Thermo Fisher Scientific). RNA isolation was succeeded by cDNA preparation from 1 μg of total RNA using iScript™ cDNA Synthesis Kit (#1708891; Bio-Rad Laboratories, Hercules, CA, United States). The quality and quantity of the RNA was measured using a Nano-Drop spectrophotometer (SimpliNano; GE Lifesciences). Before cDNA synthesis, DNA contamination was removed by pre-treating isolated RNAs with the DNase. This was followed by Real-time PCR where iTaq™ Universal SYBR® Green Supermix (#1725124; Bio-Rad Laboratories) was used with a total master mix volume of 10 μl and GAPDH was taken as the housekeeping gene. Data analysis was done by calculating delta-delta Ct. Details of the primers have been provided in the Supplementary Data 3.

## Chromatin Immunoprecipitation (ChIP) and subsequent quantitative PCR

To understand the promoter level enrichment of H3K27me3, Chromatin Immunoprecipitation (ChIP) experiments were performed using an Imprint® Chromatin Immunoprecipitation Kit (#86652 CUT & RUN assay kit, Cell signaling Technology). In brief, EA.hy926 cells (70% cell density) were subjected to GSNO (500 μM) treatment for 2 h. Treated EA.hy926 were harvested ($1 \times 10^6$ cells), washed, and cross-linked with 1% formaldehyde in DMEM (10 min at room temperature). After washing in PBS, the cell pellet was re-suspended and tagged with Concanavalin A Beads. Following the manufacturer's protocol, pulldown was performed with H3K27me3 antibody at a dilution of 1 mg/ml. Only input samples were sheared by sonication for 30 s pulses. The samples were then washed, reverse cross-linked, and treated with proteinase K to obtain purified DNA fragments through silica column. qPCR was performed using primers targeted to amplify regions of human gene promoters (Supplementary Data 3).

## Prediction of S-nitrosylation site in EZH2 protein using GPS-SNO prediction tools

To identify the cysteine residues that are likely to be S-nitrosylated in EZH2 protein, we performed an in silico analysis of the same by using GPS-SNO prediction tools as available at http://sno.biocuckoo.org/. This tool was developed by a group of scientists by manually collecting 467 experimentally verified S-nitrosylation sites in 302 unique proteins from scientific literature. This software can be used for the prediction of S-nitrosylation sites. The developer performed extensive leave-one-out validation and four, six, eight, tenfold cross-validations techniques to calculate the prediction performance and system robustness[71]. More than 250 publications have used these tools for the prediction of possible S-nitrosylation sites. We next used the human EZH2 primary protein sequence available in the Addgene for hEZH2 plasmid (#24230, Addgene) and predicted the possible cysteine residues using the GPS-SNO prediction tool. EZH2 in total has 34 cysteine residues, and according to the partially resolved crystal structure of EZH2, none of these residues are involved in di-sulphide bonds and are likely to be cysteine residues with free –SH group. The analysis through the GPS-SNO platform revealed three unique sites at cysteine 260, 329, and 700 which are likely to be S-nitrosylated.

## Site-directed mutagenesis reaction

Phusion Site-Directed Mutagenesis Kit (#F541Thermo Fisher Scientific™) was used to insert point mutations at the specific positions in the pCMV-HA hEZH2 plasmid as mentioned in the manufacturer's protocol. Prior to this, the primers were phosphorylated using T4 Polynucleotide Kinase (#EK0031, Thermo Fisher Scientific™). After the ligation step, the mutated product was transformed in competent DH5-Alpha E. coli cells. The sequence of primers for inserting point

mutations at the predicted sites to convert cysteine to serine are provided in Supplementary Data 3.

## Biotin switch assay

EA.hy926 cells or EZH2 WT/mutant overexpressing HEK-293 treated with NO donor SNP (500 μM) for 2 h were processed with the help of a Biotin Switch Assay kit (Abcam) using the standard manufacturer protocol. With this, all the "S-NO" (S-nitrosylated) groups were replaced with Biotin, forming an S-Biotin complex, which was detected by incubation with the streptavidin bound HRP reagent. After this, the samples were immunoprecipitated with EZH2 specific antibody using the above-mentioned protocol. This was followed by dot blot and immunoblotting experiments for the detection of the S-nitrosylation of the EZH2 protein.

## Detection of S-Nitrosylation using iodoTMT labelling ™ reagent

For S-nitrosylation analysis of recombinant protein, EZH2 recombinant protein (#MBS2097714, MyBioSource) were treated with GSNO (100 μM) for 30 min. The samples were then processed for iodoTMT labelling with Pierce TM S-Nitrosylation Western Blot kit (#90105, Thermo Fisher Scientific™). A total of 20 μg/ml of EZH2 recombinant protein sample was prepared for both control and GSNO treated conditions in 100 μl of HENS Buffer and processed for iodoTMT assay. For cell-based assays, nuclear extracts were isolated from cultured EA.hy926 cells followed by immunoprecipitation with EZH2 antibody. Immunoprecipitates were subjected to GSNO treatment for 30 min at room temperature followed by processing with iodoTMT protocol. For the S-nitrosylation analysis of EZH2 C329S C700S mutant, EA.hy926 was transfected with 250 ng/mL human WT pCMVHA hEZH2 plasmid (#24230, Addgene) containing point mutations at C329 and C700 and Lipofectamine 2000 for 4 h. After 48 h, the cells were subjected to GSNO stimulation for 30 min. The plates were taken out and subsequently scraped using 10% RIPA lysis buffer and processed with iodoTMT protocol. Next, MMTS at a volume of 2 μL was added to each well, after the samples were briefly vortexed for 1 min, they were incubated at room temperature for 30 min to block all free thiols. The protein was then precipitated by adding 600 μl of pre-chilled (−20 °C) acetone and freezing the samples at −20 °C for 2 h for MMTS removal. The samples were centrifuged at 4 °C for 10 min (at $12000 \times g$). The tubes were inverted to decant the acetone, and the white pellet was dried for 10–15 min. The pellet was re-suspended in 100 μl of HENS Buffer and 1 μl of the iodoTMTzero labeling reagent was added followed by the addition of 2 μl of 1 M sodium ascorbate and vortexed briefly to mix and incubated it for 2 h at room temperature. We then again precipitated by adding 600 μl of pre-chilled (−20 °C) acetone and freezing the samples at −20 °C for 2 hs to remove unbound iodoTMT and sodium ascorbate. The pellet obtained was resuspended in 30 μl of HENS buffer and the samples were then processed for immunoblotting experiments with anti-TMT antibody.

## Scratch wound healing assay

EA.hy926 cells were grown in a 24-well plate. They were pre-treated with GSK-J4 for 4 h following SNP treatment at regular intervals (0–24 h) to measure the wound healing under different treatment conditions or in combination. For, GSNO, bradykinin, and GSK126 experiments, wounds were created and the cells were treated with these compounds and healing was followed for 24 h. Imaging was done at definite intervals to calculate the wound healed by cell migration. The wound was created in a straight line. Another perpendicular wound to the first one was drawn to create a cross-shaped wound using a 1 mm microtip.

## Apoptosis assay

Dead Cell Apoptosis Kit with Annexin V FITC and propidium iodide (PI) for flow cytometry (#V13242, Thermo Fisher Scientific) was employed to evaluate cellular apoptosis of EC exposed to SNP (500 μM) for 24 h.

EA.hy926 cells were treated with SNP and further processed following manufacturer protocol. Briefly, the cells were harvested after the incubation period and washed twice in cold PBS. Following centrifugation, the cell pellet was resuspended in 1× annexin-binding buffer (approx. $1 \times 10^6$ cell/ml), and 5 μl of FITC annexin V (component A) and 1 μl of (PI) (100 μg/ml) were added to each 100 μl of cell suspension. After 15 min incubation at room temperature, the cell suspension was further diluted with 400 μl PBS and the stained cells were analyzed using the CytoFLEX flow cytometer (Beckman Coulter). Apoptosis, both early and late, was determined from the respective Annexin V and Annexin V-PI positive quadrants, after subtracting the autofluorescence.

## Cell viability assay

Viability of cells was measured by 3-(4,5-dimethylthiazol-2-yl)-2,5-diphenylte- -trazolium bromide (MTT) assay. EA,hy926 were cultured in 96-well plates overnight, followed by exposure to SNP (500 μM) for 24 h. Thereafter, MTT was added to all treated and control cells, and cells were incubated for 4 h. Formazan crystals were solubilized in DMSO, and readings were obtained at 495 nm with a differential filter of 630 nm using a micro-plate reader (Start-fax 2100). Percentage of viable cells was calculated as % viability = (mean absorbance value of drug-treated cells)/(mean absorbance value of control) × 100.

## Transformation experiments and plasmid Isolation

The ligation product from the SDM reaction was mixed with 100 μl of competent DH5-Alpha cells and kept on ice for 25 min. This was followed by heat shock at 42 °C for 30 s. After adding 900 μl of Super Optimal Broth medium, the product was kept at 37 °C for 1 h. This was followed by centrifugation at 3000 rpm for 5 min. About 800 μl of supernatant was removed, and the pellet was resuspended in the remaining 100 μl media. The product was then spread on the LB Agar plate with the help of a spreader and kept overnight at 37 °C. A single colony picked from the plate was used to inoculate 5 ml of Luria Broth (LB) containing ampicillin, which was kept overnight at 200 rpm and 37 °C. This was used for plasmid isolation, which was performed using the manufacturer's protocol (#12123, Qiagen Plasmid Minikit). The isolated plasmid was then sent for Sanger Sequencing to confirm the insertion of point mutations at the desired location and then used for further transfection experiments.

## Sample preparation for mass spectroscopy analysis to identify interacting partners

HUVEC were treated with GSNO (100 μM) for 30 min followed by lysis with RIPA buffer. Two independent biological replicates for both control and GSNO treated samples were processed for proteomics based interactome analysis. After washing the Dynabeads (SureBeads™ Protein A Magnetic Beads, in PBS-T(×3)), they were pre-incubated with the EZH2 mAb, (1:50) for 2 h. The beads were then washed and incubated overnight with a protein cell lysate containing 1000 μg of total protein for pulldown of EZH2 from the cell lysate. The cell lysate was then heated at 70 °C for 10 min and magnetized to be separated from the beads. Dynabead-bound immunoprecipitated proteins were eluted by heating the samples in 1× PBS. Eluted protein per sample was used for digestion and reduced with 5 mM tris(2-carboxyethyl)phosphine (TCEP) followed by alkylation using 50 mM iodoacetamide. The samples were next digested with Trypsin (1:50, Trypsin/lysate ratio) at 37 °C for 16 h. Digested samples were then cleaned using a C18 silica cartridge to eliminate the salt and next dried using a speed vac. Pellet obtained upon drying was resuspended in buffer A (2% acetonitrile, and 0.1% formic acid).

## Mass spectrometric analysis of peptide mixtures

Experiments were performed on an Easy-nlc-1000 system coupled with an Orbitrap Exploris mass spectrometer. Peptide sample (1 μg in

total) was loaded on C18 column 15 cm, 3.0 µm Acclaim PepMap (Thermo Fisher Scientific) and separated with a 0–40% gradient of buffer B (80% acetonitrile, and 0.1% formic acid) at a flow rate of 500 nl/min) and injected for MS analysis. LC gradients were run for 110 min. MS1 spectra were acquired in the Orbitrap (Max IT = 60 ms, AGQ target = 300%; RF Lens = 70%; R = 60 K, mass range = 375–1500; Profile data). Dynamic exclusion was employed for 30 seconds excluding all charge states for a given precursor. MS2 spectra were collected for top 20 peptides. MS2 (Max IT = 60 ms, R = 15 K, AGC target 100%).

### Data processing
All samples were processed, and RAW files generated were analyzed with Proteome Discoverer (v2.5) against the Uniprot Human database. For dual SEQUEST and AMANDA search, the precursor and fragment mass tolerances were set at 10 ppm and 0.02 Da, respectively. The protease used to generate peptides, i.e., enzyme specificity was set for trypsin/P (cleavage at the C terminus of "K/R: unless followed by "P"). Carbamidomethyl on cysteine as fixed modification and oxidation of methionine and N-terminal acetylation were considered as variable modifications for database search. Both peptide spectrum match and protein false discovery rate were set to 0.01 FDR.

### Model and protein preparation for in silico analysis
The crystal structure of the Human Polycomb Repressive Complex 2 (PDB ID: 5HYN) was acquired from the Protein Data Bank[37]. This complex exhibits multiple missing loops, particularly within the EZH2 residues at positions 1–9, 182–210, 249–256, and 345–421. To address these structural gaps, the EZH2 structure was downloaded from AlphaFold, assigned an AFDB accession code AF-Q15910-F1, and possessed an average pLDDT score of 76.25 (Uniport ID: Q15910)[72]. Subsequently, the EZH2 AlphaFoldmodel was superposed to the original PDB file (5HYN) to obtain the PRC2 complex. Finally, EED, H3K79M, and JARID2 K116m3 proteins were deleted to keep only the EZH2-SUZ12 complex for further modelling studies. The EZH2-SUZ12 complex was pre-processed using Protein Preparation Workflow tool in Schrödinger suite 2022[73]. Hydrogens were added, charges were assigned, bond orders were refined, and all water molecules and nonstandard residues were deleted before proceeding. The protein backbone was minimized by employing OPLS 2005 force field. Further residues Cys324 and Cys695 were S-nitrosylated by Vienna-PTM[74].

### Molecular dynamics (MD) simulations
The charges of both wild-type and mutant complexes were neutralized by placing a total of 10 Na$^+$ ions at positions with high electronegative potential. Both complex and counter ions were then placed in a preequilibrated cubic box of SPC/E water molecules. The periodic box of water was extended to a distance of 10 Å from the protein complex and counter ions. Another 442 NaCl molecules were added to the system to maintain a 150 mM salt concentration. The prepared systems were subjected to 50,000 steps of steepest descent energy minimization. The structural fluctuations and stability of the relaxed protein complex were analyzed by time-dependent molecular dynamics (MD) simulation studies using GROMACS 5.1.5. The particle mesh Ewald method (PME) was used for the calculation of electrostatic interactions[75]. Periodic boundary conditions were imposed in all directions. The longrange electrostatic interactions have been calculated without any truncation, while a 10 Å cutoff was applied to Lennard–Jones interactions. The LINCS-like algorithm was employed to restrain hydrogencontaining bonds only and handle long-range electrostatic interactions. SHAKE algorithm was applied to constrain the bond involving hydrogens[76]. The temperature was controlled at 300 K using Langevin dynamics with the collision frequency 1. A time step of 2 fs was used, and the structures were saved at every 10 ps interval for the entire duration of the MD run. The equilibrium phase comprised of two short

1 ns simulation in NVT and NPT ensembles, utilizing a Berendsen thermostat and Parrinello-Rahman barostat respectively at 300 K and 1 bar pressure. The unrestrained 1 µs simulation with NPT ensembles at 300 K was considered as production simulation.

### Structure analysis
Built-in features of GROMACS, such as gmx rms, rmsf, and hbond, were employed to assess the root-mean-square deviation (RMSD), RMSF, and hydrogen bonds (Hbond), respectively, throughout the trajectory. Visualization of the structures was performed using PYMOL and VMD software[77,78].

### Binding energy calculations using MM/PBSA
The MM-PBSA protocol was employed to determine the effect of PTMs addition on the binding of subunits within a chosen EZH2-SUZ12 complex. The free binding energies of each complex were computed utilizing the equations provided below. Briefly, the provided set of equations were applied to represent an imaginary AB dimer, where A corresponds to EZH2 and B to SUZ12.

$$\Delta Gbind = GAB - (GA + GB) \tag{1}$$

Where GAB is the binding free energy of the EZH2-SUZ12 complex, GA is the binding free energy of the EZH2, and GB is the binding free energy of the SUZ12.

$$\Delta\Delta G\_bind = \Delta G\_bindABnitrosylation - \Delta G\_bindABwildtype \tag{2}$$

To evaluate the impact of incorporating PTMs into the complex on their binding, we compute the disparity in binding free energy (ΔΔGbind) between the S-nitrosylated and wildtype complexes using Eq. 2. The binding free energy was determined by utilizing the 1 µs trajectory obtained from molecular dynamics simulations, and calculations were performed on every 500 frames extracted from the last 500 ns of the trajectory using the g_mmpbsa tool integrated with GROMACS[79–81].

### Statistics and reproducibility
All statistical analysis was carried out using GraphPad Prism 9.5.1 and Prism 10 Software. All the values are expressed as the mean ± SD. A two-tailed unpaired $t$-test was used to determine the statistical differences in comparisons between the two groups and One-way ANOVA with a post-hoc Tukey test was used for comparison between multiple groups. All $p$ values < 0.05 are considered statistically significant and $p$ values are indicated in the respective figures.

### Reporting summary
Further information on research design is available in the Nature Portfolio Reporting Summary linked to this article.

## Data availability
We have used PDB Code: 5HYN for modeling EZH2 and SUZ12 complex to perform MD simulation study. The raw proteomics data has been deposited to the ProteomeXchange CONSORTIUM via the PRIDE partner repository with the dataset identifier PXD050209. Source data are provided with this paper.

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

## Acknowledgements

This work was supported by the Core Research Grant from Anusandhan National Research Foundation (formerly Science and Engineering Research Board)-Department of Science and Technology, Govt. of India (CRG/2022/002209) to S.M. This work was also partly supported by a Competitive Research Grant from the Department of Biotechnology, Govt. of India (BT/PR33144/MED/30/2170/2019) to SM. AS was supported by a Senior Research Fellowship from CSIR, Govt. of India (File No- 09/719(0107)/2019-EMR-I). Y.T.K., and S.M.T. are supported by a graduate fellowship from BITS Pilani. S.T., and N.P.T. was supported by a graduate fellowship from BITS Pilani. S.K.R. is supported by a graduate fellowship from the Department of Science and Technology-Innovation in Science Pursuit for Inspired Research fellowship (DST/INSPIRE/03/2019/000582). We gratefully acknowledge the technical assistance of Mr. Suman Kumar (Confocal Facility, BITS Pilani, Pilani Campus) and BITS Pilani, Pilani Campus for providing the high-performance computing (HPC) facility for simulation study.

## Author contributions

A.S. designed and performed the experiments, analyzed the data, and wrote the first draft of the manuscript. R.B. and Y.T.K. designed and

performed the experiments, analyzed the data and revised the manuscript. S.K.R., S.C., H.J., N.P.T., and S.T. performed experiments and analyzed data. S.M.T., S.S., and Shibasish performed all the molecular simulation studies and wrote the associated methodology and results section of the manuscript. S.M. secured the funding, designed the experiments, supervised the study, wrote and edited the final draft of the manuscript.

## Funding

## Competing interests
The authors declare no competing interests.
