## [Peer Review file · Nature Communications]

S-nitrosylation of EZH2 alters PRC2 assembly, methyltransferase activity, and EZH2 stability to maintain endothelial homeostasis

Corresponding Author: Professor Syamantak Majumder

Version 0:

Reviewer comments:

Reviewer #1

(Remarks to the Author)

The manuscript explores the role of EZH2 S-nitrosylation in modulating H3K27me3 levels and endothelial cell function. The authors demonstrate that NO induces S-nitrosylation of specific EZH2 cysteine residues, causing SUZ12 dissociation from the PRC2 complex, reduced methyltransferase activity, and disrupted nuclear localization. H3K27me3 levels decrease due to early SUZ12 dissociation. Computational analysis and mutagenesis identify cysteine residues responsible for EZH2 instability and loss of catalytic activity upon S-nitrosylation. Furthermore, molecular dynamics simulations suggest impaired binding between SUZ12 and S-nitrosylated EZH2.

Overall, while the manuscript provides intriguing insights into the S-nitrosylation-dependent regulation of EZH2 and its impact on endothelial homeostasis, the evidence provided for the molecular regulation of NO-mediated S-nitrosylation of EZH2 and its effect on H3K27me3 levels in endothelial cell function is weak. And the consistency between the author's claims and the data presented in the figures should be carefully reviewed and revised to ensure clarity and accuracy. Importantly, the experimental design should be re-evaluated to establish a stronger link between the hypothesis and the experiments employed.

1. The authors should provide clarification regarding the separation of two western blot images in Figure 1A and Figure 1B, as both images depict EA.hy 926 cells exposed to NO but with different durations. Furthermore, the authors claim that there is no change in EZH2 expression after 1 hour of SNP treatment, but a significant reduction is observed after 2 hours. This reduction appears to occur within an additional hour of treatment. This discrepancy requires careful discussion and explanation to ensure the clarity of the findings. Additionally, the authors should address the inconsistency in the use of internal controls for the western blot analysis for EZH2. Supplementary Figure 1H suggests that EZH2 is primarily located in the nucleus. It would be advisable to consistently use a nuclear control for EZH2 normalization, such as Histone H3, throughout the western blot experiments.

2. The authors utilized bradykinin to induce endogenous NO production in order to investigate whether NO generated from the eNOS machinery can elicit similar effects as observed with SNP treatment. While the authors treated HUVECs with different doses of bradykinin, they performed a time-dependent treatment of bradykinin on a different type of endothelial cell source EA.hy926. It would be more appropriate and scientifically rigorous to perform the time-dependent treatment of bradykinin in the same cell type as the one used for the dose-dependent treatment. I recommend that the authors address this concern and provide a justification.

3. The authors demonstrate EZH2 and H3K27me3 levels ex vivo in rat aorta exposed to NO using the lysate of rat aortic explants. However, considering that the endothelial cells are not the predominant cell population within the aorta, it is essential to ensure that the observed molecular regulation is specifically attributed to the endothelial cells. Therefore, I recommend that the authors perform an enface staining to visualize EZH2 expression specifically in the endothelial layer.

4. In the co-immunoprecipitation experiment presented in Figure 1G, it appears that there is a noticeable difference in the expression levels of SUZ12 and EZH2 in the input data set. To better interpret this observation, it would be highly recommended for the authors to provide quantification data to support their findings.

5. It appears that the majority of EZH2 is consistently localized in the nucleus rather than the cytosol, both in the absence and presence of SNP treatment, as indicated in Figure 1H. This finding is in conflict with the author's claim of cytosolic translocation of EZH2 in EC upon NO exposure.

6. In this study, the authors aim to address concerns regarding the turnover of H3K27me₃, which is influenced not only by the methyltransferase EZH2 but also by the H3K27me₃-specific demethylases UTX and JMJD. However, I have some concerns regarding the quality of the western blot analysis, particularly for the expression of JMJD and its internal control. It remains unclear whether the expression of JMJD is altered by SNP treatment. This is a critical issue since the central point of this study is the regulation of H3K27me₃ after NO exposure through EZH2.

7. In several figures, such as Figure 2E, Figure 6A, B, and D, the authors primarily detected and reported the expression of the HA tag rather than EZH2 itself, in the cells transfected plasmid containing HA tagged EZH2. While the use of an HA tag can be a useful tool for tracking and studying protein expression, it is essential to directly demonstrate the expression of the protein of EZH2. Simply detecting the HA tag does not provide conclusive evidence of EZH2 expression or its functional implications.

8. In this study, the authors claim that EZH2 undergoes significant ubiquitination upon NO exposure in endothelial cells. However, upon careful examination of the blotting image provided, it appears that the level of ubiquitinated EZH2 in the cells remains unchanged after 30 minutes of NO exposure. It would be highly recommended for the authors to provide quantification results for this co-immunoprecipitation experiment.

9. The authors employed GSK-J4, a selective inhibitor of the H3K27me₃-specific demethylases JMJD3 and UTX, to investigate the effects on endothelial cell migration and gene expression changes, as shown in Figure 5. However, I have concerns regarding the direct relevance of using GSK-J4 to inhibit H3K27me₃ in reflecting the outcome of inhibiting EZH2 catalytic activity. The presented data does not effectively support the idea that inhibiting EZH2 catalytic activity affects endothelial cell function through the modulation of H3K27me₃. I recommend that the authors revise their experimental design and provide additional evidence to establish a more direct connection between inhibiting EZH2 catalytic activity and the functional outcomes observed in endothelial cells. This can be achieved by employing a more specific and direct approach to inhibit EZH2, such as using EZH2-specific inhibitors or genetic approaches targeting EZH2 itself.

10. The authors aimed to investigate the molecular regulation of S-nitrosylation of EZH2 in the regulation of endothelial functions in this manuscript. However, I have concerns regarding the choice of cell line used for the molecular and functional studies presented in Figure 2E and the whole Figure 6. Instead of using the endothelial cell EA.hy 926, the authors utilized the Human Embryonic Kidney (HEK)-293 cell line. The authors should discuss the rationale behind their choice of HEK-293 cells instead of endothelial cells and address any potential limitations or differences that may arise from using a non-endothelial cell line. And I recommend that the authors perform the molecular and functional studies using the EA.hy 926 cell to validate and strengthen the relevance of their findings to endothelial cells.

11. The authors try to demonstrate a novel molecular regulation of NO-mediated S-nitrosylation of EZH2 in modulating H3K27me₃ levels in endothelial cells in the manuscript. However, I would like to raise a few concerns regarding the rationale and implications of this study. Specifically, it is important to consider the relevance of these findings in vivo and the potential implications for physiological or pathological processes. Additionally, further experiments demonstrating the therapeutic potential of targeting this molecular regulation to improve endothelial dysfunction in disease models would greatly enhance the significance of the study.

Reviewer #2

(Remarks to the Author)

Dear Authors, in your paper it is described, for the first time, NO-dependent S-nitrosylation of the PRC2 component EZH2 in ECs and consequent effects such as detachment from PRC2 member SUZ12, decreased methyltransferase activity and nuclear-cytoplasmic shuttling. S-nitrosylation of EZH2 was not yet reported, therefore this finding is novel, while following data already reported for other chromatin regulators.

The flow of the experiments shown in this paper is clear. Some data are convincing, other are not. Therefore this reviewer has some suggestions to improve the MS, as follows.

Overall comment: experiments have been performed in part by using SNP in part GSNO. This reviewer suggests the use GSNO as the only exogenous NO source. SNP is highly toxic, especially at the 500 μM concentration which rapidly raise the level of RNS within the cell, leading to DNA damage.

Results section: "Nitric oxide exposure interplayed with EZH2 including PRC2 assembly, its methyltransferase activity, subcellular localization, and stability"

Major:

- As discussed above, this reviewer believes 500 μM SNP to raise too high and toxic concentration of NO. It is known that SNP induces a burst of NO (and consequent nitrosative stress). Therefore: a) in authors experimental conditions, which is the level of γH2AX? This reviewer would like to evaluate the health state of the cells; b) which are the levels of nitrites/nitrates/peroxynitrites?
- Figure 1: panels B, D and E are not PQ. Please, provide better wb images. Panel G: this reviewer has some concerns on this IP. Starting from the right panel, the authors show the input levels of PRC2 protein at 30' of NO treatment. It seems that

SUZ12, EED (please, provide a better wb image for EED) and EZH2 decrease in the presence of NO (levels of GAPDH are highly similar in ctr vs treated cells). However, the authors says that at 30' the only detected a diminished H3K27me3 (as shown in the upper panel of this figure). Further, looking at the left panel showing IP results, it seems that also auto-IP levels of EZH2 are decreased in the presence of NO, raising the hypothesis that impairment of SUZ12 recruitment could indeed depend on decreased EZH2 IP levels. Further, the results shown for SUZ12 seem to be related more to technical issues than to a real reduction at the protein level. Please provide a better wb image for IP-SUZ12. The authors should also show densitometric analyses of blots in panel G (both left and right), normalizing the levels of IP PRC2 components on IP EZH2 content in ctr vs treated cells. This should clarify the data.

- Line 145: the authors cannot use the word “degradation” at this stage, They have still not proven EZH2 to be degraded by NO (i.e. EZH2 decrease could be dependent on NO-induced translation inhibition by indirect mechanisms such as miRNA induction, for example).
- Line 172: again, the authors have not specifically demonstrated a NO-impact on EZH2 methyltransferase activity. This is an indirect statement as they showed SUZ12 detachment (to be confirmed) from EZH2. Further, since the authors show in panel H EZH2 nuclear-cytoplasmic shuttling, the reduced levels of H3K27me3 may rely also on this phenomenon. The authors should show a specific methyltransferase assay on recombinant H3K27 to verify this assumption.
- Panel H: according to the wb on fractionated cell extracts, the cytosolic translocation of EZH2 seems to be transient and limited at the 30' time point. If we normalize the data on GAPDH levels, at 60' cytosolic EZH2 returns at ctr level. Authors should show densitometric analysis of this experiment. According to this observation, confocal analyses should be performed at the 30' time point.
- Supplementary figure 1B: time of treatment is not indicated.
- Supplementary figure 2C: quality of EED, AEBP2 and histone H3 is too poor. This reviewer understand it is a supplemental data, but the quality has to be a little bit enhanced.

Minor:

- Line 122: this is not completely true. An interplay between EZH2 and eNOS has been already reported (Mitic T, Mol Ther, 2015).

Results section: “Nitric oxide exposure caused S-Nitrosylation of EZH2 leading to early SUZ12 dissociation and further altering its binding partners”

Major:

- Figure 2D, upper panel: whereas data on EZH2 S-nitrosylation are convincing, this reviewer believes that data on SUZ12 are not, especially when comparing original wb data (4th page of the related MS file). SUZ12 blot in the MS is too modified. Lower panel: the input levels of SUZ12 are decreased upon NO exposure. This is particularly evident in original wb data (page 5 of related MS file). In this regard, this reviewer has to say that original wb and blot shown in the MS do not correspond. This also supports this reviewer's doubt on SUZ12 detachment in the upper EZH2 IP.
- Figure 3B, please provide higher quality histone H3 wb data.
- The authors should show SUZ12 detachment by IP experiments as shown in figure 2.
- In MS experiment, do the author detected SUZ12 among the 48 EZH2 interacting proteins which disassociate upon NO exposure? Maybe this reviewer missed the point, but SUZ12 does not appear in the list of associated protein provided in supplementary table 1. This is a very important point
- Regarding 14-3-3 association, it would be interesting to validate this association in IP experiments

Results section: “Nitric oxide caused the degradation of EZH2 primarily through autophagosome-lysosome 223 pathway while inhibition of endogenous nitric oxide machinery reversed nitric oxide dependent degradation, activity and localization of EZH2”

Major:

- In figure 4E, a panel showing a western blot for EZH2 on fractionated cell extracts should be added

Minor: line 233: this reviewer would not use the term “heavily” ubiquitinated. If we compare the level of Ub-EZH2 with total EZH2 in figure 4A, the difference between ctr and NO-treated Ub-EZH2 decreases, also looking at the original wb data shown at page 8 of related MS file.

Results section: “Protecting the level of EZH2 downstream product H3K27me3 through inhibition of demethylases reversed nitric oxide dependent effect on endothelial migration and gene expression changes”

Major

- Real time experiments in figure 5C should be flanked by ChIPs to show H3K27me3 enrichment at promoters of the investigated genes

Results section: “S-nitrosylation of EZH2 protein at C329 and C700 causes conformational changes in EZH2-SUZ12 complex leading to loose association of SUZ12 with the SAL domain of EZH2”

Major: This section could be strengthened by IPs performed on protein extracts from EZH2 C329S/C700S double mutant expressing cells, cultured in the presence or absence of NO, and subsequent wb revealing SUZ12. (These experiments may be also shown in figure 6, as the authors prefer).

Reviewer #3

(Remarks to the Author)

In their manuscript entitled 'S-nitrosylation of EZH2 at C329 and C700 interplay with PRC2 complex assembly, 1

methyltransferase activity, and EZH2 stability to regulate endothelial functions', Sakhuja and co-workers describe that s-nitrosylation of EZH2 at cysteine residues 329 and 700 affects its stability and methyltransferase activity, thereby altering the endothelial cell gene transcriptional profile. These results advance our understanding of the mechanisms by which endothelial cell behavior is influenced by epigenetics and NO. The results are of interest to the cardiovascular field, the methodology is technically sound and the presented conclusions appear justified by the results. Yet, some improvements to the manuscript can be made.

Major comments:

- In their experiments, exogenous NO is used to study the downstream effects on EZH2 stability and activity. In a translational perspective, where would the authors suggest that these high levels of NO come from. Would this be reflected as a response to (over-)activation of eNOS and elevated endothelial NO synthesis, or else? Please provide data that the endogenous increase in NO (by increased eNOS activity) also results in the s-nitrosylation of EZH2 and reduction of its transferase activity.
- In rat aorta, the decrease in EZH2 expression after NO stimulation may occur in endothelial cells or vascular smooth muscle cells. Please use an imaging technique to show endothelial cell specificity of this response.
- Please add replicates and quantify the blots used in all figures (specifically figs. 1G and H, Suppl. Fig 1B-E, fig.2, fig 4A).
- Suppl. Figure 3 has smears on the blot and is therefore hard to interpret. Please provide a better representative example.
- EZH2 immunoblotting in Fig 4 shows 2 bands, whereas other EZH2 immunoblots only show 1 specific band. What causes this difference?
- The kinetic responses differ between the reduction of H3K27Me3 and the reduction in EZH2. What causes this difference? Is NO also affecting the demethylase activity of other enzymes? Just showing that NO has no effect on the transcription of EZH2 is not sufficient to conclude that only a reduction in EZH2 methyltransferase activity can explain this difference in kinetics.
- What are the responses to SNP for UTX and JMJD3 expression and do these differ from responses to BK? This is particularly of interest in the context of results presented in fig. 5, where NO is shown to interact with JMJD3 and UTX.
- Line 212 states that the authors "expected such association ... due to EZH2 cytosolic shuttling"; yet figure 2 does not confirm this statement. Please show more compelling evidence for cytosolic shuttling of EZH2 or revise the statement (similar in lines 214-215).
- Treatment of EA.hy926 cells with MG and Baf completely restores EZH2 expression after NO exposure. Please provide data to show this also holds true for its downstream repressive mark H3K27Me3.
- Please provide confirmative data to evidence show which NO-responsive genes are regulated by EZH2 in endothelial cells to elucidate whether the gene expression changes are derived from an interregulated mechanisms or from two synergistic pathways. ChIP-PCR or ChIP-seq should be performed for VEGFA, KDR, TBX20, TIE2, MMP2, TEK, FGF2 and ANGPT2.
- Please add the potential role of demethylating enzymes to figure 8 to indicate that the reduction in H3K27Me3 has a different kinetic than the NO-induced degradation of EZH2. The graphics is too biased now.
- There is a high redundancy between the introduction, manuscript body and discussion, whereas the novelty of the current findings are only minimally discussed. I suggest to remove the redundancy and add more discussion on which gap in knowledge the current results solve and how these contribute to a better understanding of epigenetic regulation in endothelial cells. Also add discussion on the interaction between methyltransferases and demethylases and how NO exposure may influence this.

Minor comments:

- Please provides all images of all analyzed blots in supplement. All blots should be unprocessed and uncut / uncropped blots, including size markers on the blots.
- There are multiple grammar and spelling errors throughout the manuscript. Please have the manuscript checked and corrected by a native English speaker.
- Define HTM at first mention in introduction.
- How were the concentrations of stimulants and inhibitors used determined? If concentrations were used from literature, please cite. If empirically determined, please add these data to the supplement.

Version 1:

Reviewer comments:

Reviewer #1

(Remarks to the Author)

Thank you for revising your manuscript in response to my previous comments and suggestions. I have carefully reviewed the additional experiments you have conducted to address the concerns raised in my initial feedback. I appreciate the effort you have put into strengthening your study based on the feedback provided. The additional data and analyses have enhanced the robustness of your study and clarified some of the points that were previously raised. I believe that the revisions have significantly strengthened the manuscript.

Reviewer #2

(Remarks to the Author)

Review of NCOMMS-24-08539A_R1

Dear Authors, this Reviewer is impressed by the amount of work you did and congratulate with all the lab staff and

collaborators. There are still some minor points to be clarified to make your paper faultless. Specifically:

1. Fig. S1B. In this reviewer's hand, high NO concentrations result in an increase in γ H2AX levels in a wide variety of cells, indicating DSBs on DNA. A lot of literature is in line with this statement. It is curious that Ea.hy926 cells display such a high baseline level of γ H2AX which decreases at the 4hrs time point of NO treatment. Maybe, samples have to be inverted? Is it conceivable that the last line corresponds to the negative ctr (no SNP)? Please, check the correct order of the samples, or give an explanation for the high DSBs quantity present in these cells. Further, this alternative interpretation of the data (an increase in γ H2AX levels upon NO exposure) would also correlate with the increase in nitrites concentration shown in fig, S1A. In this regard, this reviewer would like to see a time course of nitrites accumulation similar to the γ H2AX WB to correlate the data.
2. Fig. S5B This reviewer is sorry, but the densitometric analysis (and, consequently, the discussion of the related results) of wb reporting the CHX pulse-chase experiment is not convincing. I mean, if we look at the level of EZH2 at 30' and 1hr, in the presence of CHX, it seems that the major decrease is at 30', when comparing the related GAPDH levels which is much higher at 30' vs 1hr time point, whereas EZH2 levels are basically the same. Please, correct or provide a more representative figure of the experiment.
3. Include the additional analysis of MS data, which is actually for reviewer's eyes only, as a supplemental figure.

Reviewer #4

(Remarks to the Author)

In this study, the authors identified a comprehensive mechanism for NO regulation of EZH2 and PRC2 complex. The authors concluded that S-nitrosylation of EZH2 at C329 and C700 causes changes in the structure of EZH2, which leads to its dissociation with SUZ12 for quick degradation followed by altered H3K27me3. Although the authors provided some data on the beneficial role of NO in reversing hyperglycemia-driven endothelial inflammation upon S-nitrosylation of EZH2, I am not really convinced on the physiological significance of this study. Below please find my major comments.

Ea.hy926 is a somatic hybrid cell line, which can not fully reflecting the endothelial cell functions. Some key data should be further tested in primary endothelial cells.

L-NAME is not an eNOS-specific inhibitor. Further experiments with eNOS siRNA (for in vitro experiment) and eNOS knockout mice (for in vivo study) would be necessary for validating the mechanism data from this study.

This study used whole aorta for some experiments. It would be interesting to compare the data when endothelium layer is removed from rat aorta. It is also valuable to compare the difference between healthy aorta and aorta with endothelial dysfunctions from hypertensive animals.

Does direct modulation of EZH2 affect endothelial functions in vitro and in vivo?

Reviewer #5

(Remarks to the Author)

The authors presented an interesting set of experimental data using a NO donor (500uM SNP) and a transnitrosylating agent GSNO demonstrating that SNO at Cys329/700 in EZH2 results in SUZ12 dissociation from EZH2 bound PRC2 complex, sequentially reducing its methyltransferase activity, preventing its nuclear localization, and affecting its stability. The authors also made an extensive rebuttal to thoroughly address the questions from previous reviewers. Overall, their data in essence support the conclusion that SNO-EZH2 serves as novel epigenic mechanism to mediate NO signaling regulation of endothelial cell homeostasis.

While the study is of significance to advance NO signaling and it is logical to first use GPS-SNO prediction tool and simulation to identify the two SNO sites Cys329/700 in EZH2 and followed by mutations to functionally study SNO in EZH2, however, the conclusion will be solidified if these Cysteines are indeed the physiological NO-responsive SNO-sites in EZH2. The authors performed MS studies for determining EZH2-interacting proteins. It is kind of a surprise that they did not do MS identification and/or verification of the two sites in control and GSNO-treated endothelial cells or artery tissues, which would have made the pathway physiologically relevant.

Version 2:

Reviewer comments:

Reviewer #2

(Remarks to the Author)

Dear Authors,

this reviewer is satisfied by your responses. However, your MS still lacks accuracy. For example, legend of Supplementary figure 10 contains errors. Indeed, you report that GSK-126 treatment is in panel D and EZH2 siRNA in panels E and F, but what you show in the picture is exactly the contrary. Please, carefully check all the figure legends and the correspondance of the results with the figures.

Reviewer #4

(Remarks to the Author)

The authors have properly addressed my previous comments with careful revision and additional experiments.

Reviewer #5

(Remarks to the Author)

No additional comments

Response to Reviewer #1 Comments

General Comment: The manuscript explores the role of EZH2 S-nitrosylation in modulating H3K27me3 levels and endothelial cell function. The authors demonstrate that NO induces S-nitrosylation of specific EZH2 cysteine residues, causing SUZ12 dissociation from the PRC2 complex, reduced methyltransferase activity, and disrupted nuclear localization. H3K27me3 levels decrease due to early SUZ12 dissociation. Computational analysis and mutagenesis identify cysteine residues responsible for EZH2 instability and loss of catalytic activity upon S-nitrosylation. Furthermore, molecular dynamics simulations suggest impaired binding between SUZ12 and S-nitrosylated EZH2.

Overall, while the manuscript provides intriguing insights into the S-nitrosylation-dependent regulation of EZH2 and its impact on endothelial homeostasis, the evidence provided for the molecular regulation of NO-mediated S-nitrosylation of EZH2 and its effect on H3K27me3 levels in endothelial cell function is weak. And the consistency between the author's claims and the data presented in the figures should be carefully reviewed and revised to ensure clarity and accuracy. Importantly, the experimental design should be re-evaluated to establish a stronger link between the hypothesis and the experiments employed.

Response: We would like to thank the Reviewer and are grateful for the positive feedback on our work. We also appreciate the Reviewer for the insights and comments to improve the manuscript.

1. (a) The authors should provide clarification regarding the separation of two western blot images in Figure 1A and Figure 1B, as both images depict EA.hy 926 cells exposed to NO but with different durations.

Response: We initially performed independent immunoblot experiments by running cell lysates from shorter (up to 1 hour) and longer (2 hours and beyond) exposure to SNP. Thus, in the previous version of the manuscript, we had presented independent gels. However, in the revised manuscript, we have rerun the samples in a single gel and now all the time points (0 to 4 hours) are presented in a single blot for both EZH2 (Figure 1A) and H3K27me3 (Figure 1B).

(b) Furthermore, the authors claim that there is no change in EZH2 expression after 1 hour of SNP treatment, but a significant reduction is observed after 2 hours. This reduction appears to occur within an additional hour of treatment. This discrepancy requires careful discussion and explanation to ensure the clarity of the findings.

Response: We would like to thank the reviewer for bringing up this important point. Indeed, this was one of the most intriguing initial observations that changed the way we performed experiments to understand the entire phenomenon. Although, we have modestly explained the reasoning behind such observation in Page 4, Line 145 under the Result section of the previous manuscript file, we completely agree with the reviewer that such discrepancy in the observation should have been discussed properly to enhance the clarity and scientific explanation behind it. We now added a more detailed explanation in the Result and Discussion section of the revised manuscript with track changes on Page 4, Lines 162-168. To explain in brief, the scientific reasoning behind such observation, our data indicate that exposure to S-nitrosylating agents causes early (as early as 30 minutes) dissociation of SUZ12 from the EZH2-bound PRC2 complex thereby compromising the catalytic activity of the PRC2 complex. This early dissociation of SUZ12 leads to a detectable reduction in H3K27me3 level by 1 hour post exposure to S-nitrosylating agents, while we could not observe any changes in EZH2 level. We believe that during such time EZH2 engages in shuttling to cytosol (via a yet unknown mechanism) before undergoing

autophagosome-lysosome dependent degradation 2 hours post exposure to S-nitrosylating agents. Hence, based on our data, we envisage following events; (i) S-nitrosylation of EZH2 protein causes dissociation of SUZ12 from the PRC2 complex by 30 minutes; (ii) early dissociation of SUZ12 abrogates EZH2-PRC2 complex catalytic activity, thereby reducing H3K27me3 at the 1 hour mark; (iii) S-nitrosylated EZH2 eventually translocate to the cytosol destined to be degraded through autophagosome-lysosome pathway at the 2 hours mark.

(c) Additionally, the authors should address the inconsistency in the use of internal controls for the western blot analysis for EZH2. Supplementary Figure 1H suggests that EZH2 is primarily located in the nucleus. It would be advisable to consistently use a nuclear control for EZH2 normalization, such as Histone H3, throughout the western blot experiments.

Response: We have used GAPDH for the majority of the blots for normalization while for few we have used histone H3. Because we detected many proteins that localized in both nucleus and cytosol, we have therefore used GAPDH as standard loading control for the normalization of the majority of the blots. Based on the suggestion of the reviewer, in many of the blots, we have now incorporated histone H3 for the blots where EZH2 levels were detected. As we detected several proteins and to keep the consistency, the normalization analysis of the blots was primarily carried out considering GAPDH as the loading control.

2. The authors utilized bradykinin to induce endogenous NO production in order to investigate whether NO generated from the eNOS machinery can elicit similar effects as observed with SNP treatment. While the authors treated HUVECs with different doses of bradykinin, they performed a time-dependent treatment of bradykinin on a different type of endothelial cell source EA.hy926. It would be more appropriate and scientifically rigorous to perform the time-dependent treatment of bradykinin in the same cell type as the one used for the dose-dependent treatment. I recommend that the authors address this concern and provide a justification.

Response: We completely agree with the Referee. To address this point, we have performed new experiments to analyze the level of EZH2 and H3K27me3 in both HUVEC and EA.hy926 cells exposed to different concentrations and exposure times of bradykinin. New data have now been included in Supplementary Figure 2D-F. We decided to retain the HUVEC data as HUVEC is a primary endothelial cell line while EA.hy926 is a well-accepted transformed endothelial cell line. We therefore performed these experiments to ascertain that the effect of bradykinin is consistent between both primary and transformed cell lines of endothelial origin. Moreover, we performed new experiments with another strong eNOS agonist vascular endothelial growth factor (VEGF) to show that such endogenous induction of eNOS machinery by VEGF also causes a reduction in EZH2 and H3K27me3 levels. These new data have now been presented in Supplementary Figure 3 A-C.

3. The authors demonstrate EZH2 and H3K27me3 levels ex vivo in rat aorta exposed to NO using the lysate of rat aortic explants. However, considering that the endothelial cells are not the predominant cell population within the aorta, it is essential to ensure that the observed molecular regulation is specifically attributed to the endothelial cells. Therefore, I recommend that the authors perform an enface staining to visualize EZH2 expression specifically in the endothelial layer.

Response: We truly appreciate the Referee for raising this point. Indeed, Reviewer 3 has also suggested the same. Based on these suggestions, we have standardized the *en face* staining of

the rat aorta to visualize the relative level and distribution of EZH2 and H3K27me3 along with counterstaining of endothelial cells using CD144 antibody. Rat aorta exposed to GSNO were processed using standardized *en face* protocol and stained for either EZH2 or H3K27me3 with a counterstaining of endothelial cells using CD144 antibody. All these new data are now included in Figure 2G-H. As observed through immunoblot analysis using aortic rings, herein we also observed a reduction in EZH2 level as well as cytosolic localization of EZH2 upon GSNO exposure. We also detected a reduction in the level of H3K27me3 in endothelial cells within the rat aorta exposed to GSNO.

4. In the co-immunoprecipitation experiment presented in Figure 1G, it appears that there is a noticeable difference in the expression levels of SUZ12 and EZH2 in the input data set. To better interpret this observation, it would be highly recommended for the authors to provide quantification data to support their findings.

Response: Based on the suggestion of the Reviewer, in the revised manuscript, we have added the quantified data for the total level of other PRC2 proteins (SUZ12, AEBP2, JARID, EED) which has been added in Supplementary Figure 4A. We also analyzed the co-immunoprecipitation blots and added the quantified data showing dissociation of SUZ12 alone from EZH2 bound PRC2 complex post 30 minutes' exposure to SNP in Figure 1D.

5. It appears that the majority of EZH2 is consistently localized in the nucleus rather than the cytosol, both in the absence and presence of SNP treatment, as indicated in Figure 1H. This finding is in conflict with the author's claim of cytosolic translocation of EZH2 in EC upon NO exposure.

Response: We do agree with the reviewer that our confocal imaging data indicated that majority of EZH2 primarily localized within the nucleus upon SNP exposure. Although we agree that EZH2 is primarily localized in the nucleus, however, we did observe significant localization of EZH2 in cytosol upon SNP treatment while we were unable to detect any EZH2 in the cytosol of non-treated endothelial cells using confocal microscopy. Moreover, because the nucleus is a more compact structure compared to cytosol in the cell type under study, due to such a compact nature, the relative abundance of a protein like EZH2 will primarily be visualized as more in the nucleus through microscopy techniques. Nonetheless, to address this point raised by the Referee, we now performed several new experiments and analyses that strongly indicated cytosolic localization of EZH2 upon exposure to an S-nitrosylating agent. These experiments and analyses are discussed in the points below;

(i) In the revised manuscript, we performed new experiments to localize EZH2 in cells treated with GSNO using both confocal microscopy and cell fractionation followed by immunoblot analysis approach. In so doing, we detected a much more robust cytosolic localization of EZH2 upon GSNO exposure in endothelial cells. These new images and quantified data of immunoblots are now included in the revised manuscript in Figure 2D,E.

(ii) We have now performed immuno-localization and cell-fractionation coupled with immunoblot analysis using endothelial cells pre-treated with MG132 and bafilomycin followed by SNP/GSNO exposure for 2 hours. We envisaged that blocking the degradation of S-nitrosylated EZH2 through MG132 and bafilomycin treatment shall cause accumulation of the S-nitrosylated EZH2 in the cytosol. Through such analysis, we indeed observed that blocking proteasomal and autophagosome-lysosomal degradation in SNP/GSNO-treated EC causes a significant accumulation of EZH2 in the cytosol compared to only the SNP/GSNO-treated group. Thus, this set of data strongly indicate cytosolic translocation of EZH2 upon SNP/GSNO exposure which

occurs likely before its degradation primarily through the autophagosome-lysosome mechanism. All these new data are now incorporated in the revised manuscript in Supplementary Figure 9.

(iii) We have now included densitometry-based analysis of previously presented immunoblots from four different biological replicates in Figure 1E which again confirm the time-dependent cytosolic localization of EZH2 upon SNP exposure.

6. In this study, the authors aim to address concerns regarding the turnover of H3K27me3, which is influenced not only by the methyltransferase EZH2 but also by the H3K27me3-specific demethylases UTX and JMJD. However, I have some concerns regarding the quality of the western blot analysis, particularly for the expression of JMJD and its internal control. It remains unclear whether the expression of JMJD is altered by SNP treatment. This is a critical issue since the central point of this study is the regulation of H3K27me3 after NO exposure through EZH2.

Response: We completely concur with the Reviewer's point and therefore we performed a new set of experiments to detect the level of UTX and JMJD3 in SNP and GSNO-treated endothelial cells using the immunoblot technique. Densitometry analysis of such immunoblots suggested no alteration in the level of H3K27me3 targeting demethylases, UTX, and JMJD3 in EC exposed to either SNP or GSNO. This new data has now been added in Supplementary Figure 4B-D.

7. In several figures, such as Figure 2E, Figure 6A, B, and D, the authors primarily detected and reported the expression of the HA tag rather than EZH2 itself, in the cells transfected plasmid containing HA tagged EZH2. While the use of an HA tag can be a useful tool for tracking and studying protein expression, it is essential to directly demonstrate the expression of the protein of EZH2. Simply detecting the HA tag does not provide conclusive evidence of EZH2 expression or its functional implications.

Response: To check the overexpression of HA-tagged EZH2, we detected HA instead of EZH2 itself to avoid interference by the endogenous EZH2 present in these cell types. Indeed, we also presented the EZH2 data in some of the previous experiments reported in the earlier form of the manuscript in Supplementary Figure 1E (lower panel) and Supplementary Figure 3E (lower panel) in the revised manuscript. However, to address the reviewer's point, we have re-probed the blots presented in Figures 6 B, and D with EZH2-specific antibodies. These new blots also concur with the findings reported with the HA blot. Moreover, in many of the new experiments performed with the double mutant C329S C700S is now processed using the EZH2 antibody, eg. immunoprecipitation experiment carried out with the C329S C700S mutant is performed using pull down of EZH2 (Figure 6I) and cell fractionation experiment performed using either WT (Supplementary Fig. 5A) or C329S C700S mutant (Figure 6E) are also carried out using EZH2 antibody.

8. In this study, the authors claim that EZH2 undergoes significant ubiquitination upon NO exposure in endothelial cells. However, upon careful examination of the blotting image provided, it appears that the level of EZH2 in the cells remains unchanged after 30 minutes of NO exposure. It would be highly recommended for the authors to provide quantification results for this co-immunoprecipitation experiment.

Response: As suggested by the Referee, we have performed a densitometry analysis of the existing blots and added the quantified data within Figure 4A of the revised manuscript. Such data indicate a significant increase in ubiquitination of the EZH2 protein. In addition, to clearly understand the dynamics of EZH2 degradation, we have performed a cycloheximide chase

experiment. Because cycloheximide blocks the translation of any new proteins, therefore, such an assay allows the evaluation of any specific protein degradation kinetics. In so doing, we observed that 0.5-hour exposure to GSNO also caused a 26% reduction in EZH2 protein level while GSNO treatment for 1 hour reduced EZH2 protein level by 36% when the new translation of proteins including EZH2 was blocked through cycloheximide (Supplementary Figure 5B). These new data obtained from the cycloheximide chase experiment indicate an early degradation of EZH2 (as early as 0.5 hours) when newly synthesized EZH2 protein contributed through translation has been attenuated. Thus, as shown in the manuscript, early ubiquitination upon S-nitrosylation of EZH2 protein still causes induction of EZH2 protein degradation which only could be detected in a later time-point likely due to quick turnover of the EZH2 protein through translational machinery.

9. The authors employed GSK-J4, a selective inhibitor of the H3K27me3-specific demethylases JMJD3 and UTX, to investigate the effects on endothelial cell migration and gene expression changes, as shown in Figure 5. However, I have concerns regarding the direct relevance of using GSK-J4 to inhibit H3K27me3 in reflecting the outcome of inhibiting EZH2 catalytic activity. The presented data does not effectively support the idea that inhibiting EZH2 catalytic activity affects endothelial cell function through the modulation of H3K27me3. I recommend that the authors revise their experimental design and provide additional evidence to establish a more direct connection between inhibiting EZH2 catalytic activity and the functional outcomes observed in endothelial cells. This can be achieved by employing a more specific and direct approach to inhibit EZH2, such as using EZH2-specific inhibitors or genetic approaches targeting EZH2 itself.

Response: We agree with the Referee's point. We performed this experiment to address whether S-nitrosylation-mediated post-translational modification of EZH2 regulates endothelial cell function via its role as catalyst of histone H3 methylation or via its other functions (its role as cell cytoskeleton remodeling through Talin¹, small GTPase^{2,3} or its role as transcriptional activator^{4,5,6}). Because the turnover of H3K27me3 is dependent on both the methyltransferase activity of EZH2 as well the demethylase activity of JMJD3 and/or UTX, we, therefore, took a retrograde approach wherein we preserved the level of H3K27me3 in SNP-treated endothelial cells by blocking the demethylases using GSK-J4 pre-treatment followed by evaluating endothelial gene expression changes and cell migration. Our data indicated that EZH2-dependent epigenetic regulation of gene expression via H3K27me3 played a pivotal role in defining endothelial cell function. Moreover, new ChIP-qPCR analysis also confirmed the enrichment of H3K27me3 on gene promoters of VEGFa, TBX20, MMP2, FGF2, and Angiopoietin-2 which diminished upon SNP exposure. Data associated with the ChIP-qPCR based analysis were incorporated in Figure 5D of the revised manuscript.

Furthermore, as suggested by the Referee, we now have performed the gene expression changes and cell migration analysis in endothelial cells treated with EZH2-specific small molecule inhibitor GSK-126 or transfected with EZH2-specific siRNA. Through these new experiments, we observed comparable changes in gene expression of VEGFa, TBX20, MMP2, FGF2, and Angiopoietin-2 upon either GSK-126 exposure or knockdown of EZH2 through EZH2-specific siRNA. In addition, as reported earlier in endothelial cells⁷, inhibition of EZH2 by GSK-126 or EZH2 siRNA caused enhanced endothelial cell migration. All these data have now been included in Supplementary Figure 10 of the revised manuscript.

10. The authors aimed to investigate the molecular regulation of S-nitrosylation of EZH2 in the regulation of endothelial functions in this manuscript. However, I have concerns regarding the choice of cell line used for the molecular and functional studies presented

in Figure 2E and the whole Figure 6. Instead of using the endothelial cell EA.hy 926, the authors utilized the Human Embryonic Kidney (HEK)-293 cell line. The authors should discuss the rationale behind their choice of HEK-293 cells instead of endothelial cells and address any potential limitations or differences that may arise from using a non-endothelial cell line. And I recommend that the authors perform the molecular and functional studies using the EA.hy 926 cell to validate and strengthen the relevance of their findings to endothelial cells.

Response: We completely agree with the Referee's point. Because we created the mutated EZH2 construct within a plasmid vector, we, therefore, utilized the HEK-293 cell system as they are easy to transfect and over-express the desired forms of the WT and mutated EZH2. Endothelial cell lines such as HUVEC and EA.hy926 cells are more resistant to transfection with plasmid constructs. Moreover, in the current study, we also reported that the effect of nitric oxide on EZH2 protein and its function is not limited to endothelial cells alone as shown via a similar effect of SNP on EZH2 protein and catalytic product H3K27me3 in HEK-293 cells as like endothelial cells (Supplementary Figure 3D). We believe such observation in our study opens new avenues to explore the role of nitric oxide produced through other cellular machineries such as nNOS, iNOS, nitrite reductase in different cell types which could regulate EZH2 stability and activity and may play important roles in different physiological and/or pathological conditions within different tissues/organs.

However, to address the point raised by the Referee, we performed an overexpression experiment with the double mutant of EZH2 C329S C700S in EA.hy926 cells and analyzed its effect on the localization of the HA-tagged mutant in response to GSNO. As observed with HEK-293 cells, GSNO exposure was unable to cause cytosolic translocation of EZH2 C329S C700S mutant (Figure 6E) while EZH2 WT showed significant cytosolic translocation upon GSNO treatment (Supplementary Figure 5A). In addition, we also performed immunoblot analysis to evaluate the effect of GSNO on EZH2 WT or EZH2 C329S C700S mutant in EA.hy926 cells. In parallel to the findings in HEK-293 cells, GSNO failed to diminish the level of EZH2 and associated H3K27me3 in EA.hy926 cells overexpressing EZH2 C329S C700S mutant (Figure 6G-H).

11. The authors try to demonstrate a novel molecular regulation of NO-mediated S-nitrosylation of EZH2 in modulating H3K27me3 levels in endothelial cells in the manuscript. However, I would like to raise a few concerns regarding the rationale and implications of this study. Specifically, it is important to consider the relevance of these findings in vivo and the potential implications for physiological or pathological processes. Additionally, further experiments demonstrating the therapeutic potential of targeting this molecular regulation to improve endothelial dysfunction in disease models would greatly enhance the significance of the study.

Response: We like to thank the Referee for raising this important point. During our initial submission of the manuscript, we indeed wanted to know the physiological/pathological relevance of nitric oxide-dependent regulation of EZH2. We therefore performed a new set of experiments to understand the pathological relevance of such regulation of EZH2 and its associated epigenetic pathways. Our group previously reported activation of EZH2-H3K27me3 axis upon hyperglycemia leading to endothelial inflammation through deactivation of KLF2 thereby suppressing the expression of eNOS and upregulating the expression of inflammatory adhesion molecule ICAM1⁸. In the same study, we reported that suppression of eNOS upon hyperglycemia was one of the key drivers of endothelial inflammation, however, we did not evaluate whether such downregulation of eNOS could in turn regulate EZH2 stability and function. Indeed, previous

findings reported compromised eNOS activation and limited nitric oxide production were shown to be responsible for diabetic vascular dysfunction contributing to increased stroke size⁹.

We, therefore, assessed the effect of GSNO in reversing the hyperglycemia-dependent inflammatory switch of endothelial cells in cultured endothelial cells and rat aorta exposed to intermittent high glucose conditions. Hyperglycemia challenge increased the level of EZH2 and H3K27me3 in both cultured endothelial cells and rat aorta which was reversed upon co-administration of GSNO (Figure 5E,F,H,I). Moreover, GSNO treatment also normalized the levels of hyperglycemia-induced expression of inflammatory adhesion molecule ICAM1 in both cultured endothelial cells and rat aorta (Figure 5G,J). Because immune cells including monocytes adhere to EC expressing adhesion molecules such as ICAM1, we therefore assessed monocyte adhesion in EC exposed to hyperglycemia in the absence and presence of GSNO using the protocol tested previously by our group¹⁰. Hyperglycemia which leads to the expression of ICAM1 by endothelial cells promoted monocyte adhesion to endothelial bed while GSNO treatment abrogated hyperglycemia-induced attachment of monocytes to EC (Figure 5K). Therefore, these new data employing *in vitro* and *ex vivo* models of hyperglycemia advocate the possibility of S-nitrosylation-mediated regulation of EZH2 as a potential therapeutic strategy to block onset and progression of diabetic vascular complications. However, we also like to accept that these observations are purely in *in vitro* and *ex vivo* models, and future experiments in preclinical models are necessary to confirm the pathological relevance of the current findings. These new data, associated inferences, and any limitations are now discussed in the revised Result and Discussion sections of the manuscript.

We also performed a thorough literature review in this area and found that S-nitrosylation via endogenous nitric oxide machinery including endogenous GSNO has protective role in the cardiovascular system. For example, S-nitrosylation is a NO-based signaling mechanism regulating endothelial protein trafficking and suppression of nuclear factor κ B (NF κ B)-dependent expression of proinflammatory cytokines and adhesion molecules¹¹. In addition, S-nitrosylation of N-ethylmaleimide sensitive factor suppresses exocytosis of granules (i.e. Weibel–Palade bodies) and thereby externalization of the adhesion molecule P-selectin which inhibits leukocyte rolling and thus vascular inflammation¹². A similar mechanism is operative in platelets, reducing activation, adhesion, aggregation, and thrombosis¹³. Studies have demonstrated inhibitory S-nitrosylation of both NF κ B and its activating enzyme complex, inhibitory κ B kinase¹⁴. Enzyme such as GSNO reductase (GSNOR) selectively metabolizes endogenous GSNO thereby depleting the pool of GSNO it shifts the equilibrium and consequently limiting the levels of S-nitrosylated proteins. A knockout mouse (GSNOR^{-/-}) has been generated and manifests increased levels of S-nitrosylated proteins¹⁵. Moreover, GSNOR^{-/-} mice exhibit decreased cardiac infarct size and remodeling and increased cardiac function and survival after myocardial infarction¹⁶. Thus, S-nitrosylation mediated anti-inflammatory actions of nitric oxide are relevant to a wide range of cardiovascular disease processes, including atherosclerosis, sepsis, and autoimmune disorders. However, because S-nitrosylation of EZH2 has never been reported, whether such cardiovascular protective effects of nitric oxide or GSNO are partly driven by the regulation of EZH2 through S-nitrosylation need to be comprehensively established through future pre-clinical and clinical studies. Herein, our new data hint in that direction by showing that S-nitrosylation of EZH2 has a vascular protective role at least in diabetes settings.

Response to Reviewer #2 Comments

Dear Authors, in your paper it is described, for the first time, NO-dependent S-nitrosylation of the PRC2 component EZH2 in ECs and consequent effects such as detachment from PRC2 member SUZ12, decreased methyltransferase activity and nuclear-cytoplasmic shuttling. S-nitrosylation of EZH2 was not yet reported, therefore this finding is novel, while following data already reported for other chromatin regulators.

The flow of the experiments shown in this paper is clear. Some data are convincing, other are not. Therefore, this reviewer has some suggestions to improve the MS, as follows.

Response: We like to thank the Reviewer for a positive review of our manuscript and for providing insightful comments that helped us to improve the manuscript.

Overall comment: experiments have been performed in part by using SNP in part GSNO. This reviewer suggests the use GSNO as the only exogenous NO source. SNP is highly toxic, especially at the 500 μ M concentration which rapidly raise the level of RNS within the cell, leading to DNA damage.

Response: We agree with the Referee and therefore performed many additional experiments with GSNO. In the majority of the new experiments performed, in place of SNP, we have used GSNO as the S-nitrosylating agent (Figures 2A-H, 3F-I, 4F-G, 5E-K, 6E-I, Supplementary Figures 4D, 5A-B, 6A-C, 7, 8A-B, 9B-C, 10A). However, because in our original work we have performed extensive experiments with SNP and as in our experimental conditions endothelial cells exposed to SNP showed healthy cellular conditions as shown by no changes in cell viability (MTT assay, Supplementary Figure 1C), cellular apoptosis (Annexin V-PI flow cytometry, Supplementary Figure 1D), and DNA damage associated protein γ H2AX level (Supplementary Figure 1B), we, therefore, decided to keep the data obtained in cells/tissues exposed to SNP. Moreover, the detection of nitrite levels using Griess Assay indicated a comparable increase in cellular nitrite levels in both SNP- and GSNO-treated endothelial cells (Supplementary Figure 1A), thus hinting at comparable nitrosative stress in both SNP and GSNO.

Results section: “Nitric oxide exposure interplayed with EZH2 including PRC2 assembly, its methyltransferase activity, subcellular localization, and stability”

Major:

As discussed above, this reviewer believes 500 μ M SNP to raise too high and toxic concentration of NO. It is known that SNP induces a burst of NO (and consequent nitrosative stress). Therefore: a) in authors experimental conditions, which is the level of γ H2AX?

Response: As suggested by the Referee, we performed immunoblot assays to detect the level of γ H2AX. Such analysis indicated none to a modest reduction in γ H2AX level after 4 hours of SNP exposure. This new data has now been added in Supplementary Figure 1B of the revised manuscript. In addition, we also analyzed cell viability and cellular apoptosis using MTT assay and Annexin V-PI staining followed by flow cytometry analysis respectively. Modest to no detectable changes were observed in cell viability or apoptosis of endothelial cells exposed to SNP (Supplementary Figure 1C,D).

This reviewer would like to evaluate the health state of the cells; b) which are the levels of nitrites/nitrates/peroxynitrites?

Response: We analyzed the nitrosative stress in endothelial cells exposed to SNP and GSNO through Griess assay. We have detected a significant increase in the cellular nitrite level in cells exposed to SNP or GSNO (Supplementary Figure 1A).

Figure 1: panels B, D and E are not PQ. Please, provide better wb images.

Response: As suggested, we have replaced these panels with better-quality representative images.

Panel G: this reviewer has some concerns on this IP. Starting from the right panel, the authors show the input levels of PRC2 protein at 30' of NO treatment. It seems that SUZ12, EED (please, provide a better wb image for EED) and EZH2 decrease in the presence of NO (levels of GAPDH are highly similar in ctr vs treated cells). However, the authors says that at 30' the only detected a diminished H3K27me3 (as shown in the upper panel of this figure). Further, looking at the left panel showing IP results, it seems that also auto-IP levels of EZH2 are decreased in the presence of NO, raising the hypothesis that impairment of SUZ12 recruitment could indeed depend on decreased EZH2 IP levels. Further, the results shown for SUZ12 seem to be related more to technical issues than to a real reduction at the protein level. Please provide a better wb image for IP-SUZ12.

Response: Based on the suggestions of the Referee, we have changed the immunoblot images from another biological replicate of the immunoprecipitation experiment and revised the entire Figure 1D with new immunoblot images. In addition, we have now performed a comprehensive analysis of the immunoprecipitation experiments to show the association of all other PRC2-associated proteins with EZH2. This new data has also been included in the revised Figure 1D. Moreover, we have included a densitometry analysis graph in the revised Supplementary Figure 4A to address the changes in other PRC2 complex proteins upon SNP exposure. All these revised figures and analyses indicate early loss of association of SUZ12 with EZH2 upon SNP exposure without altering the association of other PRC2 proteins. Furthermore, the total protein level of other PRC2-associated proteins remained unaltered upon SNP exposure.

The authors should also show densitometric analyses of blots in panel G (both left and right), normalizing the levels of IP PRC2 components on IP EZH2 content in ctr vs treated cells. This should clarify the data.

Response: As suggested by the Referee and as described in the previous response, we have now included densitometric analyses of blots for immunoprecipitation experiment in the revised Figure 1D, 2F, 3D.

Line 145: the authors cannot use the word “degradation” at this stage, They have still not proven EZH2 to be degraded by NO (i.e. EZH2 decrease could be dependent on NO-induced translation inhibition by indirect mechanisms such as miRNA induction, for example).

Response: We thank the Referee for bringing up this important point and we do agree with the Referee that regulation in the EZH2 protein level can be driven by NO-mediated translational inhibition of EZH2 transcript. Therefore, we performed a cycloheximide chase experiment to understand the dynamics of reduction in EZH2 protein level upon GSNO exposure. Cultured endothelial cells were pre-treated with cycloheximide to block the translational machinery (to block the synthesis of any new copies of proteins) followed by exposing the cells to GSNO for different time points (0.5 and 1 hour) and analyzing the level of EZH2 through immunoblot experiment. In

so doing, we observed that 0.5 hour exposure to GSNO also caused 26% reduction in EZH2 protein level while GSNO treatment for 1 hour reduced EZH2 protein level by 36% when new translation of proteins including EZH2 are blocked through cycloheximide. As shown in Supplementary Figure 5B, herein, we could not detect any alteration in EZH2 protein level when cells were exposed to only GSNO for 0.5 or 1 hour indicating translation-dependent replenishment of EZH2 protein during the early time points upon GSNO treatment.

More importantly, This new set of data obtained from the cycloheximide chase experiment indicate that removing the translation-dependent regulation of EZH2 upon GSNO exposure causes early reduction in EZH2 protein level suggesting that EZH2 protein undergoes degradation upon GSNO/SNP exposure. Thus, as shown in the manuscript, early ubiquitination upon S-nitrosylation of EZH2 protein still causes induction of EZH2 protein degradation which only could be detected in a later time-point likely due to quick turnover of the EZH2 protein through translational machinery.

Line 172: again, the authors have not specifically demonstrated a NO-impact on EZH2 methyltransferase activity. This is an indirect statement as they showed SUZ12 detachment (to be confirmed) from EZH2. Further, since the authors show in panel H EZH2 nuclear-cytoplasmic shuttling, the reduced levels of H3K27me3 may rely also on this phenomenon. The authors should show a specific methyltransferase assay on recombinant H3K27 to verify this assumption.

Response: As suggested by the Referee, we have now performed the methyltransferase activity assay using EpiQuik Histone Methyltransferase Activity/Inhibition Assay Kit (H3K27). We performed this assay using nuclear extract of endothelial cells which were then exposed to GSNO or SNP for 1 hour. Non-treated nuclear extract of the cells was considered as a control. This evaluation indicated a reduction in the methyltransferase activity of the endothelial nuclear extract treated with either GSNO or SNP. This new data has been included as Figure 3H-I in the revised manuscript.

Panel H: according to the wb on fractionated cell extracts, the cytosolic translocation of EZH2 seems to be transient and limited at the 30' time point. If we normalize the data on GAPDH levels, at 60' cytosolic EZH2 returns at ctr level. Authors should show densitometric analysis of this experiment. According to this observation, confocal analyses should be performed at the 30' time point.

Response: As suggested by the Referee, we have performed the densitometry analysis of the blots presented for cell fractionation and such analysis is added in Figure 1E, 2D, 6E, Supplementary Figures 5A, 6A-C, 9A-B in the revised manuscript. EZH2 significantly translocate to the cytosol at both 30 minutes and 60 minutes post SNP/GSNO exposure. Moreover, we have now performed new assays using different experimental models to assess the effect of S-nitrosylating agents on EZH2 translocation. These experimental details and observations are discussed below;

(i) We have now performed the EZH2 localization study in endothelial cells treated with GSNO. Both cellular fractionation followed by immunoblot assay and immunofluorescence followed by confocal microscopy indicated cytosolic translocation of EZH2 in endothelial cells upon GSNO exposure. These data are incorporated in revised manuscript as Figure 2D,E.

(ii) Through new experiment, we also analyzed the localization of EZH2 in WT EZH2-HA overexpressing HEK293 cells which underwent GSNO treatment. Cell fractionation followed by

immunoblot analysis substantiates our findings that GSNO causes cytosolic localisation of the overexpressed EZH2 (Supplementary Figure 5A). Moreover, we also performed immunofluorescence followed by confocal microscopy of WT EZH2-HA overexpressing EA.hy926 endothelial cells exposed to GSNO which also confirmed cytosolic translocation of WT EZH2-HA in endothelial cells upon such treatment (Figure 6F).

(iii) As suggested by Reviewer 1, we have now performed immuno-localization and cell-fractionation coupled with immunoblot analysis using endothelial cells pre-treated with MG132 and bafilomycin followed by SNP/GSNO exposure for 2 hours. We envisaged that blocking the degradation of S-nitrosylated EZH2 through MG132 and bafilomycin treatment shall cause accumulation of the S-nitrosylated EZH2 in cytosol. Through such analysis, we indeed observed that blocking proteasomal and autophagosome-lysosomal degradation in SNP/GSNO treated EC causes significant accumulation of EZH2 in the cytosol compared to only SNP/GSNO treated group. Thus, this set of data strongly indicates cytosolic translocation of EZH2 upon SNP/GSNO exposure occurs likely before its degradation primarily through autophagosome-lysosome mechanism. All these new data have now been incorporated in the revised manuscript in Supplementary Figure 9.

Supplementary figure 1B: time of treatment is not indicated.

Response: Details of time of the treatment time points have now been incorporated within the legends of the respective figure.

Supplementary figure 2C: quality of EED, AEBP2 and histone H3 is too poor. This reviewer understand it is a supplemental data, but the quality has to be a little bit enhanced.

Response: These blots are now replaced with better-quality blots. Moreover, we have also included the densitometric analysis of the blots (Supplementary Figure 4A).

Minor:

Line 122: this is not completely true. An interplay between EZH2 and eNOS has been already reported (Mitic T, Mol Ther, 2015).

Response: We agree with the reviewer, however, in the given work, authors reported how EZH2 could epigenetically regulate the expression of eNOS. Herein, we aim to address how eNOS dependent nitric oxide production in turn regulate EZH2 through S-nitrosylation. Nonetheless, as suggested by the Referee, we have now rephrased such specific statements as indicated and in addition incorporated a short discussion about EZH2-dependent regulation of eNOS along with referring to the work of Mitic et al., 2015 on Page 18, line 637-643 in the revised manuscript with track changes.

Results section: “Nitric oxide exposure caused S-Nitrosylation of EZH2 leading to early SUZ12 dissociation and further altering its binding partners”

Major:

Figure 2D, upper panel: whereas data on EZH2 S-nitrosylation are convincing, this reviewer believes that data on SUZ12 are not, especially when comparing original wb data (4th page of the related MS file). SUZ12 blot in the MS is too modified.

Response: As suggested by the referee, we have now readjusted the blot accordingly. We have equally increased the brightness (increase brightness by 50 scale for the entire blot without

altering the contrast of the original blot) of the entire blot to allow visualization of the difference in SUZ12's association with EZH2. Because we have used Biorad ChemiDoc system, even a shorter exposure sometime yields relatively dark bands. We therefore adjusted the background of the entire blot to visualize the difference. I hope such limited adjustment of the blots for representation will be acceptable for the Referee. Moreover, we now have analyzed the immunoblots for the given immunoprecipitation experiments with specific experimental conditions which revealed a significant reduction in SUZ12 association with EZH2 upon induction of cultured endothelial cells with bradykinin. Furthermore, in parallel to these observations, our new data also showed that GSNO exposure caused early dissociation of SUZ12 from EZH2. Therefore, all together, we showed that exposure to either SNP or bradykinin or GSNO caused early dissociation of SUZ12 from EZH2.

Lower panel: the input levels of SUZ12 are decreased upon NO exposure. This is particularly evident in original wb data (page 5 of related MS file). In this regard, this reviewer has to say that original wb and blot shown in the MS do not correspond. This also supports this reviewer's doubt on SUZ12 detachment in the upper EZH2 IP.

Response: Based on the suggestion, we now have presented the original blot without any modification. We have not observed any changes in the level of SUZ12 nor in this blot or in the data presented in Figure 3D where we have analyzed the level of SUZ12 upon SNP exposure.

Figure 3B, please provide higher quality histone H3 wb data.

Response: We have now represented a new set of blot for this Figure.

The authors should show SUZ12 detachment by IP experiments as shown in figure2.

Response: As suggested by the Reviewer, we performed new immunoprecipitation experiment using endothelial cells exposed to GSNO followed by evaluating the association of EZH2 with SUZ12. Through such experiments and analysis, we confirmed early (within 30 minutes of GSNO exposure) dissociation SUZ12 from EZH2 protein (Figure 2F).

In MS experiment, do the author detect SUZ12 among the 48 EZH2 interacting proteins which disassociate upon NO exposure? Maybe this reviewer missed the point, but SUZ12 does not appear in the list of associated protein provided in supplementary table 1. This is a very important point. Regarding 14-3-3 association, it would be interesting to validate this association in IP experiments

Response: We like to thank the Reviewer for raising this very pertinent question. Our MS data showed the interaction of SUZ12 with EZH2 in both control and GSNO-treated groups. As a matter of fact, all other components of the PCR2 complex including EED, AEBP2, and JARID2 were also associated with EZH2 in both control and GSNO sets. Therefore, in the MS data, SUZ12 did not appear in the list of unique proteins to be associated with EZH2 only in the control group. We indeed envisaged the same as SNP/GSNO/bradykinin did not cause a complete loss of SUZ12 association with EZH2 (as observed in Figure 1D, 2F, 3D). Moreover, we also performed a new set of MS analysis experiments to validate these findings. However, as the current Reviewer has highlighted this point, we therefore carefully analyzed another parameter obtained through the MS evaluation wherein the relative abundance of each of the EZH2-interacting proteins was provided. We normalized the abundance level of each of the PRC2 complex proteins including SUZ12, EED, AEBP2, and JARID2 to that of the EZH2 protein to obtain an EZH2 normalized ratio. In so doing, we observed an interesting pattern. SUZ12 association was reduced by 30%

upon GSNO exposure as also observed through our co-immunoprecipitation studies (as observed in Figure 1D, 2F, 3D). Furthermore, in the MS data, EED and AEBP2 association with EZH2 remained unchanged (Figure given below) as also observed through co-immunoprecipitation studies in SNP-treated EC (Figure 1D). Surprisingly, in the MS data, we also detected a diminished association of JARDI2 with EZH2 upon GSNO challenge (33% reduction, Figure given below) which was not observed in our immunoprecipitation followed by immunoblot analysis at least in SNP-treated groups (Figure 1D). We have included this Figure as part of the Response to the Reviewer's Comments and not incorporated this data as a Figure into the main manuscript file. Nonetheless, if the Reviewer suggests, we shall be happy to incorporate this data as part of the Figures in the revised manuscript file.

Results section: “Nitric oxide caused the degradation of EZH2 primarily through autophagosome-lysosome 223 pathway while inhibition of endogenous nitric oxide machinery reversed nitric oxide dependent degradation, activity and localization of EZH2”

Major:

In figure 4E, a panel showing a western blot for EZH2 on fractionated cell extracts should be added

Response: As proposed by the Reviewer, we now performed a new set of cell fractionation and immunofluorescence followed by confocal microscopy experiments to analyze the relative distribution of EZH2 in SNP or GSNO-treated endothelial cells when these cells were pretreated with MG132+Baflomycin. As can be observed in Supplementary Figure 9A-C, we see robust cytosolic accumulation of EZH2 protein in cells pretreated with MG132+Baflomycin followed by either SNP or GSNO challenge. Such observation suggests that upon SNP/GSNO exposure EZH2 translocate to cytosol followed by degradation through the autophagosome-lysosome/proteasome machinery while abrogating these degradation machineries accumulates EZH2 in the cytosol due to the inability to undergo degradation.

Minor: line 233: this reviewer would not use the term “heavily” ubiquitinated. If we compare the level of Ub-EZH2 with total EZH2 in figure 4A, the difference between ctr and NO-treated Ub-EZH2 decreases, also looking at the original wb data shown at page 8 of related MS file.

Response: We concur with the Referee's point and have changed this statement. However, as we performed densitometric analysis of all the replicates, we did observe a nearly threefold increase in ubiquitination which was statistically significant. We thus believe that the ubiquitination of EZH2 protein occurs after exposure to SNP which likely drives its degradation primarily through the autophagosome-lysosome pathway. We have now added the densitometric data analysis bar graph in Figure 4A of the revised manuscript.

Results section: "Protecting the level of EZH2 downstream product H3K27me3 through inhibition of demethylases reversed nitric oxide dependent effect on endothelial migration and gene expression changes"

Major

Real time experiments in figure 5C should be flanked by ChIPs to show H3K27me3 enrichment at promoters of the investigated genes

Response: We again thank the Referee for bringing this important point. As proposed by the Referee, we have now performed ChIP assay using CUT&RUN Protein-DNA Interaction Assay Kit. Through these new experiments, we detected loss of H3K27me3 enrichment on the gene promoters of FGF2, TBX20, Angiopoietin-2, VEGFa, and MMP2 when endothelial cells were exposed to GSNO (Figure 5D). Therefore, GSNO-driven reduction in EZH2 and its catalytic product H3K27me3 causes loss of enrichment of repressive H3K27me3 mark in the promoters of the specific genes thereby enhancing their transcription.

Results section: "S-nitrosylation of EZH2 protein at C329 and C700 causes conformational changes in EZH2-SUZ12 complex leading to loose association of SUZ12 with the SAL domain of EZH2"

Major: This section could be strengthened by IPs performed on protein extracts from EZH2 C329S/C700S double mutant expressing cells, cultured in the presence or absence of NO, and subsequent wb revealing SUZ12. (These experiments may be also shown in figure 6, as the authors prefer).

Response: Again many thanks to the Referee for suggesting this experiment. We have now performed the recommended immunoprecipitation experiment. As can be observed from the data in Figure 6I, SNP exposure could not alter the binding capacity of SUZ12 to C329S C700S mutant of EZH2. Thus, this data hints at the crucial role of Cysteine 329 and Cysteine 700 in maintaining SUZ12 binding with the EZH2 protein of the PRC2 complex.

Response to Reviewer #3 Comments

In their manuscript entitled 'S-nitrosylation of EZH2 at C329 and C700 interplay with PRC2 complex assembly, 1 methyltransferase activity, and EZH2 stability to regulate endothelial functions', Sakhuja and co-workers describe that s-nitrosylation of EZH2 at cysteine residues 329 and 700 affects its stability and methyltransferase activity, thereby altering the endothelial cell gene transcriptional profile. These results advance our understanding of the mechanisms by which endothelial cell behavior is influenced by epigenetics and NO. The results are of interest to the cardiovascular field, the methodology is technically sound and the presented conclusions appear justified by the results. Yet, some improvements to the manuscript can be made.

Response: Thanks to the Reviewer for the positive feedback on our work. We also thank the Reviewer for the insights and comments to improve the manuscript.

Major comments:

In their experiments, exogenous NO is used to study the downstream effects on EZH2 stability and activity. In a translational perspective, where would the authors suggest that these high levels of NO come from. Would this be reflected as a response to (over-)activation of eNOS and elevated endothelial NO synthesis, or else? Please provide data that the endogenous increase in NO (by increased eNOS activity) also results in the S-nitrosylation of EZH2 and reduction of its transferase activity.

Response: We like to thank the Referee for highlighting this important point. We indeed have used bradykinin, a natural inducer of eNOS to increase endogenous levels of NO. Many of the data presented in the previous version of the manuscript or in the revised manuscript were generated by using bradykinin as a natural inducer, eg. Supplementary Figure 2D-F, Figure 3D, Supplementary Figure 10A. Through such data, we clearly showed that bradykinin causes a reduction in EZH2 level as well as inhibits the level of its catalytic product H3K27me3. Moreover, we also showed that bradykinin causes S-nitrosylation of EZH2 and SUZ12 dissociation in endothelial cells (Figure 3D). Furthermore, to affirm these observations, we performed new experiments using vascular endothelial growth factor (VEGF), another natural inducer of eNOS to elevate endogenous NO levels. In parallel to bradykinin, VEGF also caused a time-dependent reduction in EZH2 protein and H3K27me3 level in endothelial cells suggesting that activation of endogenous NO-producing machinery caused a similar effect as that of external NO sources such as SNP. The new VEGF data are included in the revised manuscript as Supplementary Figure 3A-C.

In rat aorta, the decrease in EZH2 expression after NO stimulation may occur in endothelial cells or vascular smooth muscle cells. Please use an imaging technique to show endothelial cell specificity of this response.

Response: We truly appreciate the Referee for raising this point. Indeed, Reviewer 1 has also suggested the same. Based on these suggestions, we have standardized the en face staining of the rat aorta to visualize the relative level and distribution of EZH2 and H3K27me3 along with counterstaining of endothelial cells using CD144 antibody. Rat aorta exposed to GSNO were processed using standardized en face protocol and stained for either EZH2 or H3K27me3 with a counterstaining of endothelial cells using CD144 antibody. All these new data have been included in Figure 2G,H. As observed through immunoblot analysis using aortic rings, herein we also observed a reduction in EZH2 level as well as cytosolic localization of EZH2 upon GSNO exposure. We also detected a reduction in the level of H3K27me3 in endothelial cells within the rat aorta exposed to GSNO.

Please add replicates and quantify the blots used in all figures (specifically figs. 1G and H, Suppl. Fig 1B-E, fig.2, fig 4A).

Response: As suggested by the Referee, densitometric analysis of all the blots were carried out and the plotted data have been included in the revised manuscript for the respective figures.

Suppl. Figure 3 has smears on the blot and is therefore hard to interpret. Please provide a better representative example.

Response: We have now performed a fresh experiment with recombinant EZH2 exposed to GSNO followed by processing the sample for Iodo-TMT assay. We then performed both dot blot

and SDS-PAGE followed by immunoblot. The older blots were replaced with the new blots demonstrating EZH2 S-nitrosylation while also showing the S-nitrosylation of the recombinant EZH2 upon GSNO exposure using dot blot.

EZH2 immunoblotting in Fig 4 shows 2 bands, whereas other EZH2 immunoblots only show 1 specific band. What causes this difference?

Response: We are not sure if the Reviewer is referring to Figure 4A where we have shown ubiquitination. In this immunoblot, we have loaded an immunoprecipitated sample to confirm EZH2 ubiquitination. Because we have loaded an EZH2 enriched sample and the blots are first probed with ubiquitin antibody followed by stripping and re-probing for EZH2, the blot looks a little spread. However, after careful observation, we did not clearly detect two individual bands in the immunoblot.

The kinetic responses differ between the reduction of H3K27Me3 and the reduction in EZH2. What causes this difference? Is NO also affecting the demethylase activity of other enzymes? Just showing that NO has no effect on the transcription of EZH2 is not sufficient to conclude that only a reduction in EZH2 methyltransferase activity can explain this difference in kinetics.

Response: We like to thank the reviewer for bringing this important point. A similar point was also raised by Reviewer 1. Indeed, this was one of the most intriguing initial observations that changed the way we performed experiments to understand the entire phenomenon. Although we have modestly explained the reasoning behind such observation in Page 4, Line 145 under Result section of the previous manuscript file, we completely agree with the Reviewer that such discrepancy in the observation should have been discussed properly to enhance the clarity and scientific explanation behind. We have now added more detailed explanation in the Result section of the revised manuscript on Page 5, Paragraph 2. To explain in brief, the scientific explanation behind such observation, our data indicates that exposure to S-nitrosylating agents causes early (as early as 30 minutes) dissociation of the SUZ12 from the EZH2 bound PRC2 complex, thereby, compromising the catalytic activity of the PRC2 complex. This early dissociation of SUZ12 leads to a detectable reduction in H3K27me3 level post 1 hour exposure to S-nitrosylating agents by when we could not observe any changes in EZH2 level. We believe that during such time EZH2 engages in shuttling to cytosol (via a yet unknown mechanism) before undergoing autophagosome-lysosome dependent degradation 2 hours post exposure to S-nitrosylating agents. Hence, based on our data, we envisage the following events; (i) S-nitrosylation of EZH2 protein causes dissociation of SUZ12 from the PRC2 complex by 30 minutes; (ii) early dissociation of SUZ12 abrogates EZH2-PRC2 complex catalytic activity and thereby reducing in H3K27me3 at the 1 hour time point; (iii) S-nitrosylated EZH2 eventually translocate to cytosol to be degraded through autophagosome-lysosome pathway at the 2 hours mark.

To address the effect of NO on methyltransferase activity of EZH2, we have now performed the methyltransferase activity assay using EpiQuik Histone Methyltransferase Activity/Inhibition Assay Kit (H3K27). We performed this assay using nuclear extract of endothelial cells which were then exposed to GSNO or SNP for 1 hour. Non-treated nuclear extract from EC was considered as a control. This evaluation indicated a reduction in the methyltransferase activity of the endothelial nuclear extract treated with either GSNO or SNP for 1 hour. These new data conclusively proved that SNP/GSNO exposure limits the methyltransferase activity of EZH2 and the PRC2 complex. This new data has been included as Figure 3H,I in the revised manuscript.

What are the responses to SNP for UTX and JMJD3 expression and do these differ from responses to BK? This is particularly of interest in the context of results presented in fig. 5, where NO is shown to interact with JMJD3 and UTX.

Response: We completely concur with the Reviewer's point and therefore we performed a new set of experiments to detect the level of UTX and JMJD3 in SNP and GSNO-treated endothelial cells using the immunoblot technique. Densitometric analysis of such immunoblots suggested no alteration in the level of H3K27me3 targeting demethylases, UTX, and JMJD3 in EC exposed to either SNP or GSNO. This new data has been added in Supplementary Figure 4 B-D.

Line 212 states that the authors "expected such association ... due to EZH2 cytosolic shuttling"; yet figure 2 does not confirm this statement. Please show more compelling evidence for cytosolic shuttling of EZH2 or revise the statement (similar in lines 214-215).

Response: There may have been some confusion. This line discussed about the interaction of 14-3-3 epsilon with EZH2 protein in non-treated conditions which has been presented in Figure 2K and Supplementary Tables 1-2. This data was acquired by high throughput mass spec-based interactome analysis which indicated the association of EZH2 protein with shuttling protein 14-3-3 epsilon in un-treated endothelial cells which was completely abolished upon GSNO exposure. Therefore, the authors request the Referee to go over the data presented in Figure 2K and Supplementary Tables 1-2.

Moreover, as also suggested by Reviewer 2, we now have performed co-immunoprecipitation experiment using an EZH2 antibody to confirm the loss of association with 14-3-3 upon GSNO exposure to cultured EC. We performed this experiment to validate some of our observations through the MS data. Through such analysis, we confirmed that 14-3-3 associates with EZH2 in un-treated endothelial cells which was significantly diminished after GSNO exposure. Therefore, our data parallels with the observation made through mass spec analysis depicting loss of 14-3-3 association with EZH2 upon GSNO challenge.

Treatment of EA.hy926 cells with MG and Baf completely restores EZH2 expression after NO exposure. Please provide data to show this also holds true for its downstream repressive mark H3K27Me3.

Response: As suggested by the Referee, we have now analyzed the level of H3K27me3 in endothelial cells pre-treated with MG-Baf followed by SNP or GSNO exposure. In a non-SNP/GSNO exposure conditions, we detected a modest yet statistically insignificant increase in H3K27me3 levels in endothelial cells treated with only a combination of MG and Baf (Figure 4E,G). This may be surprising, however, we believe that simply protecting a pool of EZH2 that would have naturally undergone degradation may not enhance the H3K27me3 level as these fractions of degradation-protected EZH2 likely have limited catalytic activity. Therefore, although the protein level of EZH2 may have been protected by MG+Baf treatment, however, such degradation-restricted pool of EZH2 are likely to be ubiquitinated and functionally inactive. Nonetheless, MG-Baf pre-treatment failed to rescue the level of H3K27me3 in endothelial cells challenged with SNP or GSNO (Figure 4E, G). We were initially surprised by this observation, however, based on our findings in this manuscript, we could explain why we observed such failure in restoring the level of H3K27me3 by MG-Baf. Because GSNO/SNP causes catalytic deactivation of EZH2 independent of its degradation, therefore, simply restoring the level of EZH2 by MG-Baf is likely not sufficient for restoring the level of H3K27me3. These non-degraded yet likely S-nitrosylated EZH2 in GSNO/SNP and MG-Baf treated conditions are catalytically inactive and are incapable of restoring the level of its catalytic product H3K27me3.

Please provide confirmative data to evidence show which NO-responsive genes are regulated by EZH2 in endothelial cells to elucidate whether the gene expression changes are derived from an interregulated mechanisms or from two synergistic pathways. ChIP-PCR or ChIP-seq should be performed for VEGFA, KDR, TBX20, TIE2, MMP2, TEK, FGF2 and ANGPT2.

Response: We again thank the Referee for bringing up this important point. As proposed by the Referee, we have now performed ChIP assay using the CUT&RUN Protein-DNA Interaction Assay Kit. Through these new experiments, we detected loss of H3K27me3 enrichment on the gene promoters of FGF2, TBX20, Angiopoietin-2, VEGFa, and MMP2 when endothelial cells were exposed to GSNO (Figure 5D). Therefore, GSNO-driven reduction in EZH2 and its catalytic product H3K27me3 causes loss of enrichment of repressive H3K27me3 mark in the promoters of the specific genes thereby enhancing their transcription.

Please add the potential role of demethylating enzymes to figure 8 to indicate that the reduction in H3K27Me3 ha a different kinetic than the NO-induced degradation of EZH2. The graphics is too biassed now.

Response: As suggested by the Referee, we now comprehensively revised the graphical presentation in Figure 8 to incorporate all the key findings of the study.

There is a high redundancy between the introduction, manuscript body and discussion, whereas the novelty of the current findings are only minimally discussed. I suggest to remove the redundancy and add more discussion on which gap in knowledge the current results solve and how these contribute to a better understanding of epigenetic regulation in endothelial cells. Also add discussion on the interaction between methyltransferases and demethylases and how NO exposure may influence this.

Response: We thank the Reviewer for highlighting the pitfalls in the text of the manuscript. We have now extensively revised the manuscript to primarily highlight the gap in knowledge and the novelty of the study in the context of how nitric oxide signaling dictates the epigenetic landscape of endothelial cells. Significant changes were made in the Introduction and Discussion sections of the manuscript to highlight these points in the revised manuscript.

Minor comments:

Please provides all images of all analyzed blots in supplement. All blots should be unprocessed and uncut / uncropped blots, including size markers on the blots.

Response: We have included all the Raw blots presented in the Figures of the manuscript. We have followed the standard format of *Nature Communications*, wherein all the uncut raw blots which are represented in the Figures are included in the Raw data file.

There are multiple grammar and spelling errors throughout the manuscript. Please have the manuscript checked and corrected by a native English speaker.

Response: All the authors have thoroughly proofread the manuscript once again and any existing grammatical or spelling mistakes have been rectified.

Define HTM at first mention in introduction.

Response: It has now been defined in the revised version of the manuscript.

How were the concentrations of stimulants and inhibitors used determined? If concentrations were used from literature, please cite. If empirically determined, please add these data to the supplement.

Response: All the references justifying the use of specific concentrations of the inhibitors have now been mentioned in the revised manuscript.

References

1. Gunawan, M. *et al.* The methyltransferase Ezh2 controls cell adhesion and migration through direct methylation of the extranuclear regulatory protein talin. *Nat. Immunol.* **16**, 505–516 (2015).
2. Dobenecker, M.-W. *et al.* Signaling function of PRC2 is essential for TCR-driven T cell responses. *J. Exp. Med.* **215**, 1101–1113 (2018).
3. Su, I. -hsin *et al.* Polycomb group protein ezh2 controls actin polymerization and cell signaling. *Cell* **121**, 425–436 (2005).
4. Kim, J. *et al.* Polycomb- and Methylation-Independent Roles of EZH2 as a Transcription Activator. *Cell Rep.* **25**, 2808-2820.e4 (2018).
5. Vanden Bempt, M. *et al.* Aberrant MYCN expression drives oncogenic hijacking of EZH2 as a transcriptional activator in peripheral T-cell lymphoma. *Blood* **140**, 2463–2476 (2022).
6. Jiao, L. *et al.* A partially disordered region connects gene repression and activation functions of EZH2. *Proc. Natl. Acad. Sci. U. S. A.* **117**, 16992–17002 (2020).
7. Mitić, T. *et al.* EZH2 modulates angiogenesis in vitro and in a mouse model of limb ischemia. *Mol. Ther. J. Am. Soc. Gene Ther.* **23**, 32–42 (2015).
8. Thakar, S., Katakia, Y. T., Ramakrishnan, S. K., Pandya Thakkar, N. & Majumder, S. Intermittent High Glucose Elevates Nuclear Localization of EZH2 to Cause H3K27me3-Dependent Repression of KLF2 Leading to Endothelial Inflammation. *Cells* **10**, 2548 (2021).
9. Li, Q. *et al.* Deficient eNOS phosphorylation is a mechanism for diabetic vascular dysfunction contributing to increased stroke size. *Stroke* **44**, 3183–3188 (2013).
10. Katakia, Y. T. *et al.* Angular difference in human coronary artery governs endothelial cell structure and function. *Commun. Biol.* **5**, 1044 (2022).
11. Lowenstein, C. J. Nitric oxide regulation of protein trafficking in the cardiovascular system. *Cardiovasc. Res.* **75**, 240–246 (2007).
12. Matsushita, K. *et al.* Nitric oxide regulates exocytosis by S-nitrosylation of N-ethylmaleimide-sensitive factor. *Cell* **115**, 139–150 (2003).
13. Morrell, C. N. *et al.* Regulation of platelet granule exocytosis by S-nitrosylation. *Proc. Natl. Acad. Sci. U. S. A.* **102**, 3782–3787 (2005).
14. Reynaert, N. L. *et al.* Nitric oxide represses inhibitory kappaB kinase through S-nitrosylation. *Proc. Natl. Acad. Sci. U. S. A.* **101**, 8945–8950 (2004).
15. Liu, L. *et al.* Essential roles of S-nitrosothiols in vascular homeostasis and endotoxic shock. *Cell* **116**, 617–628 (2004).
16. Lima, B. *et al.* Endogenous S-nitrosothiols protect against myocardial injury. *Proc. Natl. Acad. Sci. U. S. A.* **106**, 6297–6302 (2009).

Response to Reviewer #1 Comments

Reviewer #1 (Remarks to the Author):

Thank you for revising your manuscript in response to my previous comments and suggestions. I have carefully reviewed the additional experiments you have conducted to address the concerns raised in my initial feedback. I appreciate the effort you have put into strengthening your study based on the feedback provided. The additional data and analyses have enhanced the robustness of your study and clarified some of the points that were previously raised. I believe that the revisions have significantly strengthened the manuscript.

Response: We would like to thank the Reviewer for appreciating our efforts to revise the manuscript based on their comments.

Response to Reviewer #2 Comments

Reviewer #2 (Remarks to the Author): Review of NCOMMS-24-08539A_R1

Dear Authors, this Reviewer is impressed by the amount of work you did and congratulate with all the lab staff and collaborators. There are still some minor points to be clarified to make your paper faultless.

Response: We would like to thank the Reviewer for appreciating our efforts to revise the manuscript based on their comments and also like to thank for providing a few additional points to be addressed to improve the paper further.

1. Fig. S1B. In this reviewers' hand, high NO concentrations result in an increase in γ H2AX levels in a wide variety of cells, indicating DSBs on DNA. A lot of literature is in line with this statement. It is curious that Ea.hy926 cells display such a high baseline level of γ H2AX which decreases at the 4hrs time point of NO treatment. Maybe, samples have to be inverted? Is it conceivable that the last line corresponds to the negative ctr (no SNP)? Please, check the correct order of the samples, or give an explanation for the high DSBs quantity present in these cells. Further, this alternative interpretation of the data (an increase in γ H2AX levels upon NO exposure) would also correlate with the increase in nitrites concentration shown in fig, S1A. In this regard, this reviewer would like to see a time course of nitrites accumulation similar to the γ H2AX WB to correlate the data.

Response: We completely agree with the Referee's point. However, as we carefully went through all the existing data and performed another new set of experiments to be ascertain further we found that these transformed endothelial cell line, EA.hy926 indeed have high baseline of γ H2AX level and further with SNP exposure we could not detect any changes in the γ H2AX level. In contrast, in some cases we observed modest reduction in γ H2AX level after 4 hours of SNP challenge. To better match with the graph representing the analyzed data, we have replaced the representative blot with another better representative blot. As suggested by the Referee, we also performed Griess assay to quantify the level of nitrite in EA.hy926 cells exposed to SNP for different time points ranging from 0 to 4 hours. In doing so, we detected time-dependent increase in cellular nitrite level upon SNP treatment (Supplemental Figure 1A).

However, as the Referee emphasized, we could not stop challenging this observation and therefore decided to perform an SNP time kinetics experiment in the primary endothelial cells, HUVEC. In HUVEC as well, we found a time-dependent increase in cellular nitrite level upon SNP treatment (Supplemental Figure 1E). Interestingly, γ H2AX level also time-dependently increased in HUVEC with a robust induction within 1 hour post SNP challenge which was further enhanced in 2 and 4 hours post SNP exposure (Supplemental Figure 1F). Therefore, these new data indicated that primary endothelial cells like HUVEC are more sensitive to SNP exposure than HUVEC-derived transformed endothelial cell lines such as EA.hy926 cells, which exhibited high basal levels of γ H2AX. Because, we have used many natural inducers (VEGF, Bradykinin) and external stimuli (GSNO, SNP) to impart S-nitrosylation of EZH2 in diverse cells and tissues (HUVEC, EA.hy926 cells, HEK293 cells, rat aorta tissue, and kidney tissues (as included to address the comments of Reviewer 4)), we believe that enhanced nitrite and γ H2AX observed under SNP do not compromise the natural regulation processes imparted by cellular eNOS-nitric oxide machinery to regulate EZH2 and PRC2 complex through S-nitrosylation.

2. Fig. S5B This reviewer is sorry, but the densitometric analysis (and, consequently, the discussion of the related results) of wb reporting the CHX pulse-chase experiment is not convincing. I mean, if we look at the level of EZH2 at 30' and 1hr, in the presence of CHX, it seems that the major decrease is at 30', when comparing the related GAPDH levels which is much higher at 30' vs 1hr time point, whereas EZH2 levels are basically the same. Please, correct or provide a more representative figure of the experiment.

Response: We agree with the Referee. As suggested by the Referee, we have now replaced the blot with a better representative blot.

3. Include the additional analysis of MS data, which is actually for reviewer's eyes only, as a supplemental figure.

Response: Thank you for suggesting this. We indeed also believe that this MS-based data further strengthens our Co-IP data. Therefore, as suggested by the Referee, in the revised manuscript, we have now included this data in Supplemental Figure 7A.

Response to Reviewer #4 Comments

Reviewer #4 (Remarks to the Author):

In this study, the authors identified a comprehensive mechanism for NO regulation of EZH2 and PRC2 complex. The authors concluded that S-nitrosylation of EZH2 at C329 and C700 causes changes in the structure of EZH2, which leads to its dissociation with SUZ12 for quick degradation followed by altered H3K27me3. Although the authors provided some data on the beneficial role of NO in reversing hyperglycemia-driven endothelial inflammation upon S-nitrosylation of EZH2, I am not really convinced on the physiological significance of this study. Below please find my major comments.

Response: We like to thank the Reviewer for a positive review of our manuscript and for providing insightful comments that helped us to improve the manuscript. We apologies that we could not

make the physiological significance of the study more apparent in the manuscript. However, based on the feedback provided by the Referee, we have now edited manuscript to include appropriate discussion on the physiological significance of the study. More importantly, the new experiments performed and incorporated in the revised manuscript based on the suggestion of the Referee indeed helped us to highlight the patho-physiological relevance of the study. These new experiments and associated response describing the physiological relevance of the study are discussed in the answer to next few points raised by the current Referee.

As discussed in our earlier response to Reviewer 1 query (Comment No. 11), we also performed a thorough literature review in this area and found that S-nitrosylation via endogenous nitric oxide machinery including endogenous GSNO has protective role in the cardiovascular system. For example, S-nitrosylation is a NO-based signaling mechanism regulating endothelial protein trafficking and suppression of nuclear factor κ B (NF κ B)-dependent expression of proinflammatory cytokines and adhesion molecules¹. In addition, S-nitrosylation of N-ethylmaleimide sensitive factor suppresses exocytosis of granules (i.e. Weibel–Palade bodies) and thereby externalization of the adhesion molecule P-selectin which inhibits leukocyte rolling and thus vascular inflammation². A similar mechanism is operative in platelets, reducing activation, adhesion, aggregation, and thrombosis³. Studies have demonstrated inhibitory S-nitrosylation of both NF κ B and its activating enzyme complex, inhibitory κ B kinase⁴. Enzyme such as GSNO reductase (GSNOR) selectively metabolizes endogenous GSNO thereby depleting the pool of GSNO it shifts the equilibrium and consequently limiting the levels of S-nitrosylated proteins. A knockout mouse (GSNOR^{-/-}) has been generated and manifests increased levels of S-nitrosylated proteins⁵. Moreover, GSNOR^{-/-} mice exhibit decreased cardiac infarct size and remodeling and increased cardiac function and survival after myocardial infarction⁶. Thus, S-nitrosylation mediated anti-inflammatory actions of nitric oxide are relevant to a wide range of cardiovascular disease processes, including atherosclerosis, sepsis, and autoimmune disorders. This part has been discussed in Discussion section of the manuscript in Page18-19, Line 651-666. However, because S-nitrosylation of EZH2 has never been reported, whether such cardiovascular protective effects of nitric oxide or GSNO are partly driven by the regulation of EZH2 through S-nitrosylation need to be comprehensively established through future pre-clinical and clinical studies. Herein, our new data hint in that direction by showing that S-nitrosylation of EZH2 has a vascular protective role at least in diabetes settings.

Although we agree with the Referee about the physiological significance point, however, in the present study we primarily wanted to emphasize how induction of endogenous eNOS machinery or external nitric oxide donors or S-nitrosylating agent including endogenous GSNO converges to epigenetic regulation through post-translationally modifying EZH2 protein of the PRC2 complex. Our lab is now actively engaged in analyzing how this new mechanism of regulation of EZH2 and PRC2 complex plays role in disease onset and progression.

EA.hy926 is a somatic hybrid cell line, which cannot fully reflecting the endothelial cell functions. Some key data should be further tested in primary endothelial cells.

Response: We agree with the Referee. We like to point out that in our previous submission, some of the reported experiments were carried out in HUVEC, eg. Figure 2K-L, 5E-G, Figure 5K, Supplementary Figure 2B-E. Moreover, we also wanted to highlight that the S-nitrosylation

dependent regulation of EZH2 is likely a cell type independent phenomenon (as we showed similar effect in HEK293 cells, please refer to Supplementary Figure 3D-E) and may have implications in the regulation of cellular homeostasis/pathogenesis in varieties of cell type or tissues. However, as suggested by the Referee, we repeated some of the important experiments in HUVEC to show consistency in data between immortalized endothelial cell line, EA.hy926 cells, and primary cells HUVEC. The following experiments were carried out to recapitulate the data in primary endothelial cells;

(i) We repeated the endogenous eNOS inducer VEGF time kinetics experiment in HUVEC and analyzed the level of EZH2 and H3K27me3 in these cells. As observed in EA.hy926 cells, we observed significant reduction in EZH2 level upon 2 hours of induction with VEGF whereas significant reduction in H3K27me3 was detected within 1 hour post-VEGF induction which further reduced at 2 hours. This new data has now been included as Supplemental Figure 3D.

(ii) We analyzed the levels of EZH2 and H3K27me3 in cells treated with GSNO for different time points. Again as observed in EA.hy926 cells, we observed a significant reduction in EZH2 level when HUVEC were subjected to GSNO for 2 hours. In contrast, a reduction in H3K27me3 level was detected within 1 hour of GSNO exposure which further reduced after 2 hours. These data again in parallel to the observations in EA.hy926 cells. This new data has now been included as Supplemental Figure 4E-F.

(iii) We have now performed wound healing assay using HUVEC monolayer where cells were either exposed to SNP or to SNP in combination with GSK-J4. As similar to our observation in EA.hy926 cells, wound healing analysis exhibited faster healing of wound when exposed to SNP while GSK-J4 treatment significantly abrogated SNP-induced endothelial wound healing. This new data has now been included as Figure 5C

(iv) We have previously reported a reduction in eNOS levels in endothelial cells exposed to intermittent hyperglycemia⁷. Herein, we therefore analyzed the level of eNOS and cellular nitrite level (indirect indicator of cellular nitric oxide level) using immunoblot and Griess assay respectively. We detected a significant reduction in eNOS as well as cellular nitrite level in HUVEC challenged with intermittent hyperglycemia. This new data has now been included as Figure 5F-G.

(v) Based on the next query raised by the Referee, we also performed all the eNOS siRNA based experiments in HUVEC cells (Figure 4I-K). Description of the data has been outlined in answer to the subsequent query raised by the Referee.

(vi) Based on the point raised by Reviewer 2, we also performed the analysis of cellular nitrite and γ H2AX level in HUVEC exposed to SNP for different time points including 0, 0.5, 1, 2, and 4 hours. This new data has now been included as Supplemental Figure 1E-F.

(vii) Moreover, many of the existing and/or new experiments were performed *ex vivo* with living tissues or *in vivo* in the organ level. All these data are in accordance with our observation in both EA.hy926 cells and HUVEC.

L-NAME is not an eNOS-specific inhibitor. Further experiments with eNOS siRNA (for in vitro experiment) and eNOS knockout mice (for in vivo study) would be necessary for validating the mechanism data from this study.

Response: We agree with the Referee and appreciate for suggesting the proposed experiments. We have now performed the eNOS siRNA based *in vitro* experiment and the results are discussed hereafter. We used VEGF as natural stimuli to induce eNOS-dependent nitric oxide production which we have shown to impact EZH2 and H3K27me3 levels. Through the knockdown of eNOS gene transcripts, we showed that VEGF-dependent reduction in EZH2 and H3K27me3 levels were reversed upon eNOS knockdown in HUVEC cells (Figure 4I-K). Therefore, these data suggested that eNOS plays an essential role in the VEGF-dependent downregulation of EZH2 protein level and its catalytic product H3K27me3.

Although we could perform this in vitro experiment to address the Reviewer's comment, however, because we do not have eNOS knockout mice and acquiring and housing such mice strains will require months if not years, we are unable to perform the suggested experiments in eNOS knock out mice. Because the main focus/novelty of the study was to prove how S-nitrosylation (which could be imparted by any nitric oxide producing machinery including endogenously generated GSNO) of EZH2 protein regulates the overall functioning of the PRC2 complex, we therefore believe that eNOS knockout mice study although would have been good but may not be essential at this point for the current study. Furthermore, we have included few lines in the Discussion section of the revised manuscript mentioning this as the limitation of the study in Page 19, Line 691-696. In addition, we also discussed how in disease settings where loss of eNOS or its catalytic product nitric oxide sensitize the rodent in disease onset and progression especially in diabetic kidney disease model and how this is supported by the fact of our previously published work⁷ and new data that in such disease settings both EZH2 and its catalytic products were significantly altered Figure 5O-Q.

This study used whole aorta for some experiments. It would be interesting to compare the data when endothelium layer is removed from rat aorta.

Response: We like to thank the Referee for outlining this interesting experimental plan. As suggested by the Referee, we removed the endothelial layer of the freshly isolated rat aorta using mild collagenase treatment. Upon overnight equilibration, we then exposed these aortas to bradykinin treatment as we reported in Figure 4J-K. In doing so, we detected that bradykinin failed to make any significant changes in EZH2 and H3K27me3 levels in rat aorta without an endothelium layer. Removal of the endothelial cell layer was confirmed by the absence of eNOS in endothelium free aorta and the positive signal of eNOS in the aorta having endothelium layer. This new data has now been included as Supplemental Figure 2G-H.

It is also valuable to compare the difference between healthy aorta and aorta with endothelial dysfunctions from hypertensive animals.

Response: We appreciate the point raised by the Referee. We also agree that the use of a hypertension model would have been good to show the physiological significance of the present

study. However, we believe that our diabetic model also fits the current findings in several ways. First, several others including our lab showed a dysregulation of eNOS and nitric oxide machinery in endothelial cells in hyperglycemia settings both *in vitro* and *in vivo*⁷⁻⁹. Moreover, our previous publications and that of others indicated heightened expression of EZH2 and its catalytic product in endothelial cells exposed to hyperglycemia both *in vitro* and *in vivo*^{7,10}. Diabetic kidney disease which is known as microvascular complication has been manifested with reduced of nitric oxide production which significantly contributes to disease progression¹¹⁻¹³. More importantly, eNOS knock-out mice exhibit robust DKD phenotype much earlier than WT mice having copies of eNOS gene suggesting that eNOS plays a significant role in delaying the disease onset and progression¹⁴⁻¹⁵. Therefore, we believe endothelial dysfunction in diabetic settings could be one of the models for evaluating the pathophysiological significance of the current study. In our future studies, we will certainly analyze the physiological significance of the study in the hypertension model.

However, to address the Referee's comment, we further performed the analysis of EZH2 and H3K27me3 in cortical kidney tissue lysates from non-diabetic and diabetic rats manifesting DKD phenotype. Such analysis revealed a significant increase in EZH2 protein level as well as its catalytic product H3K27me3 in cortical kidney tissue lysates prepared from rats with DKD (Figure 5Q). Interestingly though we observed a significant increase in total eNOS level (Figure 5O) in diabetic rat's kidney tissue compared to non-diabetic control, however, nitrite level as measured using Griess assay showed a significant reduction in nitrite level in diabetic rat kidney tissue lysate (Figure 5P). Such observations of increase in eNOS level yet a reduction in nitric oxide are in parallel to many previously reported studies^{11-13,16-18} that eNOS level elevated in kidney tissues, however nitric oxide level remains low. This could be due activation of eNOS though different phosphorylation or other mechanisms is required for its activity and nitric oxide production which was previously shown to be altered in hyperglycemia settings⁸. We believe that reduced nitric oxide levels in kidney tissues in diabetic rat likely be responsible for the stabilization of EZH2 protein and increase in its catalytic product as observed in our study.

Does direct modulation of EZH2 affect endothelial functions *in vitro* and *in vivo*?

Response: We have already reported this data in our earlier version of the manuscript in Supplementary Figure 10. Therein, we performed the gene expression changes and cell migration analysis in endothelial cells treated with EZH2-specific small molecule inhibitor GSK-126 or transfected with EZH2-specific siRNA. Through these new experiments, we observed comparable changes in gene expression of VEGFa, TBX20, MMP2, FGF2, and Angiopoietin-2 upon either GSK-126 exposure or knockdown of EZH2 through EZH2-specific siRNA. In addition, as reported earlier in endothelial cells⁷, inhibition of EZH2 by GSK-126 or EZH2 siRNA caused enhanced endothelial cell migration. All these data can be found in Supplementary Figure 10 of the manuscript.

Response to Reviewer #5 Comments

Reviewer #5 (Remarks to the Author):

The authors presented an interesting set of experimental data using a NO donor (500uM SNP) and a transnitrosylating agent GSNO demonstrating that SNO at Cys329/700 in EZH2

results in SUZ12 dissociation from EZH2 bound PRC2 complex, sequentially reducing its methyltransferase activity, preventing its nuclear localization, and affecting its stability. The authors also made an extensive rebuttal to thoroughly address the questions from previous reviewers. Overall, their data in essence support the conclusion that SNO-EZH2 serves as novel epigenetic mechanism to mediate NO signaling regulation of endothelial cell homeostasis.

Response: We would like to thank the Reviewer for appreciating our efforts to revise the manuscript based on the comments raised by previous Referees.

While the study is of significance to advance NO signaling and it is logical to first use GPS-SNO prediction tool and simulation to identify the two SNO sites Cys329/700 in EZH2 and followed by mutations to functionally study SNO in EZH2, however, the conclusion will be solidified if these Cysteines are indeed the physiological NO-responsive SNO-sites in EZH2. The authors performed MS studies for determining EZH2-interacting proteins. It is kind of a surprise that they did not do MS identification and/or verification of the two sites in control and GSNO-treated endothelial cells or artery tissues, which would have made the pathway physiologically relevant.

Response: We would like to thank the Referee for bringing this point. We agree that we have not utilized MS-based confirmation technique, however, we now performed S-nitrosylation analysis of EZH2 WT and EZH2 C329S C700S mutant upon GSNO treatment in cells overexpressing these forms using IodoTMT technique. In so doing, we found that double mutation of EZH2 at C329 and C700 caused robust and significant reduction in S-nitrosylation of EZH2 compared to WT variant of EZH2 when exposed to GSNO. Therefore, this new data clearly indicates that these two sites are indeed S-nitrosylated. This new data has now been included as Supplemental Figure 12C. In addition, we also like to mention that in the manuscript (Figure 6A, in previous form of the manuscripts and in current submission), we already reported the loss of GSNO dependent S-nitrosylation signal of all EZH2 mutant carrying mutation at C260 or C329 or C700 sites compared to EZH2 WT variant using Biotin Switch Assay protocol.

Moreover, we have also validated this prediction with a recently introduced prediction tools named pLMSNOSite¹⁹. Therefore, these data altogether attest that all the predicted sites of C260, C329, and C700 in EZH2 could get S-nitrosylated, however, in our study, we could find that S-nitrosylation of C329 and C700 but not C260 lead to functional/structural changes in EZH2 protein thereby affecting its activity and downstream signaling regulated by this enzyme.

References

1. Lowenstein, C. J. Nitric oxide regulation of protein trafficking in the cardiovascular system. *Cardiovasc. Res.* 75, 240–246 (2007).
2. Matsushita, K. et al. Nitric oxide regulates exocytosis by S-nitrosylation of N-ethylmaleimide-sensitive factor. *Cell* 115, 139–150 (2003).
3. Morrell, C. N. et al. Regulation of platelet granule exocytosis by S-nitrosylation. *Proc. Natl. Acad. Sci. U. S. A.* 102, 3782–3787 (2005).

4. Reynaert, N. L. et al. Nitric oxide represses inhibitory kappaB kinase through S-nitrosylation. *Proc. Natl. Acad. Sci. U. S. A.* 101, 8945–8950 (2004).
5. Liu, L. et al. Essential roles of S-nitrosothiols in vascular homeostasis and endotoxic shock. *Cell* 116, 617–628 (2004).
6. Lima, B. et al. Endogenous S-nitrosothiols protect against myocardial injury. *Proc. Natl. Acad. Sci. U. S. A.* 106, 6297–6302 (2009).
7. Thakar S. et al. Intermittent High Glucose Elevates Nuclear Localization of EZH2 to Cause H3K27me3-Dependent Repression of KLF2 Leading to Endothelial Inflammation. *Cells*. 2021 Sep 26;10(10):2548.
8. Du X.L. et al. Hyperglycemia inhibits endothelial nitric oxide synthase activity by posttranslational modification at the Akt site. *J Clin Invest.* 2001 Nov;108(9):1341-8.
9. Noyman I. et al. Hyperglycemia reduces nitric oxide synthase and glycogen synthase activity in endothelial cells. *Nitric Oxide.* 2002 Nov;7(3):187-93.
10. Sánchez-Ceinos J. et al. Repressive H3K27me3 drives hyperglycemia-induced oxidative and inflammatory transcriptional programs in human endothelium. *Cardiovasc Diabetol.* 2024 Apr 5;23(1):122.
11. Craven PA, Studer RK, DeRubertis FR. Impaired nitric oxide-dependent cyclic guanosine monophosphate generation in glomeruli from diabetic rats. Evidence for protein kinase C-mediated suppression of the cholinergic response. *J Clin Invest.* 1994 Jan;93(1):311-20.
12. Satoh M. et al. NAD(P)H oxidase and uncoupled nitric oxide synthase are major sources of glomerular superoxide in rats with experimental diabetic nephropathy. *Am J Physiol Renal Physiol.* 2005 Jun;288(6):F1144-52.
13. Craven PA, Studer RK, DeRubertis FR. Impaired nitric oxide release by glomeruli from diabetic rats. *Metabolism.* 1995 Jun;44(6):695-8.
14. Yuen D.A. et al. eNOS deficiency predisposes podocytes to injury in diabetes. *J Am Soc Nephrol.* 2012 Nov;23(11):1810-23.
15. Nakagawa T. et al. Diabetic endothelial nitric oxide synthase knockout mice develop advanced diabetic nephropathy. *J Am Soc Nephrol.* 2007 Feb;18(2):539-50.
16. Veelken R. et al. Nitric oxide synthase isoforms and glomerular hyperfiltration in early diabetic nephropathy. *J Am Soc Nephrol.* 2000 Jan;11(1):71-79.
17. Sugimoto H. et al. Increased expression of endothelial cell nitric oxide synthase (ecNOS) in afferent and glomerular endothelial cells is involved in glomerular hyperfiltration of diabetic nephropathy. *Diabetologia.* 1998 Dec;41(12):1426-34.
18. Choi K.C. et al. Alterations of intrarenal renin-angiotensin and nitric oxide systems in streptozotocin-induced diabetic rats. *Kidney Int Suppl.* 1997 Sep;60:S23-7.
19. Pratyush P, Pokharel S, Saigo H, Kc DB. pLMSNOSite: an ensemble-based approach for predicting protein S-nitrosylation sites by integrating supervised word embedding and embedding from pre-trained protein language model. *BMC Bioinformatics.* 2023 Feb 8;24(1):41.

Answer to the Reviewer Comments

Reviewer #2 (Remarks to the Author):

Dear Authors, this reviewer is satisfied by your responses. However, your MS still lacks accuracy. For example, legend of Supplementary figure 10 contains errors. Indeed, you report that GSK-126 treatment is in panel D and EZH2 siRNA in panels E and F, but what you show in the picture is exactly the contrary. Please, carefully check all the figure legends and the correspondance of the results with the figures.

Answer: We would like to thank the Reviewer for thorough review of our manuscript and pointing out certain discrepancies within the legend section of the manuscript. We have corrected the legend of the Supplementary Figure 10. Based on the Referee's feedback, we carefully reviewed all the Figures and associated legends and corrected any such discrepancies if found.

Reviewer #4 (Remarks to the Author):

The authors have properly addressed my previous comments with careful revision and additional experiments.

Answer: We thank the reviewer for the positive evaluation of the revision.

Reviewer #5 (Remarks to the Author):

No additional comments

Answer: We thank the reviewer for the positive evaluation of the revision.